# Integrative Taxonomy Revealed Cryptic Diversity in the West African Grasshopper Genus *Serpusia* Karsch, 1891 (Orthoptera: Catantopinae)

**DOI:** 10.3390/insects16101020

**Published:** 2025-10-01

**Authors:** Jeanne Agrippine Yetchom Fondjo, Alain Christel Wandji, Reza Zahiri, Oliver Hawlitschek, Claudia Hemp

**Affiliations:** 1Staatliches Museum für Naturkunde Karlsruhe, 76133 Karlsruhe, Germany; reza.zahiri@smnk.de; 2Laboratory of Zoology, The University of Yaoundé 1, Yaoundé P.O. Box 812, Cameroon; wandjichristel@gmail.com; 3Department of Evolutionary Biology and Environmental Studies, University of Zurich, CH-8057 Zurich, Switzerland; oliver.hawlitschek@uzh.ch; 4Department of Plant Systematics, University of Bayreuth, 95440 Bayreuth, Germany; claudia.hemp@uni-bayreuth.de

**Keywords:** systematics, molecular phylogeny, short-horned grasshopper, tropical Africa

## Abstract

**Simple Summary:**

*Serpusia* is a wingless grasshopper genus that is restricted to African Rainforests. *Serpusia* species exhibit largely external resemblance to species of several related genera, including *Aresceutica*, *Pseudophialosphera*, *Segellia*, *Auloserpusia*, *Pteropera*, *Coenona*, *Serpusilla*, and *Veseyacris.* The taxonomic status of *Serpusia* remains challenging, and its classification is unstable. Despite the alarming ongoing habitat loss in the African Rainforests, this genus has never been revised using modern molecular techniques. In this study, we used both morphological and molecular approaches to provide insights into the current diversity of this genus. We sequenced two mitochondrial markers (COI-5P and 16S rDNA) and constructed the phylogenetic relationships among species of *Serpusia* and closely related genera. It appeared that *Serpusia* is more diverse than what is currently known, encompassing five species, four of which we described in this study. In addition, one of the previously known species of *Serpusia*, the so-called S. succursor, is genetically very distinct from the true *Serpusia* and also encompasses hidden species. We then established a new genus, *Paraserpusia*, to accommodate *S. succursor* and its hidden species, which were formally described in this study. Overall, this study highlights the importance of combining molecular and morphological data to uncover and formally describe previously underestimated groups.

**Abstract:**

Background/Objectives: Despite their ecological significance, DNA barcoding data for African rainforest Orthoptera remain underrepresented globally, limiting progress in species discovery, biodiversity assessment, and conservation. This study aimed to generate molecular data for morphologically identified *Serpusia* Karsch, 1891 species to evaluate their taxonomic status. Methods: Specimens were collected from multiple sites in Cameroon and analyzed using DNA barcoding with COI-5P and 16S rDNA markers. Species delimitation was performed with Automatic Barcode Gap Discovery, and phylogenetic relationships were inferred using Maximum Likelihood and Bayesian Inference. Additionally, external morphology and the male phallic complex were examined. Results: Molecular analyses delineated 19 MOTUs, five corresponding to *Serpusia opacula*, seven to *Serpusia succursor* and the remainder to outgroups. Similarity-based assignments matched these MOTUs to 19 BINs. Phylogenetic reconstruction revealed *S. opacula* and *S. succursor* as two genetically distinct clades, with the *S. opacula* group more closely related to *Aresceutica* Karsch, 1896 than to the *S. succursor* group. Accordingly, we established a new genus, *Paraserpusia* gen. nov., to accommodate *S. succursor*. Within the *S. opacula* group, five species are recognized: one previously described (*S. opacula*) and four new species (*S. kennei* sp. nov., *S. missoupi* sp. nov., *S. seinoi* sp. nov., and *S. verhaaghi* sp. nov.). The former *S. succursor*, now *Paraserpusia succursor*, is divided into six well-supported lineages, five of which are formally described here (*P. hoeferi* sp. nov., *P. husemanni* sp. nov., *P. kekeunoui* sp. nov., *P. tamessei* sp. nov., and *P. tindoi* sp. nov.). A haplotype network based on COI-5P sequences corroborates three major clades corresponding to the *S. opacula* group, the *S. succursor* group, and *Aresceutica*. Diagnostic morphological differences between *Serpusia* and *Paraserpusia* are consistently supported across characters. Conclusions: This integrative approach reveals substantial hidden diversity within *Serpusia* and highlights the importance of combining molecular and morphological data to uncover and formally describe previously overlooked taxa.

## 1. Introduction

The genus *Serpusia* Karsch, 1891 was established, with *S. opacula* Karsch, 1891 designated as the type species. It was described alongside several morphologically similar genera, including *Pteropera* Karsch, 1891; *Segellia* Karsch; and *Apoboleus* Karsch, 1891. According to the Orthoptera Species File (OSF), five species are currently recognized from West-Central Tropical African forests: *S. opacula*, restricted to Cameroon and Nigeria; *S. catamita* Karsch, 1893, distributed across Togo, Ghana, Ivory Coast, Liberia, Guinea-Conakry and Sierra Leone; *S. succusor* (Karsch, 1896), occurring in the Democratic Republic of the Congo, Cameroon, Nigeria, and Togo; *S. blanchardi* (Bolívar, 1905), reported from Liberia; and *S. inflata* (Ramme, 1929), found in Liberia, Guinea, and Sierra Leone [1]. *Serpusia* is a flightless genus of catantopine grasshoppers, typically abundant in both primary and secondary forests [2]. Most species exhibit marked sexual dimorphism, with males typically much smaller than females, which tend to be more robust [3]. The taxonomic position of *Serpusia* remains disputed, as its evolutionary relationships to other African rainforest grasshoppers are unresolved, and its classification is unstable. This uncertainty stems largely from its external resemblance to species of several related genera, including *Aresceutica*, *Pseudophialosphera*, *Segellia*, *Auloserpusia*, *Pteropera*, *Coenona*, *Serpusilla*, and *Veseyacris* [4,5,6].

Based on external morphology and male internal genitalia, the close relationships among *Auloserpusia*, *Aresceutica*, *Serpusia*, and *Pseudophialosphera* were discussed, and the possibility of synonymizing *Serpusia* partly with *Auloserpusilla* and partly with *Aresceutica* was suggested [5]. *Ptemoblax insidiosus* Bolívar, 1908 was further synonymized with *S. succusor* (Karsch, 1896) by [6], and *Ptemoblax lemarineli* Bolívar, 1911 was assigned to *Serpusia* by [7]. However, the same species has subsequently been treated as *S. lemarineli* (Bolívar, 1911) by [8,9]. While revising the East African genus *Aresceutica*, *Ptemoblax lemarineli* (*S. lemarineli*) Bolívar, 1911 was formally transferred as a new combination to the genus *Aresceutica* based on details of the phallic morphology [4]. Additionally, the aforementioned authors proposed removing *S. catamita*, *S. blanchardi*, and *S. inflata* from *Serpusia* and placing them in a new genus, thereby restricting *Serpusia* to only two species: *S. opacula* and *S. succusor*. For nearly a century, reliance on a limited set of morphological characters has led to an underestimation of the diversity within this West African grasshopper genus. Despite the morphological difficulty of distinguishing *Serpusia* species from closely related genera, the genus *Serpusia* has never been comprehensively revised, nor has any interspecific molecular delineation been undertaken to date.

The present study applied the distance-based species delimitation method Automatic Barcode Gap Detection (ABGD), together with similarity-based analyses, molecular phylogenetics, and morphological examinations, to assess the poorly studied African rainforest grasshopper genus *Serpusia* and its close relatives. Our aim was to quantify genetic divergence and test for cryptic or undescribed diversity within morphologically defined taxa. By integrating morphological characters with sequence data, we identified genetically and morphologically diagnosable species that had previously been overlooked or misidentified. Based on these findings, we provided identification keys to distinguish valid genera and species and formally described the newly recognized taxa.

## 2. Materials and Methods

### 2.1. Specimen Collection and Morphological Studies

A total of 101 specimens belonging to the morphologically identified *S. opacula* and *S. succusor* groups were examined. These specimens were collected between 2016 and 2025 from various locations in Cameroon. Additional material of the specimens of *S. catamita* housed at the Staatliches Museum für Naturkunde Stuttgart, Germany (SMNK) was also studied. Photographs of *S. blanchardi* and *S. inflata* were kindly provided by the Museo Nacional de Ciencias Naturales, Madrid (MNCN) and the Natural History Museum, London (NHM), respectively, and were examined for comparison. Dissections and preparations of male genitalia followed the standard methodologies of [10,11]. Terminology for male genitalia and female spermatheca is consistent with the conventions established by [12,13]. Selected specimens were subsequently used for DNA analysis (Table 1). All examined material is preserved as vouchers in the entomological collection at the Staatliches Museum für Naturkunde Karlsruhe, Germany (SMNK).

Habitus photographs of selected specimens were taken at the Zoologisches Museum Hamburg, Germany (ZMH), using a high-resolution DUN Inc. stacking system (DUN Inc., Visalia, CA, USA). In addition, images of male and female genitalia were captured at the SMNK Karlsruhe using a Keyence VHX-7000 digital microscope (Keyence, London, UK).

Measurements were obtained using a digital calliper (precision 0.01 mm). All measurements (Table 2) are expressed in millimetres (mm). Male and female specimens were measured separately. For each specimen, the following traits were recorded: Head’s length (HeadL), from the anterior end of the vertex to the anterior margin of the pronotum; Head’s width (HeadW), distance between the outer edges of the eyes, measured from the front or from above; Antenna’s length (AntenL), from the scape to the tip of antenna; Interocular distance (I.O.D.), measured dorsally; Fastigium length (FastigL), dorsally, from a line joining the most anterior profile of the eyes to the tip of the fastigium; Pronotum length, along the midline (PronotL); Pronotum width (PronotW), at the lateral basal lobes of the pronotum; Tegmina length (TegL), from the confluence of the C and Sc veins to the apex; Hind tibia length (TL), excluding the condyle, measured from the base to the dorsal insertion of the external spurs; Hind femur (FL), from its base to the tip of the knee; Width of the hind femur (FW), distance between the two parallel lines running through the dorsal and ventral extremities, aligned with the long axis of the femur; Body length (BodyL), from the tip of the head to the posterior end of the abdomen. For each body part, mean values and standard deviation (SD) were calculated (see Table 2). Distributional data were obtained from geographical coordinates recorded during fieldwork and from the Orthoptera Species File (OSF) website. A distribution map of all species was generated in QGIS 3.4.12.


*Depositories*


MNCN Museo Nacional de Ciencias Naturales Madrid, Spain

NHM Natural History Museum, London, UK

SMNK Staatliches Museum für Naturkunde Karlsruhe, Germany

SMNK Staatliches Museum für Naturkunde Stuttgart, Germany.

The measurements of specimens examined in this study are presented in Table 2 below.

**Table 2 insects-16-01020-t002:** Measurements in millimeters (mm) of the examined *Serpusia* and *Paraserpusia* species currently known from Cameroon.

Species	*S. kennei* sp. nov.		*S. missoupi* sp. nov.		*S. opacula* Karsch, 1891		*S. seinoi* sp. nov.	
Parameters	Male	Female	Male	Female	Male	Female	Male	Female
HeadL	2.87 ± 0.01 (n = 2)	3.29 ± 0.24 (n = 4)	3.19 (n = 1)	3.80 ± 40 (n = 3)	3.03 ± 0.67 (n = 2)	3.55 ± 0.47 (n = 2)	–	3.38 ± 0.23 (n = 2)
HeadW	3.85 ± 0.10 (n = 2)	4.24 ± 0.24 (n = 4)	3.42 (n = 1)	4.03 ± 0.51 (n = 3)	3.57 ± 0.35 (n = 2)	4.76 ± 0.06 (n = 2)	–	4.48 ± 0.04 (n = 2)
AntenL	11.43 ± 0.10 (n = 2)	11.81 ± 0.93 (n = 4)	12.36 (n = 1)	11.40 (n = 1)	11.72 ± 0.93 (n = 2)	12.55 ± 1.48 (n = 2)	–	12.50 ± 0.72 (n = 2)
I.O.D.	0.65 ± 0.21 (n = 2)	1.13 ± 0.24 (n = 4)	0.79 (n = 1)	1.06 ± 0.25 (n = 3)	0.75 ± 0.07 (n = 2)	1.10 ± 0.40 (n = 2)	–	1.11 ± 0.41 (n = 2)
PronotL	4.55 ± 0.12 (n = 2)	5.39 ± 0.08 (n = 4)	4.38 (n = 1)	5.56 ± 0.15 (n = 3)	4.55 ± 0.57 (n = 2)	5.93 ± 0.34 (n = 2)	–	5.82 ± 0.04 (n = 2)
PronotW	3.35 ± 0.06 (n = 2)	4.40 ± 0.24 (n = 4)	3.80 (n = 1)	4.31 ± 0.39 (n = 3)	3.34 ± 0.28 (n = 2)	4.85 ± 0.58 (n = 2)	–	4.85 ± 0.00 (n = 2)
TegL	3.52 ± 0.18 (n = 2)	4.60 ± 0.22 (n = 4)	3.89 (n = 1)	4.55 ± 0.22 (n = 3)	3.52 ± 0.66 (n = 2)	5.29 ± 0.06 (n = 2)	–	4.23 ± 0.64 (n = 2)
TL	11.65 ± 0.92 (n = 2)	14.10 ± 0.36 (n = 4)	11.36 (n = 1)	14.21 ± 0.50 (n = 3)	11.70 ± 0.42 (n = 2)	15.74 ± 0.64 (n = 2)	–	14.47 ± 0.07 (n = 2)
FL	13.17 ± 1.29 (n = 2)	15.83 ± 0.32 (n = 4)	12.88 (n = 1)	16.03 ± 0.56 (n = 3)	13.57 ± 0.51 (n = 2)	16.92 ± 1.68 (n = 2)	–	16.20 (n = 1)
FW	2.99 ± 0.33 (n = 2)	3.96 ± 0.25 (n = 4)	3.23 (n = 1)	3.95 ± 45 (n = 3)	3.17 ± 0.45 (n = 2)	4.32 ± 0.16 (n = 2)	–	3.90 (n = 1)
BodyL	20.20 ± 0.49 (n = 2)	26.09 ± 0.10 (n = 4)	21.33 (n = 1)	26.17 ± 2.10 (n = 3)	20.27 ± 0.98 (n = 2)	27.53 ± 2.24 (n = 2)	–	26.89 ± 0.54 (n = 2)
**Species**	***Serpusia verhaaghi* sp. nov.**		***Paraserpusia hoeferi* sp. nov.**		***Paraserpusia husemanni* sp. nov.**		***Paraserpusia kekeunoui* sp. nov.**	
**Parameters**	**Male**	**Female**	**Male**	**Female**	**Male**	**Female**	**Male**	**Female**
HeadL	3.05 ± 0.24 (n = 5)	3.80 ± 0.38 (n = 6)	3.26 ± 0.55 (n = 3)	4.19 ± 0.14 (n = 2)	3.26 ± 0.11 (n = 5)	4.17 ± 0.38 (n = 7)	2.41 ± 0.39 (n = 5)	2.84 ± 0.34 (n = 6)
HeadW	3.94 ± 0.36 (n = 5)	4.47 ± 0.20 (n = 6)	3.80 ± 0.27 (n = 3)	4.69 ± 0.11 (n = 2)	3.61 ± 0.10 (n = 5)	4.97 ± 0.29 (n = 7)	2.95 ± 0.13 (n = 5)	4.01 ± 0.17 (n = 6)
AntenL	10.71 ± 0.35 (n = 5)	12.05 ± 1.08 (n = 4)	10.34 ± 0.15 (n = 2)	11.35 ± 1.04 (n = 2)	10.87 ± 0.40 (n = 4)	13.28 ± 1.29 (n = 7)	8.51 ± 0.94 (n = 5)	8.51 ± 0.76 (n = 4)
I.O.D.	0.83 ± 0.07 (n = 5)	0.98 ± 0.22 (n = 6)	0.81 ± 0.37 (n = 3)	1.37 ± 0.01 (n = 2)	1.12 ± 0.15 (n = 5)	1.39 ± 0.12 (n = 7)	0.33 ± 0.05 (n = 5)	0.17 ± 0.15 (n = 6)
PronotL	4.16 ± 0.37 (n = 5)	5.75 ± 0.28 (n = 6)	4.80 ± 0.27 (n = 3)	6.42 ± 0.44 (n = 2)	4.96 ± 0.47 (n = 5)	7.05 ± 0.23 (n = 7)	3.84 ± 0.26 (n = 5)	5.36 ± 0.41 (n = 6)
PronotW	3.39 ± 0.28 (n = 5)	4.86 ± 0.45 (n = 6)	3.25 ± 0.39 (n = 3)	4.66 ± 0.36 (n = 2)	3.79 ± 0.13 (n = 5)	4.86 ± 0.23 (n = 7)	2.50 ± 0.43 (n = 5)	3.64 ± 0.34 (n = 6)
TegL	4.09 ± 0.14 (n = 5)	5.02 ± 0.55 (n = 6)	4.05 ± 0.62 (n = 3)	5.79 ± 1.11 (n = 2)	4.01 ± 0.65 (n = 5)	5.80 ± 0.57 (n = 7)	3.72 ± 0.54 (n = 5)	4.15 ± 0.69 (n = 6)
TL	11.73 ± 0.96 (n = 5)	14.65 ± 0.58 (n = 6)	12.06 ± 0.42 (n = 3)	15.64 ± 0.13 (n = 2)	11.96 ± 1.01 (n = 5)	15.73 ± 0.91 (n = 7)	10.22 ± 0.40 (n = 5)	13.96 ± 0.55 (n = 6)
FL	13.05 ± 0.68 (n = 5)	16.16 ± 0.61 (n = 6)	13.55 ± 0.29 (n = 3)	17.42 ± 0.03 (n = 2)	13.49 ± 0.92 (n = 5)	17.63 ± 0.43 (n = 7)	11.24 ± 0.72 (n = 5)	16.00 ± 0.86 (n = 6)
FW	3.16 ± 0.14 (n = 5)	4.10 ± 0.24 (n = 6)	3.26 ± 0.09 (n = 3)	3,99 ± 0.18 (n = 2)	3.22 ± 0.39 (n = 5)	4.31 ± 0.30 (n = 7)	2.41 ± 0.12 (n = 5)	3.17 ± 0.16 (n = 6)
BodyL	20.11 ± 1.23 (n = 5)	27.04 ± 1.63 (n = 6)	20.15 ± 1.09 (n = 3)	27.25 ± 0.08 (n = 2)	21.11 ± 1.11 (n = 5)	30.60 ± 1.76 (n = 7)	18.74 ± 2.27 (n = 5)	25.54 ± 2.51 (n = 6)
**Species**	** *Paraserpusia succursor* **		***Paraserpusia tamessei* sp. nov.**			***Paraserpusia tindoi* sp. nov.**		
**Parameters**	**Male**	**Female**	**Male**		**Female**	**Male**	**Female**	
HeadL	2.79 ± 0.32 (n = 9)	3.52 ± 0.12 (n = 6)	2.99 ± 0.40 (n = 11)		3.39 ± 0.47 (n = 5)	3.30 ± 0.26 (n = 8)	4.40 ± 0.36 (n = 4)	
HeadW	3.49 ± 0.12 (n = 9)	4.58 ± 0.27 (n = 6)	3.51 ± 0.22 (n = 11)		4.58 ± 0.24 (n = 5)	3.44 ± 0.19 (n = 8)	4.43 ± 0.24 (n = 4)	
AntenL	10.12 ± 0.62 (n = 6)	10.88 ± 0.52 (n = 5)	9.50 ± 0.86 (n = 11)		10.42 ± 0.77 (n = 5)	11.39 ± 0.82 (n = 8)	12.32 ± 0.59 (n = 4)	
I.O.D.	0.84 ± 0.15 (n = 9)	1.18 ± 0.18 (n = 6)	0.93 ± 0.07 (n = 11)		1.03 ± 0.15 (n = 5)	0.87 ± 0.17 (n = 8)	1.23 ± 0.20 (n = 4)	
PronotL	4.82 ± 0.28 (n = 9)	6.18 ± 0.18 (n = 6)	4.63 ± 0.28 (n = 11)		6.23 ± 0.35 (n = 5)	4.54 ± 0.59 (n = 8)	6.13 ± 0.11 (n = 4)	
PronotW	3.36 ± 0.13 (n = 9)	4.55 ± 0.24 (n = 6)	3.17 ± 0.26 (n = 11)		4.39 ± 0.3 (n = 5)	3.18 ± 0.28 (n = 8)	4.48 ± 0.12 (n = 4)	
TegL	3.92 ± 0.27 (n = 9)	5.12 ± 0.62 (n = 6)	4.14 ± 0.28 (n = 11)		5.42 ± 0.81 (n = 5)	3.81 ± 0.31 (n = 8)	4.75 ± 0.28 (n = 4)	
TL	11.19 ± 0.37 (n = 9)	14.67 ± 0.73 (n = 6)	10.59 ± 0.86 (n = 11)		14.37 ± 1.34 (n = 5)	11.09 ± 0.47 (n = 8)	14.98 ± 0.43 (n = 4)	
FL	12.38 ± 0.25 (n = 9)	16.30 ± 0.70 (n = 6)	11.85 ± 0.97 (n = 11)		16.63 ± 1.82 (n = 5)	12.43 ± 0.71 (n = 8)	16.38 ± 0.32 (n = 4)	
FW	3.09 ± 0.15 (n = 9)	3.91 ± 0.15 (n = 6)	2.80 ± 0.41 (n = 11)		4.08 ± 0.31 (n = 5)	3.02 ± 0.18 (n = 8)	3.54 ± 0.42 (n = 4)	
BodyL	19.79 ± 0.59 (n = 9)	26.55 ± 1.78 (n = 6)	19.29 ± 1.58 (n = 11)		24.76 ± 2.87 (n = 5)	18.61 ± 0.51 (n = 8)	25.29 ± 1.69 (n = 4)	

### 2.2. Molecular Analyses

#### 2.2.1. Taxon Sampling

A total of 84 terminals, including 15 outgroup and 69 ingroup specimens, were used in the phylogenetic analysis (Table 1). The ingroup taxa were selected based on the characteristics of their external morphology and their distribution range.

The ingroup taxa comprise 11 species, of which two were previously known, while the remaining nine represent hidden species, newly described in the present study. Regarding the outgroup taxa, the study included six catantopine species, from three genera: *Aresceutica morogorica* Dirsh, 1984, *A. nguruensis* Rowell, Jago & Hemp, 2018, *A. subnuda* Karsch, 1896, *Pteropera carnapi* Ramme, 1929, *P. kennei* Yetchom & Husemann, 2024, and *Segellia nitidula* Karsch, 1891. *Segellia nitidula* was used as the root of the phylogeny.

#### 2.2.2. DNA Extraction, PCR Amplification, and Sequencing

Genomic DNA was isolated from the femoral muscle tissue of each grasshopper specimen using a high-salt extraction protocol [15] and a magnetic bead-based DNA extraction protocol [16]. Polymerase chain reaction (PCR) was employed to amplify two molecular markers: the mitochondrial genes COI-5P and 16S. The barcoding region of the mitochondrial cytochrome c oxidase subunit I (COI-5P) was amplified using the primer pairs LCO (5′-GGTCAACAAATCATAAAGATATTGG) and HCO (5′-AAACTTCAGGGTGACCAAAAAATCA) [17]. The partial fragments of the mitochondrial ribosomal gene 16S were amplified using the primer pairs 16S-F (5′-CGCCTGTTTATCAAAAACAT-) and 16S-R (5′-CCGGTCTGAACTCAGATCACGT-) [18]. PCR amplifications were carried out using MangoTaq polymerase with Loading Dye, following the manufacturer’s instructions.

PCR amplifications were performed under the following cycling conditions: an initial denaturation for 3 min at 94 °C, followed by 35 cycles of 30 s at 95 °C, a primer-specific annealing step of 45 s (COI: 50 °C, 16S: 61 °C), and an extension step of 1 min at 72 °C, with a final extension of 10 min at 72 °C. The resulting PCR amplicons were separated by electrophoresis on 1% agarose gels, subsequently stained with 0.025% Ethidium Bromide solution. Samples showing successful amplification were purified using an ExoSap Enzyme cocktail (VWR, Radnor, PA, USA). To ensure sequence accuracy, both DNA strands were sequenced by Macrogen Europe (Amsterdam, The Netherlands). Specimen collection data, consensus sequences, and sequence trace files were uploaded to the Barcode of Life Data System (BOLD) and are publicly available under the project entitled DNA Barcoding of African Forest Grasshoppers (DBAGF). All newly generated DNA sequences were deposited in GenBank under the accession numbers listed in Table 1.

#### 2.2.3. Data Analysis

The DNA sequences were assembled into contigs, trimmed, and aligned in Geneious Prime v2024.0.7 [19] using the MUSCLE algorithm [20]. To confirm species identity and hence for potential contamination, sequences were compared using the “BLAST” tool on the NCBI database (https://blast.ncbi.nlm.nih.gov/Blast.cgi; accessed on 7 May 2025). The aligned sequences were further visualized in Mesquite (http://www.mesquiteproject.org; accessed on 3 May 2025) [21]. Potential pseudogenes (numts) were investigated by translating the sequences into amino acids, using the invertebrate mitochondrial code and inspecting them for indels.

#### 2.2.4. Phylogenetic Analysis

To determine the phylogenetic positions of the genus *Serpusia* and to investigate its relationships with related genera, both maximum likelihood (ML) and Bayesian inference (BI) analyses were conducted using a concatenated dataset of COI-5P and 16S rDNA. In addition to the newly generated sequences from this study, published COI-5P and 16S rDNA sequences of closely related species selected as outgroups within the subfamily Catantopinae were retrieved from GenBank database (http://www.ncbi.nlm.nih.gov/; accessed on 7 May 2025). Maximum likelihood analyses were conducted using the IQ-TREE v. 1.6.12 [22], bootstrap support estimated from 1000 replicates and the approximate likelihood ratio test (SH-aLRT) [23]. BI analysis was performed using MrBAYES v. 3.2.7a [24] for 2 million generations, sampling 10,000 trees every 100 generations. The first 25% of samples were discarded as burn-in. Convergence was confirmed by mean split frequencies below 0.01. The resulting phylogenetic trees were visualized using FigTree v. 1.4.2 (https://github.com/rambaut/figtree/releases, accessed on 7 May 2025 [25]).

#### 2.2.5. Distance-Based Barcode Analysis to Delineate Species

In addition to the phylogenetic analyses, species delimitation was performed using the Automatic Barcode Gap Discovery (ABGD) method [26], a distance-based approach. The web interface available at https://github.com/iTaxoTools/ABGDpy (accessed on 29 May 2025) was used to identify groups of individuals potentially corresponding to species in the genus *Serpusia*. The maximum intraspecific divergence (Pmax) was set between 0.001 and 0.1, the relative gap width was set at 0.9, and a total of 20 recursive steps were applied within the primary partitions. The minimum relative gap width was left at the default value of 1.20. Pairwise distances were calculated using the K2P substitution model. This analysis was applied separately to the COI-5P and the 16S rDNA individually.

Pairwise sequence divergence among barcode sequences was calculated at species, genus and family levels using the BOLD Aligner (Amino Acid based HMM) implemented in BOLD systems [27], with Kimura’s two-parameter (K2P) distance model for COI-5P, the standard animal barcode marker. Genetic distances were summarized using the Distance Summary and Barcode Gap Analysis tools in BOLD. The barcode gap analysis provides a comprehensive overview of the distribution of the intraspecific distances as well as the distance to the nearest neighbour (NN) for each species. The existence of a barcode gap is determined through rigorous testing of species, where the NN distance represents the genetic distance between a species and its closest congeneric relative. Haplotypes diversity was assessed by constructing a median-joining haplotype network using the POPART [28].

## 3. Results

### 3.1. Molecular Analysis

#### 3.1.1. Phylogenetic Analyses

All the specimens examined morphologically were subjected to genetic analysis, and only the successfully sequenced ones were used for the genetic analysis. A total of 84 COI-5P sequences, each 658 bp in length, were incorporated into the phylogenetic analysis. Of these, 78 were newly generated from six morphospecies, including nine sequences from four outgroup species and 69 sequences from two ingroup morphospecies (*Serpusia opacula* and *Serpusia succursor*). Six sequences of *Pteropera* generated in our previous work [14] were retrieved from GenBank and included in the analyses. Furthermore, a total of 73 sequences of 16S rDNA with a length of 501 bp were newly generated, and four other sequences of this gene were downloaded from GenBank and included in the analyses. ML and BI analyses were performed on the combined dataset (COI-5P, 16S rDNA) comprising 1159 bp, yielding comparable topologies. The molecular analysis generated a substantial number of MOTUs, assigning the ingroup taxa to two main clades, namely the *S. opacula* complex and the *S. succursor* complex, each of which comprises strongly supported subclades (Figure 1). The *S. opacula* clade exhibited a stronger affinity with *Aresceutica* than with the *S. succursor* clade. It is for this reason that *S. succursor* is removed from the genus *Serpusia* and included in a new genus, *Paraserpusia* gen. nov.

The phylogenetic tree (Figure 1) shows that *Aresceutica*, *Serpusia*, and *Paraserpusia* gen. nov. are well-supported monophyletic genera (PP = 1; BS = 100). The *S. opacula* complex was included in a large clade divided into five species, one of which corresponds to the previously known *S. opacula*, and four are newly described in the present work. The five species under consideration were found to be distributed across five distinct subclades. The first subclade, comprising *S. opacula* populations observed in Bissoue and Njuma (CMJ699, CMJ708, CMJ709, CMJ711, and CMJ710), received significant support (PP = 1; BS = 100) and was designated as the sister of the *S. opacula* population present in Somalomo (CMJ1110, CMJ1111), which also received significant support (PP = 1; BS = 100). Additionally, the Mouanko population (CMJ520, CMJ726, CMJ727), which is likely the nominative species *S. opacula*, was recovered with high support (PP = 1; BS = 99). The clade represented by a widely distributed taxon from the *S. opacula* complex occurring in Bekob, Sohock, Iboti, and Djawara (CMJ576, CMJ574, CMJ213, CMJ533, CMJ575, CMJ1420) was also recovered with high support (PP = 1; BS = 99) and was sister to the *S. opacula* occurring in Kompina and Sole (CMJ1418, CMJ1421).

The *S. succursor* complex was divided into six well-supported clades, with each clade representing one potential species. In this group of species, the sister relationship between the Akom2 population (CMW76, CMW74, CMW75, CMW70, CMW69, CMW67, CMW72, CMW71, CMW68, CMW73) and the five remaining species was strongly supported by both BI and ML analyses (PP = 1; BS = 100). The tree shows that the *S. succursor* lineage of Akom2 is the basal clade of the *S. succursor* species complex. The second clade, as indicated by the Mbalmayo and Meyomessala lineages (CMJ751, CMW81, CMW82, CMW83, CMW84, CMW85, CMW86), was recovered in the ML analysis with weak support. However, it was strongly supported in the BI analysis (PP = 1; BS = 48). The present study sought to ascertain the genetic relationship between the Iboti lineage (CMJ211, CMJ212, CMJ214, CMJ215) in the third clade and the Somalomo (Dja Biosphere Reserve) lineage (CMJ1115, CMJ1117, CMJ1120, CMJ1121, CMJ1122) in the fourth clade. The BI and ML analyses (PP = 1; BS = 100) supported the abovementioned hypothesis. The fifth clade, comprising the Ongot lineage (CMJ150, CMJ151, CMJ152, CMJ387), is closely related to the clade represented by the Bangoulap, Evodoula, and Mfou populations, with a high degree of support (PP = 1; BS = 95).

However, the three lineages of *S. succursor* present in Bangoulap, Evodoula, and Mfou were neither recovered in ML nor in BI analyses, suggesting the possibility of a single species. As illustrated in Figure 1, the phylogenetic tree was constructed with full nomenclature for all newly described species, following a thorough examination and highlighting of their distinguishing morphological characteristics.

#### 3.1.2. Species Delineation by Automatic Barcode Gap Discovery (ABGD)

The genetic divergences of COI-5P sequences among the in-group lineages are shown in Figure 2.

The ABGD method, based on COI-5P sequences, displays a monomodal distribution of the individual pairwise distances (Figure 3A). A significant barcode gap was identified at a priori genetic distance thresholds ranging from 0.17 to 0.46 per cent, in the analysis of the COI-5P gene using the ABGD method. This provides substantial evidence in support of the presence of 19 distinct clades, constituting putative species (Figure 3C). The preliminary partitioning of the sequence data at each value of the prior interspecific divergence (P), demonstrated in Figure 3C, disclosed the presence of eight genetic clusters, corresponding to two recognized species and six putative species. A distinctive barcode gap was identified, delineating 10 candidate species, with varying numbers designated as the currently recognized species. This distinction was determined at a priori genetic distance thresholds of 0.77%. The ten species under consideration comprised two well-documented species and eight new taxa, the descriptions of which are provided in this study.

The recursive partition displays variations ranging from eight to 24 taxa, with prior maximal distance p = 1.00 × 10^−3^. Among the 24 taxa, 11 correspond to 11 species within the ingroup species delineated with the BI and ML analyses. The eight genetic clusters had a prior intraspecific genetic divergence value of 0.17%, in all recursive partitions. Therefore, we considered more likely than 10 species with intraspecific divergence values below 0.77% (Figure 3C).

In 16S rDNA, ABGD detected a bimodal distribution of the individual pairwise distances (Figure 4A).

The first peak comprises the within-cluster distances, and the second peak represents pairwise distances between the *Serpusia* and *Paraserpusia* clusters. A barcode gap ranging from 0.57% to 1.71% was revealed in the analysis of the 16S rDNA sequences using the ABGD method (Figure 4A). At each value of the prior interspecific divergence, the preliminary partitioning of the sequence data (P) indicated the presence of 12 genetic clusters. These correspond to two recognised species and 10 newly described species. The recursive partition displays variations ranging from 12 to 32 taxa, with prior maximal distance p = 1.67× 10^−3^, as demonstrated in Figure 4C. Among the 32 taxa, 11 correspond to 11 species within the ingroup species delineated with the BI and ML analyses. The 12 genetic clusters had a prior intraspecific genetic divergence value of 2.84% in all recursive partitions. Therefore, we considered that more likely than 24 species with intraspecific divergence values below 0.46% or as six species with intraspecific divergence values between 2.15% and 3.59% (Figure 4C).

#### 3.1.3. Intra- and Interspecific Genetic Distances

The distribution of COI-5P sequence divergence at each taxonomic level generated in BOLD is summarised in Table 3 below. The following seven morphologically identified species were included in the study: *S. opacula* (18 specimens), *S. succursor* (51 specimens), *A. subnuda* (four specimens), *A. morogorica* (three specimens), *A. nguruensis* (one specimen), *S. nitunila* (one specimen), and *P. carnapi* (three specimens). The mean intraspecific divergence for the five morphospecies represented by more than one specimen ranged from zero to 9.04%, with a high average of 3.57%. This high intraspecificity is attributable to the fact that some morphologically identified species were assigned to multiple BINs in BOLD. At the genus level, the mean interspecific divergence for identified morphospecies was 10.57% (ranging from 3.51 to 14.26%). At the family level, the mean interspecific divergence for identified morphospecies was 11.28% (ranging from 7.03.16 to 16.67%).

For each species, the mean and maximum intraspecific values were compared to the nearest neighbour distance (NN) (Table 4). *Segellia nitidula* and *A. nguruensis* were excluded from the barcode gap analysis as they were both represented by a single specimen. The genetic distance between the remaining morphologically identified species (*A. morogorica*, *A. subnuda*, *P. carnapi*, *S. opacula*, and *S. succursor*) was determined to be 7.9% using the Kimura 2 Parameter method. The minimum genetic divergence recorded was 3.62% (see Table 4 for further details). The maximum intraspecific divergence (9.71%) was observed in *S. succursor*. Except for *A. morogorica* and *S. succursor*, a barcode gap was evident in all other species (Table 4 and Figure 2).

#### 3.1.4. Morphospecies Split into More Than One BIN-Species and MOTU

BOLD generated 19 BINs for the entire dataset (Table 5), seven of which belong to *S. succursor*, five to *S. opacula* and the remaining seven to the outgroup species. The distance to the nearest neighbour (NN) was found to be less than 2% for three BINs (BOLD: AGO5270, BOLD: AGO5271 and BOLD: AGO5272), and greater than 2% for the remaining 16 BINs (Table 5). In the majority of cases, the distance of the BIN species to the NN was greater than the maximum intraspecific distance.

Furthermore, the Barcode Gap Analysis performed in BOLD yielded 19 MOTUs (Table 6), of which seven correspond to previously identified species (including outgroups) and 12 to putative species. The *S. succursor* lineages from Bangoulap, Evodoula, Mfou (*Paraserpusia tamessei* sp. nov. on the phylogenetic tree), and *A. subnuda* were each divided into two MOTUs, whereas *Paraserpusia husemanni* was divided into three. The remaining MOTUs corresponded to species as determined by phylogenetic analysis. Furthermore, two cases of COI-5P barcode sharing were identified: The *S. succursor* lineage from Ongot (CMJ150, CMJ151, CMJ152, CMJ387 (*Paraserpusia succursor* in the curated phylogenetic tree)) shares a single MOTU with *S. succursor* (CMW184, CMW185, CMW186, CMW187, CMW189, CMW190, CMW191, CMW19 (*Paraserpusia tamessei* sp. nov. in the curated phylogenetic tree)). The *S. succursor* lineage from Iboti (CMJ211, CMJ212, CMJ214, CMJ215 (*Paraserpusia höferi* sp. nov. in the curated phylogenetic tree)) is congruent with a single MOTU with the *S. succursor* lineage from Bangoulap (CMW170, CMW171, CMW172, CMW173, CMW174, CMW175, CMW176, CMW177, CMW178). The findings of this analysis contradict the results of the ABGD and phylogenetic analyses in cases of barcode sharing.

#### 3.1.5. Haplotype Network Analyses

The haplotype network, based on COI-5P gene sequences, is presented in Figure 5. The evidence indicates the presence of three primary clades. The *Serpusia* opacula species complex clade comprises one haplotype of *S. opacula* lineages from Kompina and Sole (two individuals), one of *S. opacula* lineages from Somalomo (two individuals), one of *S. opacula* lineages from Mouanko (three individuals), one of *S. opacula* lineages from Bissoue and Njuma (three individuals), and one of *S. opacula* lineages from Bekob, Djawara, Iboti, and Sohock (six individuals). These haplotypes correspond with the five species identified through phylogenetic analyses. This clade is distinguished from the nearest clade by a substantial genetic distance, characterised by more than 38 base pair substitutions. The *Aresceutica* clade is characterised by a branch indicating more than 10 bp substitutions. This clade is further subdivided into three subclades, namely *A. subnuda* (comprising four individuals), *A. nguruensis* (one individual), and *A. morogorica* (two individuals). The *S. succursor* species complex clade is distinguished by a pronounced elongation of the branch, indicative of a minimum of 28 base pair substitutions. This clade is more complex and is divided into six sub-clades, including the *S. succursor* lineage from Akom2 (six individuals), the *S. succursor* lineage from Somalomo (five individuals), the *S. succursor* lineage from Mbalmayo and Meyomessala (eight individuals), the *S. succursor* lineage from Ongot (four individuals), the *S. succursor* lineages from Bangoulap, Evodoula, and Mfou (five individuals) and the *S. succursor* lineage from Iboti (four individuals). Although the lineages of Bangoulap, Evodoula, and Mfou constitute one subclade, eleven other haplotypes deriving from these lineages are included: two in the Ongot lineage and nine in the Iboti lineage.

### 3.2. Taxonomic Treatment

#### 3.2.1. Differences Between *Serpusia* and *Paraserpusia* gen. nov.

The genera *Serpusia* and *Paraserpusia* gen. nov. can be reliably distinguished from one another in terms of their external morphology by several characteristics. In the genus *Serpusia*, the tegmina are found to be rudimentary, vestigial, and narrow, in contrast to the lobiform configuration observed in *Paraserpusia* gen. nov. The posterior margin of the pronotum exhibits a distinct incision and pronounced emargination at its midpoint, forming an acute angle in *Serpusia*. In contrast, the posterior margin of the pronotum in *Paraserpusia* gen. is rounded, exhibiting a noticeable wavy texture and a general pattern of fine mottling with light brown tones. The median carina of the pronotum is slightly tectiform and not depressed behind the posterior sulcus in *Serpusia*, while in *Paraserpusia* gen. nov., it is obtuse, fairly tectiform, and deeply depressed behind the posterior sulcus; the hind femora are black in their basal part in *Serpusia*, whereas they are dark red in their basal part in *Paraserpusia* gen. nov.

#### 3.2.2. The Genus Serpusia Karsch, 1891

With *S. opacula* as the type, species of the genus *Serpusia* are morphologically very similar.
*Serpusia opacula* Karsch, 1891Figure 6A–L

Material examined. Cameroon • 1 ♂, 1 ♀; Mouanko, in more or less stable forest; 3°38′0” N, 9°47′0” E, 89 m a.s.l.; 3 May. 2020; J.A. Yetchom Fondjo leg.; SMNK. Cameroon • 1 ♀, Mouanko, in the forest habitat; 3°38′0″ N, 9°47′0″ E, 89 m a.s.l.; 11 October 2020; J.A. Yetchom Fondjo leg.; SMNK.

Diagnosis. *Serpusia opacula* strongly resembles to *Paraserpusia succursor* in coloration, but is clearly distinct in the following characters: the elytra are rudimentary, vestigial and narrow, covering only a small portion of the tympanum and not reaching the posterior part of the metanotum (vs. elytra lobiform, with subparallel margins, slightly exceeding the posterior part of the metanotum in *P. succursor*); the hind femora are black in the basal part (vs. dark red in *P. succursor*); the posterior margin of the pronotum is strongly emarginate and acute (vs. rounded or only slightly emarginate in *P. succursor*); the male genitalia differ by their apodemes of cingulum with almost straight apex, and strongly exceeding the endophallic apodemes’ tip (vs. the apodemes of cingulum exhibit a slight curvature ventrally at the apex, with the tip of the endophallic apodemes being reached in *P. succursor*); the rami is of a length that is equal to or slightly exceeds its width in lateral view (vs. in *P. succursor*, the rami is longer than its width in lateral view); the ectophallic sheath is long (vs. in *P. succursor*, the ectophallic sheath is short); the aedeagus is long, only slightly curved upwards, and has almost straight margins and parallel tips (vs. the aedeagus is of medium size, with roughly divergent margins and broad tip in *P. succursor*); the gonopore sac is small in size and subtriangular in shape, covering approximately one-half of the endophallus (vs. the gonopore sac of *P. succursor* is of a larger size and covers approximately two-thirds of the endophallus).

*Serpusia opacula* is very similar to *S. missoupi* sp. nov. from which it can easily be distinguished by the presence of a large shiny black spot at the apex of the elytra, a large spot resulting from the fusion between the two apical spots (vs. two shiny black spots well separated in *S. missoupi* sp. nov.); furthermore, the pale spot located in the inferior-external region of the lateral lobes of the pronotum is evident and discernible (vs. pale spot inconspicuous or absent in *S. missoupi* sp. nov.); the female genitalia are distinguished by the presence of a subgenital plate with long anterior apodemes (vs. broad and short in *S. missoupi* sp. nov.), and a short spermatheca duct (vs. very long in *S. missoupi* sp. nov.).

Redescription. Male. The abdomen is characterised by a shiny, sub-unicoloured appearance and a subtle olivaceous tint. The head is erect, and the fastigium of the vertex is flattened. The lateral lobes of the pronotum exhibit an irregular black spot that is more or less clearly defined. The median carina of the pronotum is visible and is crossed by three transverse grooves. The prozona is approximately three times longer than the metazona. The pronotum, mesonotum, metanotum, and dorsal surface of the abdomen exhibit pronounced wrinkling and granularity. The elytra are rudimentary and vestigial, exhibiting a very narrow configuration.

They are adorned with two shiny black subapical spots, which are sometimes confluent, and extend to the posterior edge of the metanotum in some individuals. The inner surface of the hind femur is red, the black plume is located in the apical half, the basal third is black, and the dorsal carina is serrated. The hind tibiae and tarsi are characterised by the presence of white hairs and setae. The supra-anal plate of the male exhibits a broad basal half, longitudinal grooves in the middle, and furrowed edges. The cerci of the male are short and conical in shape.

Epiphallus (Figure 6H). The bridge is narrow, arched, and forms an obtuse angle; the ancorae are small, pointed inwards; the lophi are broad, wide, lobiform; the oval sclerites are small, roughly elongate, and strap-like; and the anterior projections are strongly reduced.

Phallic complex (Figure 6I–K). It is characterised by an elongated, less robust form. The dorsal arch of the cingulum is U-shaped and open, with the anterior margin situated beneath the zygoma that is broadly emarginate in the midline. The apodemes of cingulum are more or less straight, running approximately parallel to each other distally and strongly exceeding the distal tip of the endophallic apodemes. The endophallus is characterised by its short, thread-like flexure. The endophallic valves are distinguished by their length and slenderness, exhibiting a wide and dorso-ventrally flattened base that extends almost to the tip of the ectophallic aedeagal sheath. The endophallic valves are deeply cupped and laterally directed, while the zygoma is notable for its pronounced sclerotisation. The rami are well-developed and sclerotised, as long as or slightly longer than their width in lateral view, extending to the ventral midline of the phallus. The ventral extremities are not fused nor overlapping. The ectophallic sheath is long. The aedeagus is projecting, large in size, long, only slightly curved upwards, and with almost straight margins and parallel tips. The aedeagal valves are slender and pointed, and are enveloped or covered by a thick, semitransparent sheath that is closely appressed to the endophallic valves. The gonopore sac is small, subtriangular, and approximately half of the endophallus is covered by it, as well as the endophallic sclerites ventrally.

Female (Figure 6B,C,E,G,L). As the male, but larger, the valves of the ovipositor are small and short. The subgenital plate (Figure 6C) is pentagonal in shape, diminutive in size, with truncated posterior margins; anterior apodemes of considerable length; egg-guide narrow and short; ventral pockets of the vaginal floor of significant size; copulatory bursa large in size, almost straight, gradually narrowing towards the front; each basivalvar sclerite exhibits minimal curvature, forming an obtuse angle. The spermatheca duct (Figure 6L) is of a reduced length, and the base of the spermathecal duct opens directly at the apex of the copulatory bursa. The recurrent distal trunk of the lateral spermathecal diverticulum is five times longer than that of the proximal trunk.

Distribution. Barombi Station, Buea, Kumba, Mouanko, Tombel, Cameroon; Nigeria.

Biology and Ecology. *Serpusia opacula* is present throughout the year in Cameroon’s humid forests. The species is presently confined to the regions of Cameroon that experience an equatorial climate, exhibiting a unimodal distribution of rainfall.
*Serpusia kennei* Yetchom & Wandji sp. nov.


https://zoobank.org:act:49E3000F-8A67-4FF6-8BF7-9E19E849E178
Figure 7A–L


Material examined. *Holotype*. Cameroon • ♂; Bissoue, Ebo Forest, more or less stable forest habitat; 4°21′42″ N, 10°12′31″ E; 17 April 2022; J.A. Yetchom Fondjo leg.; SMNK, SMNK-ORTH-0000007. *Paratypes*. Cameroon • 1 ♀; Bissoue, Ebo Forest, more or less stable forest habitat; 4°21′42″ N, 10°12′31″ E; 17 April 2022; J.A. Yetchom Fondjo leg.; SMNK. Cameroon • 1 ♂, 3 ♀♀; Njuma, Ebo Forest, more or less stable forest habitat; 4°20′53″ N, 10°13′56″ E; 18 April 2022; J.A. Yetchom Fondjo leg.; SMNK.

Diagnosis. *Serpusia kennei* sp. nov. is similar to *S. opacula* (Figure 6), but can be distinguished by the shorter elytra, which do not fully cover the tympanum and do not reach the posterior edge of the metanotum (vs. longer elytra, extending beyond the posterior edge of the metanotum and fully covering the tympanum in *S. opacula*), and by the medium-sized hind legs (vs. large in *S. opacula*). The following characteristics differentiate male genitalia: the apodemes of cingulum exhibit a slight curvature inwards at the apex, not reaching the tip of the endophallic apodemes (vs. having an almost straight apex and extending well beyond the tip of the endophallic apodemes in *S. opacula*). The valves of cingulum are short (vs. long in *S. opacula*), and the aedeagus is short (vs. large or elongate in *S. opacula*), while the gonopore sac is large (vs. small in *S. opacula*). The female genitalia are distinguished by the presence of a subgenital plate, which is characterised by broader anterior apodemes (vs. long in *S. opacula*). Additionally, the spermatheca duct is notably elongated (vs. short in *S. opacula*).

This species can be distinguished from *S. seinoi* sp. nov. by the shorter elytra, which do not reach the posterior margin of the tympanum (vs. longer, extending beyond the posterior margin of the tympanum in *S. seinoi*), and by the shiny black spots at the apex of the elytra, which are well separated and sometimes confluent in females (vs. fused into a single large shiny black spot in *S. seinoi*). The female genitalia are distinguished by their pentagonal subgenital plate, with truncate posterior margins (vs. hexagonal with rounded posterior margins in *S. seinoi*); each basivalvar sclerite exhibits minimal curvature, forming an obtuse angle (vs. forming an acute angle in *S. seinoi*).

Description. Male. The general colouration is dark and opaque. The head is straight, and the fastigium of the vertex is flattened. The lateral lobes of the pronotum have a more or less clearly defined, irregular black spot. The median carina of the pronotum is perceptible and is crossed by three transverse grooves. The prozone is about three times longer than the metazona. The pronotum, mesonotum, metanotum and dorsal surface of the abdomen are strongly wrinkled and granular. The elytra are rudimentary and vestigial, and are extremely narrow and short, not reaching the posterior edge of the tympanum or the posterior edge of the metanotum. They possess two shiny black subapical spots, which are sometimes confluent in females. The hind femora are characterised by an olivaceous outer surface, a red inner surface, and a black plume in the apical half, with the basal third being black and the dorsal carina serrated. The apical half of the hind tibiae and tarsi are red, while the basal half exhibits a black plume. The hind tibiae and tarsi are adorned with white hairs and bristles.

Epiphallus (Figure 7H). Bridge short, narrow, strongly arched, and forming an obtuse angle; ancorae small, pointed inwards towards the mid-line; lophi broad, wide, lobiform; oval sclerites small, roughly elongate, and strap-like; anterior projections strongly reduced.

Phallic complex (Figure 7I–K). It is characterised by an elongated and slightly robust form. The dorsal arc of the cingular is U-shaped and closed, with the anterior margin situated beneath the zygoma and exhibiting a broad emargination along the midline. The apodemes of cingulum are notably long and predominantly straight, extending approximately parallel to each other in the distal region. A slight curvature is observed at the apex, and the apodemes do not extend to the distal tip of the endophallic apodemes. The endophallic apodemes are deeply cupped, laterally directed, and possess two channels with a U-shaped cross-section that extend ventrally and posteriorly to form the gonopore. The endophallus exhibits a thread-like flexure that is visible in the lateral view. The endophallic valves are characterised by their length, slenderness, width, dorso-ventral flattening at the base, lateral compression, and narrowing at the apex.

These valves extend almost to the tip of the ectophallic aedeagal sheath. The zygoma is sclerotised; the rami are well-developed and sclerotised. In the lateral view, the rami are longer than they are wide and do not extend to the ventral midline of the phallus. They are not fused nor overlapping, and their ventral extremities are linked by a transparent membrane. The ectophallic sheath is short, and the cingular valves are short and pointed. The aedeagus is characterised by its short, curved shape, with margins converging towards an obtuse tip. It is enveloped by a thick, semitransparent sheath, closely appressed to the endophallic valves. The gonopore sac is large, approximately covering two-thirds of the endophallus, and well covering the endophallic sclerites ventrally.

Female. Similar to the male, but larger; the valves of the ovipositor are short; the subgenital plate (Figure 7C) is pentagonal in shape, small in size, with truncated posterior margins; anterior apodemes broad; egg-guide narrow, short; ventral pockets of the vaginal floor large in size; copulatory bursa almost straight, gradually narrowing towards the front; bottom of the copulatory bursa close to the arc of the basivalvar sclerites. The basivalvar sclerite exhibits minimal curvature, forming an obtuse angle. The spermatheca duct (Figure 7L) is characterised by its considerable length, and the base of the spermathecal duct opens directly at the apex of the copulatory bursa. Notably, the recurrent distal trunk of the lateral spermathecal diverticulum is 15× longer than the proximal trunk.

Etymology. The species was named after Professor Martin Kenne, an important and recognized entomologist in Cameroon, for his dedication and scientific contribution to insect biodiversity.

Distribution. Bissoue, Njuma (Littoral region of Cameroon).

Biology and Ecology. Adults of this species can be observed throughout the year in more or less stable forests with dark undergrowth.
*Serpusia missoupi* Yetchom & Wandji sp. nov.


https://zoobank.org:act:CD715702-37F0-47F9-B923-1C79203763F7
Figure 8A–L


Material examined. *Holotype*. Cameroon • ♂; Kompina, less disturbed forests; 4°21′56″ N, 9°35′58″ E; 28 November 2018; J.A. Yetchom Fondjo leg.; SMNK, SMNK-ORTH-0000008. *Paratypes*. Cameroon • 1 ♀; Kompina, disturbed forests; 4°21′56″ N, 9°35′58″ E; 28 November 2018; J.A. Yetchom Fondjo leg.; SMNK. Cameroon • 1 ♀; Kompina, disturbed forests; 4°21′56″ N, 9°35′58″ E; 16 September 2018; J.A. Yetchom Fondjo leg.; SMNK. Cameroon • 1 ♀; Sole, disturbed forests; 4°35′0″ N, 9°48′0″ E; 28 February 2017; J.A. Yetchom Fondjo leg.; SMNK. Cameroon • 1 ♀; Sohock, disturbed forests; 5°42′11″ N, 10°32′34″ E; 3 April 2017; J.A. Yetchom Fondjo leg.; SMNK.

Diagnosis. *Serpusia missoupi* sp. nov. is similar to *S. verhaaghi* sp. nov. from which it differs by the following characters: pale spot on the lower outer edge of the lateral lobes of the pronotum barely visible to absent (vs. present and visible in *S. verhaaghi*); elytra with rounded apex (vs. truncate in *S. verhaaghi*). The male genitalia differ by their apodemes of cingulum, which are not parallel to each other, are strongly curved, sickle-like in over two-thirds of its apical part (vs. predominantly straight, extending approximately parallel to each other in the distal part, with the apical part sometimes strongly incurved, hook-like in *S. verhaaghi* sp. nov.); the aedeagus is only slightly curved upwards, and with almost straight margins and obtuse tips (vs. strongly curved upwards, with roughly flexuous margins, and broad and divergent tips in *S. verhaaghi* sp. nov.). The female genitalia differ only by their broad anterior apodemes (vs. narrow in *S. verhaaghi*).

The new species is also somewhat similar to *S. opacula* (Figure 6) in general colouration, from which it differs in the following characters: a medium body size (vs. slender, larger in *S. opacula*); the presence of a pale spot on the lower lateral lobes of the pronotum is evident (vs. present but difficult to distinguish in *S. opacula*); the elytra do not extend to the posterior edge of the metanotum, instead partially enshrouding the tympanum (vs. in *S. opacula*, the elytra extend beyond the posterior edge of the metanotum, fully enclosing the tympanum). The male phalli are distinguished by their apodemes of cingulum which are not parallel to each other, are strongly curved, sickle-like in over two-thirds of its apical part (vs. they are more or less straight, running approximately parallel to each other distally in *S. opacula*); the aedeagus is short, with obtuse tips (vs. the aedeagus is large, long, with parallel tips in *S. opacula*). The female genitalia are distinguished by their broad and short anterior apodemes (vs. which are very long in *S. opacula*), and the spermatheca duct is notably long (vs. short in *S. opacula*).

Description. Male. Of medium size. The head is straight in lateral view, the front is flattened, and the filiform antennae are longer than the head and pronotum combined. The general colouration is uniformly dark brown, the integument is opaque, and the pronotum, mesonotum, metanotum, and dorsal surface of the abdomen are strongly wrinkled and granular. The lateral lobes of the pronotum are adorned with a large, irregular, shiny black spot, and the pale spot on the lower outer edge of the lateral lobes of the pronotum is inconspicuous to absent. The elytra are rudimentary, vestigial, and extremely narrow with a rounded apex. They are adorned with two well-separated subapical shiny black spots, extending to the posterior edge of the metanotum. The elytra partially cover the tympanum and have a rounded apex. The outer surface of the hind femur is predominantly black, particularly dark on the lower outer zone, while the inner surface of the same structure exhibits a red hue, accompanied by a prominent dark black spot in the apical half. The basal third of the hind femur is dark black, and numerous small teeth mark its dorsal carina. Finally, the basal half of the hind tibia is black-purple.

Epiphallus (Figure 8H). Bridge narrow, arched; ancorae small, slanted inwards; lophi broad, wide, lobiform; oval sclerites small, roughly elongated, and subtriangular; anterior projections strongly reduced.

Phallic complex (Figure 8I–K). The phallic complex is characterised by an elongated structure. The dorsal arc of the cingulum is U-shaped, closed, and its anterior margin is situated beneath the zygoma, exhibiting a broad emargination along the midline. The apodemes of cingulum are of considerable length, not parallel to each other, strongly curved, sickle-like in over two-thirds of its apical part (Figure 8I–K), and strongly exceeding the tip of the endophallic apodemes. The endophallic apodemes are characterised by their deeply cupped structure, and from their ventral margins, two channels with a U-shaped cross-section extend ventrally and posteriorly, thereby forming the gonopore processes. The endophallus exhibits a short, thread-like flexure. The endophallic valves are long and slender, wide and dorsoventrally flattened basally, but laterally compressed and narrow apically, and extend almost to the tip of the ectophallic aedeagal sheath. The zygoma is sclerotised, and the rami are well developed and sclerotised. In the lateral view, the rami are longer than they are wide, and they extend to the ventral midline of the phallus. The rami are not fused, nor overlapping, and their ventral extremities are linked by a transparent membrane. The ectophallic sheath is short and sometimes of medium size (Figure 8J). The valves of cingulum are short and pointed. The aedeagus is short and dorsally directed, only slightly curved upwards, and with almost straight margins and obtuse tips. The aedeagal valves are enveloped or covered by a thick, semitransparent sheath, closely appressed to the endophallic valves. The gonopore sac is small.

Female. Similar to male but larger; cerci short, conical; supra-anal plate subtriangular, divided at the base and with a transverse rectilinear groove; valves of ovipositor short, 2.5× longer than wide in coalescence position.

The subgenital plate (Figure 8C) is pentagonal, with truncated posterior margins; anterior apodemes broad and short; egg-guide thin and short; ventral pockets of the vaginal chamber large; copulatory bursa almost straight, gradually narrowing towards the front. The copulatory bursa is located at the base, near the arc of the basivalvar sclerites. Each basivalvar sclerite exhibits a slight curvature, forming an obtuse angle. The spermatheca duct (Figure 8L) is characterised as simple and notably elongated, with the base of the spermathecal duct opening directly at the apex of the bursa.

Etymology. The species is named after Prof. Alain Didier Missoup in recognition of his work and achievements in the systematic and evolutionary biology of small mammals in Cameroon.

Distribution. Kompina, Sole (Littoral region of Cameroon) (Figure 17).

Biology and Ecology. The species is present throughout the year in disturbed and undisturbed forests in the Littoral evergreen forests of Cameroon.
*Serpusia seinoi* Yetchom & Wandji sp. nov.


https://zoobank.org:act:AF1502F8-4269-4346-A0D4-93AAC3C49D9A
Figure 9A–F


Material examined. *Holotype*. Cameroon • ♀; Somalomo, Dja Biosphere Reserve, forest habitat; 3°22′55″ N, 12°44′30″ E; 10 April 2022; J.A. Yetchom Fondjo leg.; SMNK-ORTH-0000009; SMNK. *Paratypes*. Cameroon • 1 ♀; Somalomo, Dja Biosphere Reserve, forest habitats; 3°22′55″ N, 12°44′30″ E; 10 April 2022; J.A. Yetchom Fondjo leg.; SMNK.

Diagnosis. *Serpusia seinoi* sp. nov. is similar to *S. opacula* (Figure 6) in its general colouration. Still, it can be distinguished from *S. opacula* specimens by its shorter tegmina, which do not reach the posterior margin of the metanotum (vs. extending well beyond in *S. opacula*). Female genitalia differ by the following characteristics: subgenital plate hexagonal, with rounded posterior margins (vs. pentagonal, with truncated posterior margins in *S. opacula*); anterior apodemes broad (vs. long in *S. opacula*); each basivalvar sclerite barely curved, forming an acute angle (vs. forming an obtuse angle in *S. opacula*); spermatheca duct long (vs. short in *S. opacula*)

Description. Female. The general colouration is dark and opaque. The head is erect, and the fastigium of the vertex is flattened. The lateral lobes of the pronotum have a more or less clearly defined irregular black spot. The median carina of the pronotum is perceptible and crossed by three transverse grooves. The prozona is about three times longer than the metazona. The pronotum, mesonotum, metanotum, and dorsal surface of the abdomen are strongly wrinkled and granular. The elytra are rudimentary and vestigial, and are very narrow, with two shiny black subapical spots, which are sometimes confluent and do not reach the posterior margin of the metanotum. The hind femora have an olivaceous outer surface, a red inner surface, a black plume in the apical half, a black basal third, and a serrated dorsal carina. The apical half of the hind tibiae and tarsi are red, while the basal half has a black plume. The hind tibiae and tarsi are hairy and bristly.

Female genitalia. The valves of the ovipositor are short, and the subgenital plate (Figure 9D) is hexagonal, small, and possesses rounded posterior margins. The anterior apodemes are broad, the egg-guide narrow and acute, the ventral pockets of the vaginal chamber are large, the copulatory bursa is straight and short, and the bottom of the copulatory bursa is close to the arc of the basivalvar sclerites. The basivalvar sclerite exhibits minimal curvature, forming an acute angle. The spermatheca duct (Figure 9E) is characterised by its considerable length, and the base of the spermathecal duct opens directly at the apex of the copulatory bursa. Notably, the recurrent distal trunk of the lateral spermathecal diverticulum is 17× longer than the proximal trunk.

**Figure 9 insects-16-01020-f009:**
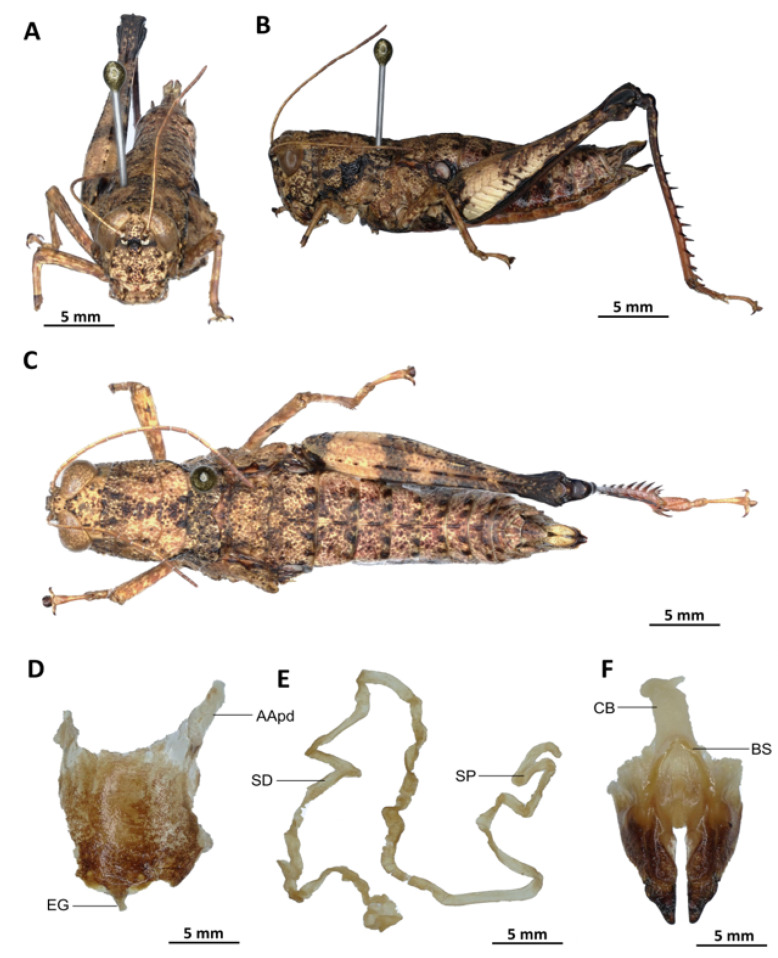
*Serpusia seinoi* sp. nov. (**A**) female frontal view; (**B**) female lateral view; (**C**) female dorsal view; (**D**) female subgenital plate; (**E**) female spermatheca; (**F**) female ovipositor. BS: basivalvare sclerites; CB: corpulatory bursa; EG: egg-guide; SD: spermathecal duct; SP: spermathecal.

Etymology. The species was dedicated to Professor Richard Seino Akwa-Njoh in recognition of his valuable contribution to the advancement of research in Cytogenetics, Biodiversity, and Evolutionary Biology of Orthoptera grasshoppers in Cameroon.

Distribution. Somalomo, Dja Biosphere Reserve (Eastern Cameroon) (Figure 17).

Biology and Ecology. Adults of this species occur throughout the year in the disturbed forests of Eastern Cameroon.
*Serpusia verhaaghi* Yetchom & Wandji sp. nov.


https://zoobank.org:act:6F5E21F1-13AF-45D1-B455-0FCA2F71EDF2
Figure 10A–M


Material examined. *Holotype*. Cameroon • ♂; Iboti, Ebo Forest, forest habitats; 04°27,999′ N, 010°27,357′ E, 742 m a.s.l.; 7 January 2022; J.A. Yetchom Fondjo leg.; SMNK, SMNK-ORTH-0000010. *Paratypes*. Cameroon • 3 ♂♂, 1 ♀; Iboti, Ebo Forest, forest habitats; 04°27,999′ N, 010°27,357′ E, 742 m a.s.l.; 7 January 2022; J.A. Yetchom Fondjo leg.; SMNK. Cameroon • 1 ♂, 1 ♀; Bekob, Ebo Forest, forest habitats; 4°21′9″ N, 10°25′16″ E; 02 Mar. 2021; J.A. Yetchom Fondjo leg.; SMNK. Cameroon • 1 ♀; Djawara, more or less stable forest; 4°27′15″ N, 10°35′58″ E; 27 March 2017; J.A. Yetchom Fondjo leg.; SMNK.

Diagnosis. The new species exhibits external morphological characteristics similar to *S. opacula* (Figure 6). The distinction between the two species is determined by the presence of shiny black spots on the elytra, which are invariably fused (vs. though sometimes confluent in *S. opacula*), the vestigial nature of the elytra with truncate apices (vs. rounded in *S. opacula*), and the elytra not reaching the posterior margin of the metanotum (vs. extending beyond the posterior margin of the metanotum in *S. opacula*). The male genitalia are characterised by the presence of a dorsal arch of cingulum closed (vs. open in *S. opacula*). In addition, the apodemes of the cingulum are occasionally observed to possess strongly curved hook-like apices (vs. with almost straight apices in *S. opacula*). The ectophallic sheath is characterised by its short or medium size, which is long in *S. opacula*. The aedeagus is short, with roughly flexuous margins and broadened and divergent tips (vs. in *S. opacula*, the aedeagus is long, more or less straight, with roughly parallel margins and tips). The female genitalia are distinguished by the presence of a subgenital plate, characterised by narrow and short anterior apodemes (vs. long in *S. opacula*). The spermatheca duct is notably elongated, a feature that is absent in *S. opacula*.

Description. Male. The species is characterised by a slight olivaceous, subunicoloured, opaque colouration and a shiny abdomen. The head is erect, and the fastigium of the vertex is flattened. The lateral lobes of the pronotum are ornamented with a more or less clearly defined irregular black spot, and the median carina of the pronotum is perceptible and crossed by three transverse furrows. The prozona is approximately three times as long as the metazona. The pronotum, mesonotum, metanotum, and dorsal surface of the abdomen are strongly wrinkled and granular. The elytra are rudimentary, vestigial, and very narrow, clearly reaching the posterior edge of the metanotum. The two shiny black spots are always fused. The inner face of the posterior femora is red, black in its apical half, and black in the basal third. The dorsal carina has teeth. The apical half of the hind tibiae and tarsi are red, while the basal half is plumed black. The hind tibiae and tarsi are characterised by the presence of white hairs and bristles. The male supra-anal plate is distinguished by a broad basal half, longitudinally grooved in the middle, and furrowed at the edges. The cerci of the male is short and conical.

Epiphallus (Figure 10I). Bridge narrow, strongly arched; ancorae small, slanted inwards; lophi broad, wide, lobiform; oval sclerites small, roughly elongated, and strap-like; anterior projections strongly reduced.

Phallic complex (Figure 10J–M). The phallic complex is characterised by an elongated and slightly robust structure. The dorsal arc of the cingulum is U-shaped, more or less closed, and its anterior margin is situated beneath the zygoma, exhibiting a broad emargination along the midline. The apodemes of cingulum are of considerable length and predominantly straight, extending approximately parallel to each other in the distal part, with the apical part sometimes strongly incurved, hook-like (Figure 10K,L), and strongly exceeding the tip of the endophallic apodemes.

The endophallic apodemes are characterised by their deeply cupped structure, and from their ventral margins, two channels with a U-shaped cross-section extend ventrally and posteriorly, thereby forming the gonopore processes. The endophallus exhibits a short, thread-like flexure that is visible in the lateral view. The endophallic valves are long and slender, wide, and dorsoventrally flattened at the base, but laterally compressed and narrow at the apex, and extend almost to the tip of the ectophallic aedeagal sheath. The zygoma is sclerotised, and the rami are well developed and sclerotised. In the lateral view, the rami are longer than they are wide, and they extend to the ventral midline of the phallus. The rami are not fused, nor overlapping, and their ventral extremities are linked by a transparent membrane. The ectophallic sheath is short and sometimes of medium size (Figure 10L). The valves of cingulum are short and pointed, and sometimes they are of medium size and broad. The aedeagus is characterised by its short and dorsally directed projection, its roughly flexuous margins, and its broad and divergent tips. The aedeagal valves are enveloped or covered by a thick, semitransparent sheath, closely appressed to the endophallic valves. The gonopore sac is large, roughly covering two-thirds of the endophallus and completely covering the endophallic sclerites ventrally, and situated more anteriorly.

Female. Similar to male, but larger; the valves of the ovipositor are short; the subgenital plate (Figure 10C) pentagonal, small, with truncated posterior margins; anterior apodemes narrow and reduced in length; egg-guide filiform and short; ventral pockets of the vaginal floor reduced; copulatory bursa almost straight, gradually narrowing towards the anterior end; bottom of the copulatory bursa close to the arc of the basivalvar sclerites. The basivalvar sclerite exhibits minimal curvature, forming an obtuse angle. The spermatheca duct (Figure 10D) is characterised by its considerable length, and the base of the spermathecal duct opens directly at the apex of the copulatory bursa. Notably, the recurrent distal trunk of the lateral spermathecal diverticulum is 15× longer than the proximal trunk.

Etymology. The species is named after Dr. Manfred Verhaagh, in recognition of his contribution to the biodiversity and species discovery of Tropical ant fauna.

Distribution. Iboti, Bekob, Sohock.

Biology and Ecology. This species occurs throughout the year in more or less disturbed forests in the coastal zones of Cameroon.

Key to species of the genus *Serpusia*

One of the five species of the genus *Serpusia* was described based on the female morphology. Here, we present two keys, one based on external morphology (including all five species) and one on the male internal phallic structures (including four species).

Based on the external morphology


Elytra rudimentary, vestigial, narrow, with truncate apices (Figure 7D,E) 
*Serpusia kennei* Yetchom & Wandji sp. nov.
-Elytra rudimentary, vestigial, narrow, with rounded apices (Figure 6, Figure 7, Figure 8, Figure 9 and Figure 10)
2
2.Elytra extending beyond the posterior margin of the metanotum (Figure 6D,E)
*Serpusia opacula* Karsch, 1891
-Elytra not reaching the posterior margin of the metanotum
3
3.The two apical shiny black spots at the apex of the elytra are always separated, female subgenital plate hexagonal (Figure 9D)
*Serpusia seinoi* Yetchom & Wandji sp. nov.
-The two apical shiny black spots at the apex of the elytra are confluent or fused
4
4.The two apical shiny black spots at the apex of the elytra are about the same size (Figure 10E,F)
*Serpusia verhaaghi* Yetchom & Wandji sp. nov.
-The internal spot is smaller (Figure 8D,E)
*Serpusia missoupi* Yetchom & Wandji sp. nov.

B.Based on the anatomy of the phallus


Dorsal arch of cingular closed
2
-Dorsal arch of cingulum open, apodemes of cingular long, with more or less straight apices, and strongly exceeding the tip of the endophallic apodemes, ectophallic sheath and valves of cingulum always long, aedeagus long and slender, with almost straight and parallel margins and tip (Figure 6I,J,K)
*Serpusia opacula* Karsch, 1891
2.Apodemes of cingulum slightly curved inwards at its apex, and not reaching the tip of the endophallic apodemes, ectophallic sheath and valves of cingulum of medium size, aedeagus short with oblique margins and converging tip (Figure 7I–K)
*Serpusia kennei* Yetchom & Wandji sp. nov.
-Apodemes of cingular sometimes strongly curved, hook-like, or sickle-like, and strongly exceeding the tip of the endophallic apodemes
3
3.Aedeagus with almost straight margins and obtuse tips (Figure 8J–K)
*Serpusia missoupi* Yetchom & Wandji sp. nov.
-Aedeagus with roughly flexuous margins, broad and divergent tips (Figure 10J–M)
*Serpusia verhaaghi* Yetchom & Wandji sp. nov.

#### 3.2.3. Other Species

Here, we redescribe *S. catamita* based on examined collection materials, whereas the original description of *S. blanchardi* and *S. inflata* by [29,30], respectively, is repeated and transcribed, as we lack specimens of these species. *Serpusia blanchardi*, *S. catamita*, and *S. inflata* should be removed from *Serpusia* and included in a separate genus. Molecular data are needed to provide insight into the systematic position of these three species.
*Serpusia catamita* Karsch, 1893Figure 11A–K

Material examined. Togo • 1 ♂, 4 ♀♀; Bismarckburg; 20 September–31 October 1990; R. Büttner S. leg.; SMNK Stuttgart; Togo • 1 ♀; West Africa; 1984; Akem Mohr leg.; SMNK Stuttgart; Togo • 1 ♀; West Africa; 1980; Akem Mohr leg.; SMNK Stuttgart.

Diagnosis. Similar to *S. opacula* in shape and colouration. The elytra are broad, with an anterior or outer margin that is initially straight and then rounded (vs. in *S. opacula,* they are narrower and truncate at the anterior margin). The supra-anal plate has a longitudinal groove at the front with raised edges, in the middle of a strong and massive transverse carina, flat behind and, after a slight inflection of the lateral edge, on each side, quite flat in the middle and pointed towards the end (vs. in *S. opacula*, behind the strong transverse carina that closes the basal half, two strong longitudinal carinae converge towards the rear, giving the impression that the posterior half is suddenly very narrow, when in fact it is not).

*Serpusia catamita* is very similar and close to *S. inflata* (Figure 12A,B,D–G), but with slightly protruding fastigium, distinctly wavy and generally finely mottled with light brown and black posterior margin of pronotum (vs. *S. inflata* has a flatter and wider fastigium, a more swollen prozona of pronotum, a more or less smooth and flat posterior edge of the pronotum); in *S. catamita*, the most pronounced bulge on the outer margin of the elytra, where the black spot is also located, is more central than apical (vs. in *S. inflata*, the elytra are rounded and broader).

Redescription. Male. The fastigium is slightly protruding forward; the antennae are one-fourth longer than the head and pronotum combined. The prozona of the pronotum is rather strongly convex, giving the metazona a relatively sunken appearance. The posterior margin of the pronotum is distinctly wavy and generally finely mottled with light brown and black. The prosternal tubercle is conical and pointed; the elytra are broad, with an anterior or outer margin that is initially straight and then rounded, protruding beyond the posterior margin of metanotum; hind tibiae light red; the supra-anal plate is broadly cordate, with acute apex; the cerci are slightly longer than the supra-anal plate, pointed; the supra-anal plate has a longitudinal groove at the front with raised edges, in the middle of a strong and massive transverse carina, flat behind and, after a slight inflection of the lateral edge, on each side, quite flat in the middle and pointed towards the end.

Male genitalia (Figure 11H–K). The description of the phallus of *S. catamita* is presented for the first time in this work. Its internal genitalia are similar to those of *Serpusia* and *Aresceutica*, but differing in some details: (i) the ancorae are slanted forwards, rather than inwards as in *Serpusia*; (ii) the lophi are curved upwards, with angular apices; (iii) the phallic complex is strongly sclerotized compared to that of *Serpusia*; (iv) the endophallic apodemes are more deeply cupped as in *Serpusia*, but the ejaculatory sac, which is bounded by the gonopore processes, is strongly reduced and is situated more posteriorly in position under the basal endophallus as in *Aresceutica*, rather than more anterioly as in *Serpusia*; (v) the flexure of the endophallus is short, thread-like, as in *Serpusia*; (vi) the apodemes of cingulum of *S. catamita* are long and more or less straight, running roughly parallel to each other distally as in *Serpusia*. The dorsal arc of the cingulum is U-shaped, as in *Serpusia*; (vii) the anterior margin of the cingulum arch under the zygoma is broadly emarginate in the midline as in *Serpusia* and *Paraserpusia* gen. nov.; (viii) the valves of cingulum are slender and pointed as in *Serpusia* and *Paraserpusia*; (ix) the endophallic valves are long and slender, dorsoventrally flattened basally as in *Serpusia*, but they are narrow basally than in the latter; (x) a complex ectophallic aedeagal sheath envelops the aedeagal valves and is closely appressed to the endophallic valves, to which it is attached.

Distribution. Centre, Bismarcksburg, Togo; Ghana.

Biology. Unknown.
*Serpusia inflata* Ramme, 1929Figure 12A,B,D–G

Description [30]. Male. The fastigium is flat, with a steep slope, transversal to the anterior, without a transversal stripe, and merging directly into the frontal stripe. The frontal bar is not deepened above the bases of the antennae, roughly punctate, then slightly deepened, with gradually converging carinae, not widened at the level of the lateral eye, but slightly widened below the lateral eye, irregular in structure, with very irregular lateral carinae, descending towards the clypeus. The antennae are one-fourth longer than the head and pronotum combined. The prozona of the pronotum is rather strongly convex, giving the metazona a relatively sunken appearance. The posterior margin of the pronotum is protruding in the shape of a blunt wing, with an almost smooth, barely wavy ridge. The prosternal tubercle is conical and pointed. The elytra are round-oval, of unequal length, from the left to the middle, from the right to the end of the first abdominal segment. The supra-anal plate is broadly cordate (heart-shaped), with a small, almost lobed bar at the base of each side; the cerci are slightly longer than the supra-anal plate, pointed in the shape of a strap; the subgenital plate is blunt, conical.

Colouration. Medium brown, head finely mottled with light spots, and the dorsal surface of prozona of the pronotum is slightly lighter in places; the lateral lobes near the upper margin (below the protuberance of the prozona) are washed, shiny, blackish, the posterior angles slightly lighter; the elytra are with a shiny black spot on the lower margin (seen at rest), about half the length of the elytra; the inner margin of hind legs is dirty red, the hind tibiae and basal tarsi are light red.

Female. Head similar to that of the male but larger; the antennae are slightly longer than head and pronotum combined; the protuberance of pronotum is swollen, especially on the lateral margins; the posterior margin is somewhat less distinct than in male, but still clearly broad, with a blunt, protruding angle and a slight depression in the middle; the elytra are broadly oval, with a slightly pointed apex at an obtuse angle, protruding somewhat beyond the first abdominal segment.

Colouration. Uniformly dark brown; elytra with a correspondingly much smaller variegated/shiny black spot.

Distribution. Kayima, Sierra Leone
*Serpusia blanchardi* Bolívar, 1905Figure 12C,H–J

Redescription. The elytra are very reduced, barely exceeding the posterior margin of the metanotum, not spotted; the hind femora with longer base, are less enlarged than in *S. opacula* and longer than in *S. opacula*; the antennae are very short.

Distribution. Liberia

Remarks. This species is known only from its type, a single female, deposited in the Entomology Collection of the National Museum of Natural Sciences, Madrid, Spain (MNCN).

Remarks. *Serpusia catamita*, *S. blanchardi*, and *S. inflata* are quite different from the true *Serpusia* externally. For instance, (i) *S. catamita* and *S. inflata* have large lobiform tegmina as *Paraserpusia* species, whereas the true *Serpusia* have vestigial, rudimentary tegmina; (ii) the forewings in the true *Serpusia* have two shiny black spots, whereas there is always only one in *S. catamita* and *S. inflata*; (iii) the posterior margin of the pronotum is distinctly incised and strongly emarginated in the middle, forming an acute angle in *Serpusia*. In contrast, it is rounded and distinctly wavy, with a light brown mottling in the middle in *S. catamita*, and more or less smooth and flat in *S. inflata*. After examining the male internal genitalia of *S. catamita* in search of additional differential characteristics, we found that the phalli of *S. catamita* as a whole differ from those of *Serpusia* in five details: (i) the ancorae of the epiphallus point directly inwards in *Serpusia*, whereas they are slanted forwards in *S. catamita*; (ii) the lophi of the epiphallus are large, lobiform, and rounded in *Serpusia,* whereas they are curved upwards, with angular apices in *S. catamita*; (iii) the phallic complex is robust and more strongly sclerotised in *S. catamita* than in *Serpusia*; (iv) the ejaculatory sac is large, roughly covering two-thirds of the endophallus and completely covering the endophallic sclerites ventrally in *Serpusia*, whereas it is strongly reduced in *S. catamita*; and (v) the endophallic valves are flattened basally in *Serpusia*, whereas they are narrow basally in *S. catamita*. In addition, the tegmina in *S. blanchardi* are much reduced and not spotted as in true *Serpusia*; the hind femora are longer basally and less enlarged than in *Serpusia*; the antennae are shorter than the head and pronotum together in *S. blanchardi*, whereas they are longer than the head and pronotum together in *Serpusia*. Based on these consistent morphological differences, we therefore suggest removing *S. catamita* and *S. inflata* from *Serpusia* and including them in a separate genus, as previously proposed by [4].

#### 3.2.4. The Genus *Paraserpusia* gen. nov.


https://zoobank.org:act:B79EBA9A-9A83-4798-B9FF-BFE2FDF1CEDD


Type species: *Paraserpusia succursor* (Karsch, 1896), by present designation.
*Ptemoblax succursor* Karsch, 1896*Serpusia succursor* (Karsch, 1896)Figure 13A–L

Material examined. Cameroon • 17 ♂♂, 11 ♀♀; Ongot, disturbed forests and forest edges; 3°51′0″ N, 11°22′1″ E; 15 June 2020; J.A. Yetchom Fondjo leg.; SMNK. Cameroon • 16 ♂♂, 13 ♀♀; Ongot, disturbed forests and forest edges; 3°51′0″ N, 11°22′1″ E; 5 December 2021; J.A. Yetchom Fondjo leg.; SMNK. Cameroon • 7 ♂♂, 3 ♀♀; Ongot, in disturbed forests and forest edges; 3°51′0″ N, 11°22′1″ E; 20 March 2022; J.A. Yetchom Fondjo leg.; SMNK.

Diagnosis. *Paraserpusia succursor* is similar to *P. hoeferi* sp. nov. in general coloration, but can be distinguished by the following characteristics: tegmina less broad, only reaching the anterior margin of the metanotum (vs. very broad, strongly widened towards the apex and fairly reaching the posterior margin of the metanotum in *P. hoeferi* sp. nov.; elytra with a rounded apex (vs. with a pointed or acute apex in *P. hoeferi* sp. nov.). Male genitalia differ by its aedeagus, which is of medium size with roughly divergent margins and divergent (vs. long with almost straight margins and parallel tip in *P. hoeferi* sp. nov.); female genitalia differ by its narrow egg-guide (vs. broad in *P. hoeferi*).

*Paraserpusia succursor* is identical to *P. kekeunoui* sp. nov. in general coloration, but can be distinguished by the following features: elytra subpointed more enlarged with a rounded apex (vs. slightly enlarged towards the apex in *P. kekeunoui* sp. nov.); inner subapical spot small or not reduced, almost the same size as the outer apical spot (vs. much reduced, rounded and smaller than the outer apical spot in *P. kekeunoui* sp. nov.); posterior margin of pronotum incised in the middle (vs. almost straight, not incised in *P. kekeunoui* sp. nov.); male genitalia differ by its dorsal arc of cingulum open (vs. close in *P. kekeunoui* sp. nov.); apodemes of cingulum slightly curved ventrally at its apex, and only reaching the tip of the endophallic apodemes (vs. almost parallel or slightly orthogonally arranged basally, converging apically towards an obtuse apex, long fairly exceeding the tip of the endophallic apodemes in *P. kekeunoui* sp. nov.); endophallic apodemes small (vs. endophallic apodemes large in *P. kekeunoui* sp. nov.); aedeagus of medium size, and with roughly divergent margins and broad tip (vs. long, with abruptly oblique margins and acute in *P. kekeunoui* sp. nov.).

*Paraserpusia succursor* is similar to *P. tindoi* sp. nov. (Figure 18) in general coloration, but can be easily distinguished by several characteristics: the male cercus is long (vs. short in *P. tindoi*); the elytra have two shiny black spots (vs. no shiny black spots in *P. tindoi* sp. nov.); the apex of the elytra is slightly widened and rounded (vs. apex of the elytra is narrowed and slightly truncated in *P. tindoi* sp. nov.); male genitalia differ by its elongate phallic complex (vs. less elongate and slightly robust in *P. tindoi* sp. nov.); apodemes of cingulum only reaching the tip of the endophallic apodemes (vs. fairly exceeding the tip of the endophallic apodemes in *P. tindoi* sp. nov.); aedeagus short or of medium size, and with roughly divergent margins and broad tip (vs. aedeagus long, with roughly convergent margins and rounded tip in *P. tindoi* sp. nov.).

Redescription. Male. Antennae black, filiform, longer than the head and pronotum combined; pronotum with lateral carinae plumed black, median carina crossed by three transverse furrows, the typical furrow strongly indented in the middle, with a deep concavity on each side; lateral lobes of the pronotum with a broad, shiny black stripe not reaching the anterior margin, inferior-external half pale, anterior and posterior margins with numerous pale lines; pale metathoracic episternites; elytra lobiform, with subparallel margins, slightly enlarged towards the apex, rounded apex, slightly exceeding the posterior margin of the metanotum, and adorned with two shiny black spots on the outer apical and inner subapical sides, spots sometimes fused in males; lower inner surface, inner basal half of posterior femora dark red, apical half blackish or dark; posterior tibiae with basal half black and apical half red; male cercus short conical, with flattened base and pointed apex, not extending beyond the extremity of the abdomen; male supra-anal plate trigonal, with dorsal surface provided with longitudinal carinae and a transverse groove.

Epiphallus (Figure 13H). Bridge narrow, arched; ancorae small, slanted inwards towards the mid-line; lophi broad, wide, lobiform; oval sclerites small, roughly elongate and strap-like; anterior projections strongly reduced.

Phallic complex (Figure 13I–K). Elongate, small; dorsal arc of cingulum U-shaped, open, its anterior margin not emarginate; the apodemes of cingulum are long and more or less straight, running roughly parallel to each other distally, but slightly curved ventrally at its apex, and only reaching the tip of the endophallic apodemes; endophallic apodemes deeply cupped, and with two channels with a U-shaped cross section running ventrally and posteriorly from their ventral margins to form the gonopore process; endophallic apodemes laterally directed; the flexure of the endophallus is short and thread-like; zygoma only slightly sclerotised; the anterior margin of the cingular arch under the zygoma is broadly emarginate in the midlinerami well developed, sclerotised somewhat, longer than its wide in lateral view, and extending to the ventral midline of the phallus, not fused nor overlaping at their ventral extremities; ectophallic sheath short; aedeagus projecting, of medium size, slightly curved, antero-dorsally directed, and with roughly divergent margins and broad tip; aedeagal valves enveloped/covered by a thick semitransparent sheath, closely appressed to the endophallic valves; endophallic valves long and slender, wider and dorsoventrally flattened basally, but laterally compressed and narrow apically.

They extend almost to the tip of the ecotophallic aedeagal sheath. The valves of cingulum are slender and pointed; the gonopore sac is large, roughly covering the 2/3 of the endophallus and well covering the endophallic sclerites ventrally; it extends more anteriorly up to the U-shaped cross section.

*Female.* Similar to male, but larger; valves of ovipositor of normal shaped, long; cercus short conical; subgenital plate (Figure 13C) hexagonal, long, with truncated posterior margins; anterior apodemes short and broad; egg-guide short, narrow; ventral pockets of the vaginal floor large; copulatory bursa long, almost straight, with parallel margins; bottom of the copulatory bursa close to the arc of the basivalvar sclerites; each basivalvar sclerite barely curved, forming a round angle; Spermatheca duct (Figure 13L) simple, of medium size; the base of the spermathecal duct opening directly at the apex of the copulatory bursa.

Distribution. Cameroon (Ongot, Centre region); Democratic Republic of Congo; Nigeria; Togo.

Habitat. This species occurs in undisturbed and disturbed forests, and in agroforests of the humid zones of Cameroon.

Biology and Ecology. It can be observed throughout the year in its natural habitats in Cameroon’s humid forests. In Cameroon, the species has a narrow distribution range.

*Paraserpusia hoeferi* Yetchom & Wandji sp. nov.


https://zoobank.org:act:54A44878-3B94-4AE1-BA86-681376E5C82F
Figure 14A–L


Material examined. *Holotype*. Cameroon • ♂; Iboti, in the Ebo Forest; 04°27.999′ N, 010°27.357′ E, 742 m a.s.l.; 7 January 2022; J.A. Yetchom Fondjo leg.; SMNK, SMNK-ORTH-0000011.

*Paratypes*. Cameroon • 4 ♂♂, 2 ♀♀; ♂; Iboti, in the Ebo Forest; 04°27.999′ N, 010°27.357′ E, 742 m a.s.l.; 7 January 2022; J.A. Yetchom Fondjo leg.; SMNK.

Diagnosis. *Paraserpusia hoeferi* sp. nov. is similar to *P. succursor* (Figure 13) in general coloration, but can be distinguished by the following characteristics: tegmina very broad, strongly widened towards the apex and reaching the posterior margin of the metanotum (vs. less broad, only reaching the anterior margin of the metanotum in *P. succursor*); elytra with a pointed or acute apex (vs. a rounded apex in *P. succursor*). Male genitalia differ by its aedeagus, which is cylindrical, long, with almost straight margins parallel tip (vs. short or of medium size, with roughly divergent margins in *P. succursor*); female genitalia differ by its broad egg-guide (vs. narrow in *P. succursor*).

Description. Male. Tegument dark brown; fastigium of vertex flattened or short; frons pale; frontal sides speckled with black in front of and behind the eyes; antennae black, filiform, longer than the head and pronotum combined; pronotum with lateral carinae plumed with black, median carinae slightly tectiforme, crossed by three transverse furrows, the typical furrow strongly indented in the middle and bordered by two deep concavities, one on each side; lateral lobes of the pronotum with a broad shiny black band not reaching the anterior margin, pale infero-external half, anterior and posterior margins with numerous pale spots; pale metathoracic episternites; elytra lobiform, greatly enlarged towards the apex, with a pointed or acute apex and exceeding the posterior margin of the metanotum, and reaching the posterior margin of the first abdominal tergite, and adorned with two shining black spots on the outer apical and inner subapical sides; on the inner side, the inner basal half of the posterior femora dark red, the apical half black or dark; posterior tibiae with basal half black and apical half red; male cercus short conical, with flattened base and pointed apex, not extending beyond the extremity of the abdomen; male supra-anal plate trigonal, with dorsal surface provided with longitudinal carinae and a transverse groove.

Epiphallus (Figure 14H). Bridge narrow, slightly more open, forming a more or less reflex angle; ancorae slanted inwards towards the mid-line; lophi broad, wide, lobiform; oval sclerites small, roughly elongate, and strap-like; anterior projections strongly reduced.

Phallic complex (Figure 14I–K). Long; the dorsal arc of cingulum is U-shaped, open, its anterior margin under the zygoma is broadly emarginate in the midline; apodemes of cingulum are long and more or less straight, running roughly parallel to each other distally, slightly curved inwards at its apex, and only reaching the tip of the endophallic apodemes; the endophallic apodemes are more deeply cupped, and from their ventral margins two channels with a U-shaped cross section run ventrally and posteriorly to form the gonopore processes; the ejaculatory sac is situated more anteriorly; the zygoma sclerotised; the flexure of the endophallus is thread-like; the rami well developed, sclerotised, longer than its wide in lateral view, and extending to the ventral midline of the phallus, not fused nor overlaping at their ventral extremities; the valves of cingulum are slender and pointed, extending along the upper margin of the endophallic valves; ectophallic sheath of short; the endophallic valves are long and slender, wider and more dorsoventrally flattened basally; aedeagus projecting, long, antero-dorsally directed, with almost straight margins and parallel and broaded tip; aedeagal valves enveloped/covered by a thick semitransparent sheath, closely appressed to the endophallic valves; gonopore sac large, roughly covering the 1/2 of the endophallus and well covering the endophallic sclerites ventrally.

Female. Similar to male, but larger; cercus short conical; ovipositor of medium size; subgenital plate (Figure 14C) hexagonal, small, with truncated posterior margins; anterior apodemes short and broad; egg-guide short, broad; ventral pockets of the vaginal floor large; copulatory bursa long, almost straight.

Bottom of the copulatory bursa close to the arc of the basivalvar sclerites; each basivalvar sclerite barely curved, forming around angle; spermatheca duct (Figure 14L) simple, long; the base of the spermathecal duct opening directly at the apex of the copulatory bursa.

Etymology. The species is named after Dr. Hubert Hoefer, in recognition of his contribution to the biodiversity and species discovery of Tropical and European spider fauna.

Distribution. Iboti, Ebo Forest, Littoral region of Cameroon (Figure 19).

Habitat. This species is associated with the bare ground in undisturbed and disturbed forests of the humid zones of Cameroon.

Biology and Ecology. It can be observed throughout the year in forest edges in the Ebo Forest zones.

Remarks. Three individuals of the juvenile stage were collected in the same habitat and locality.

*Paraserpusia husemanni* Yetchom & Wandji sp. nov.


https://zoobank.org:act:59CA9322-28A9-44E7-8A75-EC65EEC529F0
Figure 15A–L


Material examined. *Holotype*. Cameroon • ♂; Akom2, disturbed Forest; 2°49′4″ N, 10°33′9″ E, 470 m a.s.l.; 3 July 2024; A.C. Wandji leg.; SMNK, SMNK-ORTH-0000012. *Paratypes*. Cameroon • 9 ♂♂, 12 ♀♀; Akom2, disturbed Forest and cocoa farms; 2°49′4″ N, 10°33′9″ E, 470 m a.s.l.; 3 July 2024; A.C. Wandji leg.; SMNK.

Diagnosis. The new species *P. husemanni* sp. nov. is similar to *P. kekeunoui* sp. nov. from which it can be easily distinguished by: elytra with almost straight apex forming a right angle (vs. elytra with rounded apex in *P. kekeunoui*); elytra barely reaching the front edge of the first abdominal segment (vs. elytra extending well beyond the front edge of the first abdominal segment in *P. kekeunoui* sp. nov.); posterior margin of pronotum curved in the middle (vs. almost straight, not curved in *P. kekeunoui* sp. nov.); male genitalia differ by its long aedeagus, with almost straight or only slightly oblique margins and an obtuse tip (vs. long, with margins abruptly oblique and sharply narrowing towards an acute apex in *P. kekeunoui* sp. nov.); the gonopore sac is small, and is situated more anteriorly (vs. of medium size, is situated more posteriorly in position under the basal endophallus in *P. kekeunoui* sp. nov.).

The new species *P. husemanni* sp. nov. is also similar to *P. tindoi* sp. nov. in general coloration, but can be easily distinguished by a number of characteristics: the elytra are decorated with two conspicuous shiny black spots at the apical and subapical ends (vs. elytra without shiny black spots in *P. tindoi* sp. nov.); elytra barely reaching the front edge of the first abdominal segment (vs. elytra extending beyond the front edge of the first abdominal segment in *P. tindoi* sp. nov.); apex of elytra almost straight, forming a right angle with the lower margin (vs. apex of elytra truncate, obtuse to rounded in *P. tindoi* sp. nov.); male genitalia differ by its elongated phallic complex (vs. slightly robust in *P. tindoi* sp. nov.); apodemes of cingulum almost parallel or slightly orthogonally arranged, long (vs. curved, almost forming a horseshoe-like profile basally, oblique apically in *P. tindoi* sp. nov.); endophallic apodemes large/wide (vs. small in *P. tindoi* sp. nov.); aedeagus conical, long, slightly curved upwards, with almost straight or only slightly oblique margins narrowing towards the obtuse tip (vs. cylindrical, long, strongly curved, and with converging margins and rounded tip in *P. tindoi* sp. nov.); the gonopore sac is situated more anteriorly (vs. it is situated more posteriorly in position under the basal endophallus in *P. tindoi* sp. nov.).

Description. Male. Fastigium of the vertex flattened; tegument pale, plume of black; fros pale; subocular facial spot single, rounded, central; antenna filiform, longer than the head and pronotum combined in both sexes; median carina of the pronotum obtuse, tectiform, crossed by three transverse furrows, the typical furrow bordered by two hollow concavities mottled brilliant black, one on each side.

Posterior margin of the pronotum slightly curved in the middle; lateral lobes of pronotum with a shiny black band; infero-posterior half of lateral lobes of pronotum, pro-, meso- and methatoracic episternites pale; elytra lobiform, not enlarged, with subparallel margins and an almost straight/subrectilinear apex, with two well-formed and separate shiny black spots; elytra barely reaching the anterior margin of the first abdominal segment; inferoexternal surface of posterior femora dark red, internal basal half of posterior femora pale, internal apical half dark, with a black band; posterior tibiae basal half dark and apical half light red; male and female cerci conical.

Epiphallus (Figure 15H). Bridge narrow; ancorae small, slanted inwards towards the midline; lophi wide, lobiform; oval sclerites small; anterior projections strongly reduced.

Phallic complex (Figure 15I–K). Elongate; dorsal arch of cingulum U-shaped, closed, its anterior margin under the zygoma is broadly emarginate in the midline; ectophallic sheath long; apodemes of cingulum almost parallel or slightly orthogonally arranged, long, ritching or slightly exceeding the tip of endophallic apodemes; the endophallic apodemes are wide, more deeply cupped, and from their ventral margins two channels with a U-shaped cross section run ventrally and posteriorly to form the gonopore processes; the flexure of the endophallus is thread-like; zygoma well sclerotised; rami well developed, sclerotised, longer than its wide in lateral view, and extending to the ventral midline of the phallus, and overlaping but not fused at their ventral extremities; aedeagus projecting, long, slightly curved upwards, with almost straight or only slightly oblique margins and an obtuse apex with a ridge-like structure; aedeagal valves enveloped/covered by a thick semitransparent sheath, closely appressed to the endophallic valves; the endophallic valves are long and slender, wide and dorsoventrally flattened basally; the valves of cingulum are slender and pointed; the gonopore sac is small, progressively narrowing towards the posterior margin, and is situated more anteriorly.

Female. Similar to male, but larger; valves of ovipositor narrow, long; subgenital plate (Figure 15C) hexagonal, short, with emaginated posterior margins; anterior apodemes short and broad; egg-guide of medium size; ventral pockets of the vaginal floor large; copulatory bursa long, almost straight, with parallel margins; bottom of the copulatory bursa close to the arc of the basivalvar sclerites; each basivalvar sclerite barely curved, forming around angle; spermatheca duct (Figure 15L) simple, long; the base of the spermathecal duct opening directly at the apex of the copulatory bursa.

Etymology. The species is named after Prof. Dr. Martin Husemann, a distinguished taxonomist in Germany, for his dedication and significant contribution to the evolution of biodiversity, population genetics, phylogenetics, and biogeography of Orthoptera.

Distribution. Akom 2 (Southern Cameroon).

Biology and Ecology. This species occurs throughout the year in disturbed forests and cocoa farms of Southern Cameroon, where it is abundant.

*Paraserpusia kekeunoui* Yetchom & Wandji sp. nov.


https://zoobank.org:act:4A87E215-C956-4970-A52C-CBE5335BE7FB
Figure 16A–L


Material examined. *Holotype*. Cameroon • ♂; Mbalmayo, disturbed forest; 3°29′17″ N, 11°30′6″ E, 679 m a.s.l.; 3 August 2024; A.C. Wandji leg.; SMNK, SMNK-ORTH-0000013. *Paratypes*. Cameroon • 1 ♂; Meyomessala, disturbed forest; 3°7′29″ N, 12°29′48″ E; 18 August 2021; J.A. Yetchom Fondjo leg.; SMNK. Cameroon • 7 ♂♂, 6 ♀♀; Mbalmayo, disturbed forest, forest edges, and cocoa farms; 3°29′17″ N, 11°30′6″ E, 679 m a.s.l.; 3 August 2024; A.C. Wandji leg.; SMNK.

Diagnosis. *Paraserpusia kekeunoui* sp. nov. is identical to *P. succursor* (Figure 13) in general coloration, but can be distinguished by the following features: elytra slightly enlarged towards the apex, apex subpointed (vs. more enlarged with a rounded apex in *P. succursor*); inner subapical shiny black spot much reduced, rounded and fairly smaller than the outer apical spot (vs. inner subapical spot little or not reduced, almost the same size as the outer apical spot in *P. succursor*); posterior margin of pronotum almost straight, not incised (vs. incised in the middle in *P. succursor*). Male genitalia differ by its apodemes of the cingulum that are almost parallel or slightly orthogonally arranged basally, convergent apically, long, and fairly exceeding the tip of the endophallic apodemes (vs. slightly curved ventrally at its apex, and only reaching the tip of the endophallic apodemes in *P. succursor*); endophallic apodemes large (vs. endophallic apodemes small in *P. succursor*); aedeagus long with abruptly oblique margins and converging, narrowing sharply towards an acute tip (vs. short or of medium size, and with roughly divergent margins and broad tip in *P. succursor*); the gonopore sac is situated more posteriorly in position under the basal endophallus (vs. more anteriorly in *P. succursor*).

The new species is similar to *P. hoeferi* sp. nov. (Figure 14) from which it can easily be distinguished by several characteristics: elytra strongly reduced, barely reaching the anterior margin of the metanotum (vs. elytra greatly enlarged towards the apex and reaching the posterior margin of the metanotum in *P. hoeferi* sp. nov.); inner subapical glossy black spot on elytra greatly reduced to one point (vs. almost equal in size in *P. hoeferi* sp. nov.); posterior margin of pronotum almost straight, not curved (vs. curved in the middle in *P. hoeferi* sp. nov.). Male genitalia differ by its strongly arched epiphallus bridge (vs. slightly arched in *P. hoeferi* sp. nov.), its apodemes of cingulum almost parallel or slightly orthogonally arranged basally, fairly exceeding the tip of the endophallic apodemes (vs. slightly curved inwards at its apex, and only reaching the tip of the endophallic apodemes in *P. hoeferi* sp. nov.); its slightly upcurved aedeagus, and with abruptly oblique margins narrowing sharply towards the tip (vs. antero-dorsally directed, and with almost straight margins, and parallel and broaded tip in *P. hoeferi* sp. nov.); the gonopore is situated more posteriorly in position under the basal endophallus (vs. more anteriorly in *P. hoeferi* sp. nov.).

Description. Male. Tegument dark brown; fastigium of vertex flattened; frons pale; frontal sides mottled with black in front of and behind the eyes; filiform antenna longer than the head and pronotum combined, with a black apex; lateral carinae of pronotum mottled with black, median carina tectiform, crossed by three transverse furrows, the typical furrow strongly indented and bordered by two hollow concavities, one on each side; posterior margin of the pronotum almost straight, not incised in the middle; lateral lobes of the pronotum with a broad shiny black stripe not reaching the anterior margin, anterior and posterior margins with numerous small pale spots; the lower half of the lateral lobes of the pronotum pale; elytra lobiform, slightly enlarged, with a subpointed apex barely reaching the anterior margin of the metanotum, ornamented with two shiny black spots on the outer apical and inner subapical, the inner subapical strongly reduced to a point; neck conical; inner basal half of posterior femora dark red, apical half red with a black band; posterior tibiae basal half dark and apical half red.

Epiphallus (Figure 16H). The bridge of the epiphallus is narrow, strongly arched; the ancorae are small, slanted inwards towards the midline; the lophi are wide, lobiform; oval sclerites are longer than wide; the anterior projections are strongly reduced.

Phallic complex (Figure 16I–K). Elongate; dorsal arc of cingulum U-shaped, close, its anterior margin under the zygoma is broadly emarginate in the midline; ectophallic sheath short; apodemes of cingulum almost parallel or slightly orthogonally arranged basally, convergent apically, long fairly exceeding the tip of the endophallic apodemes; endophallic apodemes large, deeply cupped, and from their ventral margins two channels with a U-shaped cross section run ventrally and posteriorly to form the gonopore processes; the flexure of the endophallus is thread-like; zygoma well sclerotized.

Rami well developed, sclerotised, longer than wide in lateral view, and extending to the ventral midline of the phallus, not fused nor overlapping at their ventral extremities; aedeagus projecting, long, slightly curved upwards, with abruptly oblique margins and converging narrow acute tip, with a ridge-like structure at the apex; aedeagal valves enveloped/covered by a thick semitransparent sheath, closely appressed to the endophallic valves; the endophallic valves are long and slender, wider and more dorsoventrally flattened basally, but laterally compressed and narrow apically; the valves of cingulum are slender and pointed; the gonopore sac is of medium size, and is situated more posteriorly in position under the basal endophallus.

Female. Similar to male, but larger; cercus short conical; valves of ovipositor of medium size; subgenital plate (Figure 14C) hexagonal, large, with slightly emaginated posterior margin; anterior apodemes broad; egg-guide short, broad; ventral pockets of the vaginal floor large; copulatory bursa long, almost straight, with parallel margings; bottom of the copulatory bursa close to the arc of the basivalvar sclerites; each basivalvar sclerite barely curved, forming around angle; spermatheca duct (Figure 16L) simple, of medium size; the base of the spermathecal duct opening directly at the apex of the copulatory bursa.

Etymology. The species was named in honour of Professor Sévilor Kekeunou, an important Orthoptera expert in Cameroon, for his dedication and scientific contribution to the advancement of the knowledge of research in Orthoptera Ecology, Biodiversity Research, and Taxonomy.

Distribution. Meyomessala, Mbalmayo, Cameroon.

Habitat. This species is associated with the herbaceous vegetation of forest zones.

Biology and Ecology. *Paraserpusia kekeunoui* sp. nov. is abundant throughout the year in disturbed forests, forest edges, and cocoa farms of the Centre and Southern Cameroon.

*Paraserpusia tamessei* Yetchom & Wandji sp. nov.


https://zoobank.org:act:9045AED9-B996-4504-BDA4-8984DCAD3CB5
Figure 17A–L


Material examined. *Holotype*. Cameroon • ♂; Bangoulap, disturbed forest; 5°5′58″ N, 10°32′20″ E, 1358 m a.s.l.; 29 December 2024; A.C. Wandji & M. Mbadjoun Nzike leg.; SMNK, SMNK-ORTH-0000014. *Paratypes*. Cameroon • 12 ♂♂, 8 ♀♀; Bangoulap, disturbed forest; 5°5′58″ N, 10°32′20″ E, 1358 m a.s.l.; 29 December 2024; A.C. Wandji & M. Mbadjoun Nzike leg.; SMNK. Cameroon • 7 ♂♂, 5 ♀♀; Evodoula, in disturbed Forest, forest edges, and cocoa farms; 4°4′54″ N, 11°11′41″ E, 612 m a.s.l.; 10 January 2025; A.C. Wandji leg.; SMNK. Cameroon • 3 ♂♂, 1 ♀; Mfou, in disturbed Forest and cocoa farms; 3°48′32″ N, 11°40′32″ E, 697 m a.s.l.; 25 January 2025; A.C. Wandji & M. Mbadjoun Nzike leg.; SMNK.

Diagnosis. *Paraserpusia tamessei* sp. nov. is similar to *P. succursor* (Figure 13) from which it can be distinguished by the following characteristics: the tegmina strongly exceed the posterior margin of the metanotum, but not reaching the posterior edges of the first abdominal tergite, with a pointed or acute apex (vs. they are broad, strongly widened towards the apex, and barely reaching the anterior margin of the metanotum, with rounded apex in *P. succursor*); the aedeagus of the male phalli is cylindrical, long, with roughly convergent margins and rounded tip (vs. it is cylindrical, of medium size, with roughly divergent margins and broad tip in *P. succursor*).

The new species is also similar to *P. hoeferi* sp. nov. (Figure 14) from which it can be distinguished by the following characteristics: the tegmina strongly exceed the posterior margin of the metanotum, but not reaching the posterior edges of the first abdominal tergite (vs. they strongly exceed the posterior margin of the metanotum, and fairly reaching the posterior edges of the first abdominal tergite in *P. hoeferi* sp. nov.); the aedeagus of the male phalli is cylindrical, long, with roughly convergent margins and rounded tip (vs. it is cylindrical, long, with almost straight margins and parallel tip in *P. hoeferi* sp. nov.).

The new species is also identical to *P. husemanni* sp. nov. (Figure 15) from which it differs by the tegmina having a pointed or acute apex (vs. they have an almost straight apex, forming a right angle in *P. husemanni* sp. nov.); the aedeagus of the male phalli is cylindrical, long, with roughly convergent margins and a rounded tip (vs. it is conical, long, with almost straight margins and an obtuse tip in *P. husemanni* sp. nov.).

Description. Male. Fastigium of the vertex slightly protroduing forwards; tegument pale, plume of black; frons pale; subocular facial spot single, rounded, central; antenna filiform, much longer than the head and pronotum combined in both sexes; median carina of the pronotum obtuse, tectiform, crossed by three transverse furrows, the typical furrow bordered by two hollow concavities mottled brilliant black, one on each side; the posterior margin of the pronotum posterior is more or less wavy; lateral lobes of pronotum with a shiny black band; infero-posterior half of lateral lobes of pronotum, pro-, meso- and methatoracic episternites pale; a large brown band extending from the cheeks to the apical 1/3 of the lateral lobes of pronotum in soe individuals; elytra lobiform, large/wide, with truncate, acute apex, and with two well-formed and separated shiny black spots; elytra strongly exceeding the posterior margin of the metanotum, but not reaching the posterior edges of the first abdominal tergite; inferoexternal surface of posterior femora dark red, internal basal half of posterior femora pale, internal apical half with alternated pale and black bands; posterior tibiae basal half dark and apical half light red; male cerci conical.

Epiphallus (Figure 17H). Bridge narrow; ancorae small, slanted inwards towards the midline; lophi wide, lobiform; oval sclerites small; anterior projections strongly reduced.

Phallic complex (Figure 17I–K). Elongate; dorsal arch of cingulum U-shaped, open, its anterior margin under the zygoma is broadly emarginate in the midline; ectophallic sheath long; apodemes of cingulum almost parallel or slightly orthogonally arranged, long, only reaching the tip of endophallic apodemes; endophallic apodemes wide, more deeply cupped, and from their ventral margins two channels with a U-shaped cross section run ventrally and posteriorly to form the gonopore processes; flexure of the endophallus thread-like; zygoma well sclerotised; rami well developed, sclerotised, longer than its wide in lateral view, and extending to the ventral midline of the phallus, and overlapping but not fused at their ventral extremities; aedeagus long, slightly curved upwards, with parallel margins and an rounded apex with a ridge-like structure; aedeagal valves covered by a thick semitransparent sheath, closely appressed to the endophallic valves; endophallic valves long and slender, wide and dorsoventrally flattened basally; valves of cingulum slender and pointed; gonopore sac large and situated more posteriorly as in *Serpusia*.

Female. Similar to male, but larger; valves of ovipositor long; subgenital plate (Figure 17C) hexagonal, short, with emaginated posterior margins; anterior apodemes short and broad; egg-guide of thin; ventral pockets of the vaginal floor large; copulatory bursa long, almost straight, with parallel margins; bottom of the copulatory bursa close to the arc of the basivalvar sclerites; each basivalvar sclerite barely curved, forming an obtuse angle; spermatheca duct (Figure 17L) simple, long; the base of the spermathecal duct opening directly at the apex of the copulatory bursa.

Etymology. The species is named after Professor Joseph Lebel Tamesse, a distinguished entomologist and researcher in Cameroon, for his dedication and contribution to the taxonomy and ecology of various insect groups, including Psyllids.

Distribution. Bangoulap, Evodoula, Mfou (West Highlands and Centre regions, Cameroon).

**Figure 17 insects-16-01020-f017:**
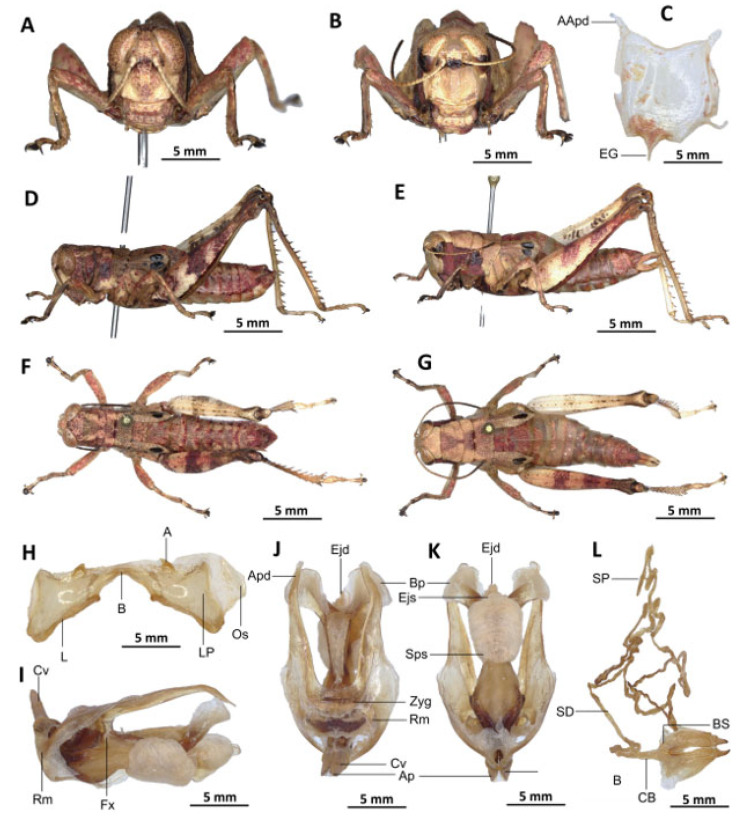
*Paraserpusia tamessei* sp. nov. (**A**) male frontal view; (**B**) female frontal view; (**C**) female subgenital plate; (**D**) male lateral view; (**E**) female lateral view; (**F**) male dorsal view; (**G**) female dorsal view; (**H**) epiphallus dorsal view; (**I**) phallic complex lateral view; (**J**) phallic complex dorsal view; (**K**) phallic complex ventral view; (**L**) female spermatheca. A: ancorae; AApd: anterior apodeme; Ap: apical valves of penis; Apd: apodeme of cingulum; B: bridge of epiphallus; Bp: basal valves of penis; BS: basivalvare sclerites; CB: corpulatory bursa; Cv: valve of cingulum; EG: egg-guide; Ejd: ejaculatory duct; Ejs: ejaculatory sac; Fx: flexure of the endophallus; L: lophus of epiphallus; LP: lateral plate of epiphallus; Os: oval sclerite; Rm: ramus of cingulum; SD: spermathecal duct; SP: spermatheca; Sps: spermatophore sac; Zyg: zygoma of cingulum.

Biology and Ecology. This species is abundant in disturbed forests and cocoa farms in the West Highlands and Centre regions of Cameroon.

*Paraserpusia tindoi* Yetchom & Wandji sp. nov.


https://zoobank.org:act:7D9A3B1B-DCE6-4FE7-A177-3304F87EBA43
Figure 18A–L


Material examined. *Holotype*. Cameroon • ♂; Somalomo, Dja Biosphere Reserve, forest habitat; 3°22′55″ N, 12°44′30″ E, 607 m a.s.l.; 10 April 2022; J.A. Yetchom Fondjo leg.; SMNK, SMNK-ORTH-0000015. *Paratypes*. Cameroon • 6 ♂♂, 4 ♀♀; Somalomo, Dja Biosphere Reserve, forest habitat; 3°22′55″ N, 12°44′30″ E, 607 m a.s.l.; 10 April 2022; J.A. Yetchom Fondjo leg.; SMNK. Cameroon • 5 ♂♂, 2 ♀♀; Somalomo, in the Dja Biosphere Reserve, forest habitat; 3°22′55″ N, 12°44′30″ E, 607 m a.s.l.; 10 April 2022; J.A. Yetchom Fondjo leg.; SMNK.

Diagnosis. This species is similar to *P. succursor* (Figure 13) in general coloration, but can be easily distinguished by several characteristics: the male cercus is short (vs. long in *P. succursor*); the elytra have no shiny black spots (vs. two shiny black spots in *P. succursor*); the apex of the elytra is narrowed and slightly truncated (vs. apex of the elytra is slightly widened and rounded in *P. succursor*); male genitalia differ by its less elongate and slightly robust phallic complex (vs. of medium size and elongate in *P. succursor*), apodemes of cingulum fairly exceeding the tip of the endophallic apodemes (vs. only reaching the tip of the endophallic apodemes in *P. succursor*); aedeagus long, with converging margins and rounded tip (vs. aedeagus of medium size, with roughly divergent margins and broad tip in *P. succursor*); the gonopore is situated more posteriorly in position under the basal endophallus (vs. more anteriorly in *P. succursor*).

The new species is also similar to *P. hoeferi* sp. nov. (Figure 14) from which it can be easily distinguished by the following characteristics: elytra reduced, with no black spots and a narrowed and slightly truncated apex (vs. elytra large and greatly enlarged towards the apex, and a pointed apex with two shiny black spots in *P. hoeferi* sp. nov.); elytra extending beyond the posterior margin of the metanotum and limited to the anterior margin of the first abdominal segment (vs. elytra not reaching the posterior margin of the metanotum in *P. hoeferi*). Male genitalia differ by its apodemes of cingulum fairly exceeding the tip of the endophallic apodemes (vs. only reaching the tip of the endophallic apodemes in *P. hoeferi* sp. nov.); aedeagus cylindrical, long, with converging margins and rounded tip (vs. cylindrical, long, with almost straight margins and parallel tip in *P. hoeferi* sp. nov.); gonopore sac of medium size, and situated more posteriorly in position under the basal endophallus (vs. more anteriorly in *P. hoeferi* sp. nov.).

This species is also similar to *P. kekeunoui* sp. nov. (Figure 16) from which it can be distinguished by the following characteristics: elytra without a shiny black spot (vs. two shiny black spots in *P. kekeunoui*); inner surface of posterior femora completely red (vs. inner surface of posterior femora apically red, basally pale in *P. kekeunoui*); anterior and posterior edges of pronotum curved in the middle (vs. not curved, almost straight in *P. kekeunoui*); antennae longer than head and pronotum combined in both sexes (vs. antennae shorter or as long as head and pronotum combined in *P. kekeunoui*); male genitalia differ by its less elongated and slightly robust phallic complex (vs. elongate in *P. kekeunoui*); the apodemes of cingulum are curved, forming a horseshoe-like profile basally, oblique apically (vs. almost parallel or slightly orthogonally arranged basally, convergent apically in *P. kekeunoui*); endophallic apodemes small (vs. large in *P. kekeunoui*); aedeagus of medium size, strongly curved, antero-dorsally directed, and with converging margins and rounded tip (vs. conical long, slightly curved upwards, with abruptly oblique margins narrowing sharply towards an acute tip in *P. kekeunoui*).

Description. Male. Head straight in lateral view, fastigium of the vertex, frons and cheeks pale; subocular facial spot rounded in the centre; antennae pale-black, filiform, longer than the head and pronotum combined; a broad shiny black ribbon running from the back of the eyes, across the superolateral face of the lateral lobes of the pronotum and extending to the lateral-superior face of the abdomen; dorsal surface of pronotum pale, with black plume and three transverse furrows; typical furrow strongly indented on either side of median carina; anterior margin of pronotum slightly curved in the middle, posterior margin of pronotum with numerous furrows dorsally; lateral lobes of pronotum with a shiny black band clearly reaching the anterior margin; infero-anterior and infero-posterior margins of lateral lobes of pronotum pale; pale pro-, meso-, and metathoracic episternites; elytra lobiform, narrowing towards the apex, with parallel margins, and a slightly truncate apex, slightly exceeding the posterior margin of the metanotum; shining black spots on the elytra complement absent; inferior margins of the elytra outlined in black; prosternal tubercle short conical with flattened base; front and middle legs pale; lower outer surface, inner surface of posterior femora dark red; outer surface of posterior femora with three broad black facis or spots, variegated with pale spots; posterior tibiae with dark basal half and light red apical half; external apical spines on posterior tibiae absent; arolium and claws well developed; ventral surface of abdomen pale; male supra-anal plate trigonal, with a groove through the basal third; male cerci short conical with slightly flattened base and acute apex; male subgenital plate conical with rounded apex.

Epiphallus (Figure 18H). The bridge is narrow, arched; the ancorae are small and slanted inwards towards the midline; lophi wide, lobiform; oval sclerites small and roughly triangular; anterior projections strongly reduced.

Phallic complex (Figure 18I–K). Less elongate and lightly robust; dorsal arch of cingulum is U-shaped, strongly open, and curved to form a slightly horseshoe-like profile in dorsal view, its anterior margin under the zygoma is broadly emarginate in the midline; ectophallic sheath short; apodemes of cingulum curved, forming a horseshoe-like profile basally, oblique apically, and fairly exceeding the tip of the endophallic apodemes; endophallic apodemes small, deeply cupped and ventrally directed, and from their ventral margins two channels with a U-shaped cross section run ventrally and posteriorly to form the gonopore processes; the flexure of the endophallus is very slender; zygoma only slightly sclerotised; rami well developed, sclerotised somewhat, longer than its wide in lateral view, and extending to the ventral midline of the phallus, not fused nor overlaping at their ventral extremities; the valves of cingulum are slender and pointed; the endophallic valves are long and slender, wide and dorsoventrally flattened basally, but more reduced than in *P. surccusor*; aedeagus cylindrical, long, strongly curved upwards, antero-dorsally directed, and with oblique margins and rounded tip; aedeagal valves covered by a thick semitransparent sheath, closely appressed to the endophallic valves; the gonopore sac is of medium size, and is situated more posteriorly in position under the basal endophallus.

Female. Similar to male, but larger; cerci conical; supra-anal plate trigonal with obtuse apex, crossed by a groove in its basal half; valves of ovipositor narrow, long; subgenital plate (Figure 18C) hexagonal, long, with truncated posterior margins; anterior apodemes broad; egg-guide of medium size; ventral pockets of the vaginal floor large; copulatory bursa long, almost straight, with parallel margins; bottom of the copulatory bursa close to the arc of the basivalvar sclerites; each basivalvar sclerite barely curved, forming a round angle; spermatheca duct (Figure 18L) simple, of medium size; the base of the spermathecal duct opening directly at the apex of the copulatory bursa.

**Figure 18 insects-16-01020-f018:**
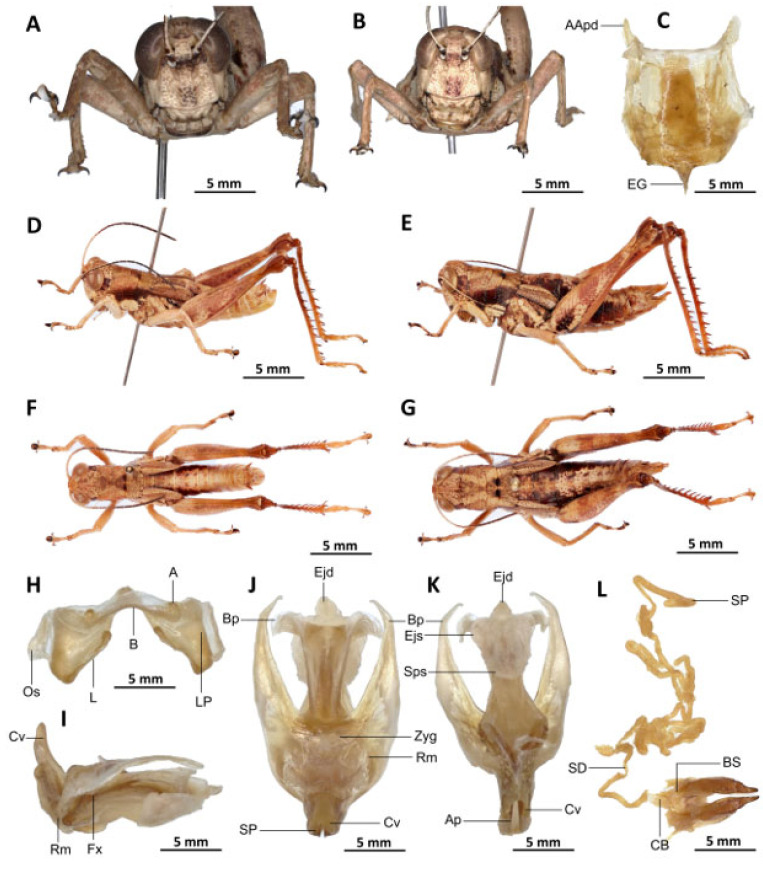
*Paraserpusia tindoi* sp. nov. (**A**) male frontal view; (**B**) female frontal view; (**C**) female subgenital plate; (**D**) male lateral view; (**E**) female lateral view; (**F**) male dorsal view; (**G**) female dorsal view; (**H**) epiphallus dorsal view; (**I**) phallic complex lateral view; (**J**) phallic complex dorsal view; (**K**) phallic complex ventral view; (**L**) female spermatheca. A: ancorae; AApd: anterior apodeme; Ap: apical valves of penis; Apd: apodeme of cingulum; B: bridge of epiphallus; Bp: basal valves of penis; BS: basivalvare sclerites; CB: corpulatory bursa; Cv: valve of cingulum; EG: egg-guide; Ejd: ejaculatory duct; Ejs: ejaculatory sac; Fx: flexure of the endophallus; L: lophus of epiphallus; LP: lateral plate of epiphallus; Os: oval sclerite; Rm: ramus of cingulum; SD: spermathecal duct; SP: spermatheca; Sps: spermatophore sac; Zyg: zygoma of cingulum.

Etymology. The species was named in honour of Professor Maurice Tindo, a distinguished entomologist and researcher in Cameroon, for his significant contribution and achievement in the field of insect ecology, biodiversity, species interactions, and conservation biology.

Distribution. Somalomo, Dja Biosphere Reserve (Cameroon) (Figure 19).

Biology and Ecology. This species is associated with disturbed forests in southern Cameroon. Adults are present throughout the year in the natural environment. Mating is observed between April and June.

Key to species of the genus *Paraserpusia* gen. nov.

Except for *P. tindoi* sp. nov., whose phallus is slightly robust as in *Aresceutica*, all *Paraserpusia* species have a very similar phallic complex, but differ from each other by the shape of the aedeagus. As such, five types of aedeagus can be found in *Paraserpusia* (Figure 11, Figure 12, Figure 13, Figure 14, Figure 15 and Figure 16): (1) aedeagus cylindrical, short or of medium size, with divergent margins and broad tip, (2) aedeagus cylindrical, long with almost straight margins and parallel tip, (3) aedeagus cylindrical, long, with convergent margins and rounded tip; (4) aedeagus conical, long, with abruptly oblique margins narrowing towards an obtuse tip, and (5) aedeagus conical, long, with almost straight or only slightly oblique margins and obtuse apex having a ridge-like structure.

A.Based on external morphology


Tegmina lobiform, without shiny black spots (Figure 18D–G)
*Paraserpusia tindoi* Yetchom & Wandji sp.nov.
-Tegmina lobiform, with two shiny black spots
2
2.Tegmina broad, strongly widened towards the apex, and reaching the posterior margin of the metanotum
4
-Tegmina broad, strongly widened towards the apex, and not reaching the anterior margin of the metanotum
3
3.Tegmina with rounded apex (Figure 13D–G)
*Paraserpusia succursor* (Karsch, 1896)
-Tegmina with a subpointed apex (Figure 16D–G)
*Paraserpusia kekeunoui* Yetchom & Wandji sp. nov.
4.Tegmina with almost straight apex forming a right angle (Figure 15D–G)
*Paraserpusia husemanni* Yetchom & Wandji sp. nov.
-Tegmina with a pointed or acute apex
5
5.Tegmina strongly exceeding the posterior margin of the metanotum, but not reaching the posterior edges of the first abdominal tergite (Figure 17D–G)
*Paraserpusia tamessei* Yetchom & Wandji sp. nov.

-Tegmina strongly exceeding the posterior margin of the metanotum, and fairly reaching the posterior edges of the first abdominal tergite (Figure 14D–G)
*Paraserpusia hoeferi* Yetchom & Wandji sp. nov.The distribution range of all species treated in this study is shown in Figure 19 below.

**Figure 19 insects-16-01020-f019:**
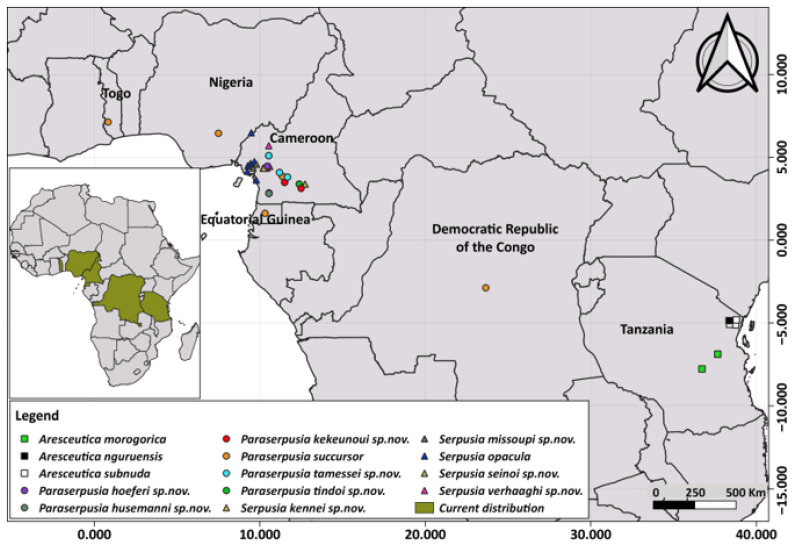
Distribution of *Aresceutica*, *Paraserpusia*, and *Serpusia* species in African forests.

B.Based on the structures of the phallus


Phallic complex elongate, not robust
2
-Phallic complex less elongate and slightly robust, apodemes of cingulum curved, forming a horseshoe-like profile basally, oblique apically (Figure 18I–K)
*Paraserpusia tindoi* Yetchom & Wandji sp. nov.
2.Apodemes of cingulum slightly curved ventrally at its apex, and only reaching the tip of the endophallic apodemes
3
-Apodemes of cingulum almost parallel or slightly orthogonally arranged, long exceeding the tip of the endophallic apodemes
4
3.Aedeagus cylindrical, of medium size, with roughly divergent margins and broad tip (Figure 13I–K)
*Paraserpusia succursor* (Karsch, 1896)
-Aedeagus cylindrical, long, with almost straight margins and parallel tip (Figure 14J)
*Paraserpusia hoeferi* Yetchom & Wandji sp. nov.
4.Aedeagus cylindrical, long, with roughly convergent margins and a rounded tip (Figure 17J)
*Paraserpusia tamessei* Yetchom & Wandji sp. nov.  Aedeagus conical5
5.Aedeagus conical, long, with abruptly oblique margins and acute tip; gonopore sac situated more posteriorly under the basal endophallus (Figure 16J)
*Paraserpusia kekeunoui* Yetchom & Wandji sp. nov.
-Aedeagus conical, long, with almost straight or only slightly oblique margins and an obtuse tip; gonopore sac is small, and is situated more anteriorly (Figure 15J)
*Paraserpusia husemanni* Yetchom & Wandji sp. nov.

Taxonomic decisions and justification

Although *P. succursor* and *P. tamessei* overlap geographically, they form distinct COI-16S clades, and they also exhibit discrete forewing and genitalia differences—thus supporting their recognition as separate species. *Paraserpusia tindoi* and *P. hoeferi* form distinct COI-16S clades, and do not overlap geographically as *P. tindoi* occurs exclusively in the Eastern region of Cameroon (Somalomo) and *P. hoeferi* in the Littoral region of Cameroon (Iboti). Both also exhibit discrete forewing and genitalia differences—thus supporting their recognition as separate species. *Paraserpusia kekeunoui* occurs exclusively in Mbalmayo (centre region of Cameroon), whereas *P. husemanni* is restricted to the southern region of Cameroon (Akom 2). Both species form distinct COI-16S clades and exhibit discrete forewing and genitalia differences—thus supporting their recognition as separate species. Additionally, although *S. missoupi* and *S. verhaaghi* overlap geographically (both species being found in the Littoral region of Cameroon), they exhibit discrete forewing and genitalia differences, and form well-supported clades in the phylogenetic tree inferred from a combined COI and 16S dataset—thus supporting their recognition as separate species. *Serpusia kennei* is geographically distributed in the Littoral region of Cameroon, whereas *S. seinoi* occurs exclusively in the Eastern region of Cameroon. Both species exhibit discrete forewing morphology and form well-supported clades in the phylogenetic tree inferred from a combined COI and 16S dataset—thus supporting their recognition as separate species.

## 4. Discussion

Efficacy of DNA barcoding. Many large-scale attempts have been made to generate DNA barcodes of orthopteran insects from diverse geographical regions worldwide. So far, only two studies have been conducted in Cameroon, focusing on genera of the subfamilies Pyrgomorphinae [31] and Catantopinae [14]. This has contributed to a better understanding of the actual diversity of Acridoidea from Cameroon. The present study highlighted the efficacy of using DNA to delineate species boundaries in the West African grasshopper genus *Serpusia* from Cameroon and the East African genus *Aresceutica*. Our combined molecular dataset, analysed with BI and ML methods, resulted in a strongly supported phylogenetic tree supporting five distinct species within the *Serpusia opacula* complex and six distinct species within the *Serpusia succursor* complex.

For our entire dataset, our results showed the existence of 19 BIN species with a wide distribution range, seven of which belong to *S. succursor*, five to *S. opacula,* and the remaining seven to the outgroup species. The five separated BIN species observed in *S. opacula* correspond to sister clusters in the phylogenetic tree and may represent true hidden undescribed diversity within *Serpusia*. Although the *S. succursor* populations from Bangoulap, Evodoula, and Mfou were not strongly supported on the phylogenetic tree, these populations were assigned to two BINs (BOLD: AGK3117 and BOLD: AGK3116). These widespread BIN species are likely to represent true cryptic species, as no consistent differences were observed in their external morphology or in the structures of the phallic complex of the examined material from the three locations. Further studies on material from a wider geographic area are needed to better elucidate the phylogeographical aspect of this diversity. Two cases of barcode sharing were found within the *S. succursor* complex, which were not consistent with the Phylogenetic results. The barcode and BIN sharing in closely related species of Orthoptera may be caused by the presence of mitochondrial pseudogenes (numts), hybridization, and incomplete lineage sorting [32,33,34]. Evidence of barcode and BIN sharing has previously been reported in the closely related bush-cricket species of the genus *Gampsocleis* by [35].

The recovery of more than one MOTU for *Aresceutica subnuda*, *Paraserpusia husemanni* sp. nov., and *Paraserpusia tamessei* sp. nov. in the barcode gap analysis may indicate possible cryptic diversity within these species. Indeed, BI analysis showed a polytomy for the specimens of *P. tamessei* collected from Bangoulap, Evodoula, and Mfou. Bangoulap, located in the West Highlands of Cameroon, is geographically separated from Evodoula and Mfou (both located in the semi-deciduous forest zones of the Centre region of Cameroon) by 214 km and 324 km, respectively. The geographical isolation of the Bangoulap lineage from the lineages of Evodoula and Mfou may suggest allopatric speciation of *P. tamessei* within its distribution range. This speciation mechanism is considered the most likely in animal taxa [36]. Cryptic species refer to morphologically indistinguishable taxa [37] but genetically distinct [38], making them difficult to identify based solely on phenotypic traits. This phenomenon often occurs when a single species becomes fragmented into isolated populations over time, leading to genetic divergence that is not immediately reflected in morphology. Factors contributing to the emergence of cryptic species include recent speciation events [39], genetic divergence [40], and stabilizing selection slowing phenotypic evolution [41,42], all of which can result in similar morphological appearances despite reproductive isolation. Allopatric speciation—driven by geographical barriers or human-induced factors such as habitat fragmentation—plays a significant role in the development of cryptic species [43,44] by isolating populations and preventing gene flow [45]. These isolated populations may have unique genetic signatures, leading to hidden evolutionary lineages that can easily be misidentified. Such misidentification can have substantial implications for biodiversity assessments, conservation strategies, and ecosystem management [43], as failing to recognize cryptic species may result in an underestimation of biodiversity and the potential loss of important genetic resources. Understanding cryptic species is crucial for effective conservation efforts. Their often small and isolated populations make them particularly vulnerable to extinction, underscoring the need for targeted strategies to protect these hidden lineages [46]. Recognizing and addressing the existence of cryptic species is vital for preserving ecosystems and ensuring that conservation planning is based on an accurate understanding of biodiversity. Allopatric speciation has also been demonstrated in *Oxya japonica* (Thunberg, 1824) by [47].

In addition, this study detected a mean intraspecific genetic distance of 3.57%, which could be used as a threshold for species delineation. An intraspecific genetic distance of >3.00% has been proposed as a threshold for arthropod species delimitation [48,49]. However, this barcode gap may vary among different taxa due to different molecular evolutionary rates within mitochondrial genes [50,51].

Taxonomic implications. The genus *Serpusia* has not been previously revised, and the previously known *S. succursor* has long been included in the genus *Serpusia* based on the external morphology and the structure of the phallic complex. Although this species shares some morphological characters with *S. opacula*, the molecular analyses showed that *Serpusia* s. str. and *Aresceutica* are closely related genera. The relationship between these genera was previously reported by [4] based on the analyses of the external morphology and the details of the phallic complex. In addition, the phylogenetic analyses revealed major splits between the *S. opacula* complex and the *S. succursor* complex, suggesting separate genera. Therefore, we removed the *S. succursor* species complex from the genus *Serpusia* and included it in a new genus that we erected, *Paraserpusia* gen. nov. The genus *Serpusia* should only contain species with rudimentary, narrow, and vestigial tegmina, whereas *Paraserpusia* and *Aresceutica* comprise species with lobiform tegmina.

Due to certain resemblances, *Serpusia* and *Aresceutica* were attempted to be synonymized based on a single morphological description [5]. However, one morphological character that reliably distinguishes the two genera were found, precisely the shape of the prosternal tubercle, which is short, acutely conical, and vertical in *Serpusia*, whereas it is long, subconical, antero-posteriorly compressed, and tilted backward in *Aresceutica* [4]. However, despite their similar external morphology, many discriminatory characters on the internal phallic structures were found [4]. For instance, they reported that the valves of the cingulum of *Serpusia* are slender and pointed, not short and laterally flattened as in *Aresceutica*.

Moreover, we found that the size of the valves of the cingulum is variable within the genus *Serpusia*, some species have slender valves of the cingulum, as in true *S. opacula*, whereas other species have short valves of the cingulum, as in *S. kennei* sp. nov. and *S. verhaaghi* sp. nov. In addition, the size and shape of the aedeagus are variable within *Serpusia*. We distinguished three types of aedeagus within the examined *Serpusia* specimens: (i) aedeagus long and slender, with almost straight and parallel margins and tip (Figure 7); (ii) aedeagus short with broad and divergent tip (Figure 11); (iii) aedeagus short with oblique margins and convergent tip (Figure 8).

In terms of external morphology, we found reliable distinguishing characteristics between the genera *Serpusia* and *Paraserpusia* gen. nov. First, the tegmina in *Serpusia* are rudimentary and narrow, whereas in *Paraserpusia* gen. nov., they are lobiform as in *Aresceutica*. Secondly, the posterior margin of the pronotum is distinctly incised and strongly emarginated in the middle, forming an acute angle in *Serpusia*. In contrast, it is rounded and distinctly wavy, with light brown mottling in the middle, in *Paraserpusia* gen. nov. Thirdly, the median carina of the pronotum is slightly tectiform and not depressed behind the posterior sulcus in *Serpusia*; in *Paraserpusia* gen. nov., it is obtuse and fairly tectiform, with a deep depression behind the posterior sulcus. These differences were not highlighted by [3,4,8,9,52,53], who classified both species as belonging to the same genus.

Unfortunately, the so-called *Serpusia blanchardi*, *S. catamita,* and *S. inflata* were not included in the molecular analyses, as we did not have specimens of *S. inflata and S. blanchardi* at the time of the analyses. In addition, the DNA extracted from the samples of *S. catamita* that were kindly loaned to us by the Natural History Museum Stuttgart, Germany (SMNK), was very fragmented and thus could not be sequenced with success. However, these species are distinguished from the true *Serpusia* by consistent morphological differences. We therefore strongly agree with [4], who suggested removing *S. catamita* and *S. inflata* from *Serpusia* and placing them in a separate genus. Therefore, further investigations relying on the anatomy of the phallic complex and molecular analyses are needed to clarify the taxonomic position of the so-called *S. blanchardi*, *S. catamita,* and *S. inflata*.

Phylogeographic patterns. Our haplotype network analysis, which was based on the COI-5P fragment of *Serpusia* species complex and *Aresceutica*, divided the 78 sequences into three distinct clades. The first clade is represented by the *S. opacula* species complex and includes its respective species and all sampled locations in Cameroon. The second clade is represented by the *S. succursor* species complex and includes all other specimens from various locations in Cameroon. The third clade is represented by *Aresceutica* and includes all sampled locations in Tanzania. The separation of *S. opacula* complex, *S. succursor* complex, and *Aresceutica* is consistent with the results of the phylogenetic analysis. This study shows that *Serpusia* sensu stricto has a restricted distribution in Cameroon, occurring exclusively in the Littoral and Southwest regions. These regions are characterized by an equatorial climate with unimodal rainfall distribution. This genus is typically found in relatively stable tropical rainforests, where it is very scarce. *Serpusia opacula* was reported in the forest area on both sides of the Nigeria-Cameroon border zone by [54]. The type species, *S. opacula*, was collected at Barombi station in the Southwest Region of Cameroon. Our findings are consistent with those of [2,55], who found that *Serpusia* was restricted to forest habitats in the Littoral region of Cameroon. However, *S. opacula* was reported from Mbalmayo (Centre Region, Cameroon) and Bangoulap (West Region, Cameroon) [56]. Molecular analyses revealed that the identification by [56] was confounded with the hidden species *Paraserpusia kekeunoui* sp. nov. and *Paraserpusia tamessei* sp. nov., respectively. The current distribution of *Serpusia* in Cameroon comprises four lineages of *S. opacula* (Barombi Station, Buea, Kumba, and Mouanko), four lineages of *S. verhaaghi* sp. nov. (Bekob, Djawara, Iboti, and Sohock), two lineages of *S. kennei* sp. nov. (Bissoue and Njuma), two lineages of *S. missoupi* sp. nov. (Kompina and Sole), and one lineage of *S. seinoi* sp. nov. (Somalomo). In contrast, the newly erected *Paraserpusia* gen. nov., with *Paraserpusia succursor* (formerly *S. succursor*) as the type species, is widely distributed in Cameroon. It is present in the Centre, East, Littoral, South, and West regions, with the highest species diversity in the Centre region. However, *S. succursor* (now *P. succursor*) was reported in Buea, Kumba, and Tombel in the Southwest region of Cameroon [56]. Re-examination of specimens collected by these authors from the locations mentioned above showed that the species reported by these authors was *S. opacula* rather than ‘*S. succursor*’. Species of this genus show a high tolerance of varying climatic conditions and habitats. For example, the Centre, East, Littoral, South, and West regions have an equatorial climate characterized by heavy rainfall and constant average temperatures. Two climatic types are distinguished: (i) the classic Guinean climate, which prevails in the Centre Region and has four seasons (bimodal distribution of rainfall); and (ii) the Cameroon climate, which prevails in the Southwest, Littoral, and West Highlands and has two seasons (unimodal distribution of rainfall). *Paraserpusia* species inhabit highly disturbed forests, forest edges, clearings, and agrosystems such as cocoa plantations. They are therefore characteristic of disturbed habitats, where they occur in high abundance. *P. tamessei* sp. nov. was the most common species, being present and abundant in Bangoulap, Evodoula, and Mfou. *P. kekeunoui* sp. nov. was present in two locations (Mbalmayo and Meyomessala), while *P. succursor*, *P. hoeferi* sp. nov., *P. husemanni* sp. nov., and *P. tindoi* sp. nov. were each restricted to a single location.

## 5. Conclusions

Molecular species delimitation analyses generated a larger number of MOTUs than morphologically defined species. If these MOTU splits are proven to be true, the West and Central Tropical African grasshopper genera *Serpusia* and *Paraserpusia* gen. nov. probably likely comprise a substantial proportion of cryptic or undescribed taxa. Future amplification of additional molecular markers, particularly from nuclear DNA, may be especially valuable for clarifying problematic taxa identified in this study. Our results provide new insights into the diversity of West African tropical grasshoppers, a fauna that has been largely overlooked. They also highlight the potential of refined techniques and finer taxonomic resolution to improve biodiversity assessment in African rainforest ecosystems. To achieve this, expand across diverse habitats in Cameroon, combined with improved methodological approaches, will be required. Such efforts are essential for mapping the true distribution patterns of *Serpusia* and *Paraserpusia*, and for resolving the status of their species complexes, which may include more species than currently recognized.

## Figures and Tables

**Figure 1 insects-16-01020-f001:**
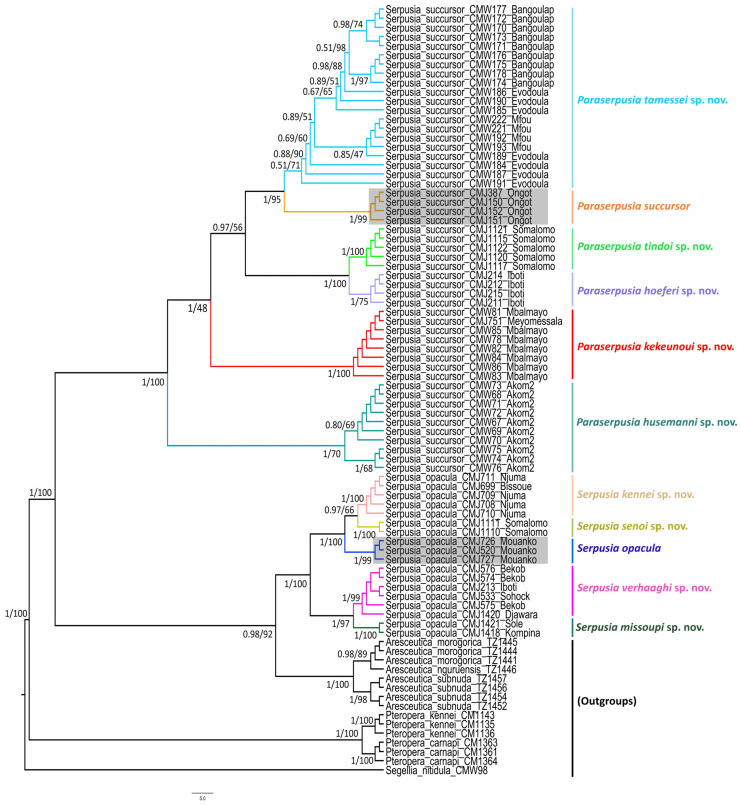
Phylogenetic tree inferred from Bayesian Inference and Maximum Likelihood analyses, using the combined dataset of the COI-5P and 16S rDNA sequences of initially identified Serpusia populations. Numbers at the nodes indicate posterior probabilities and bootstrap values (PP/BS). Branch colours identify species, and pale grey boxes indicate the two previously known species. The voucher codes and the localities follow the names of taxa.

**Figure 2 insects-16-01020-f002:**
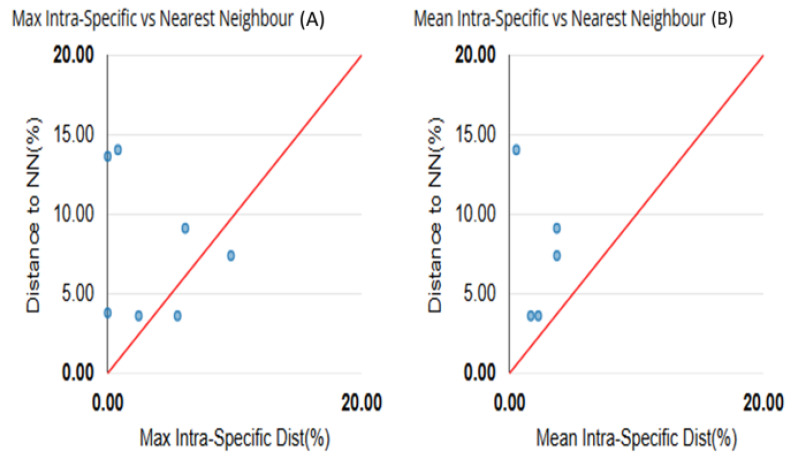
Scatterplots showing the overlap of the maximum (**A**) and mean (**B**) intra-specific distances K2P for 84 specimens vs. the inter-specific (nearest neighbour) distances for the COI-5P marker. Dots above the 1:1 line indicated the presence of a barcode gap.

**Figure 3 insects-16-01020-f003:**
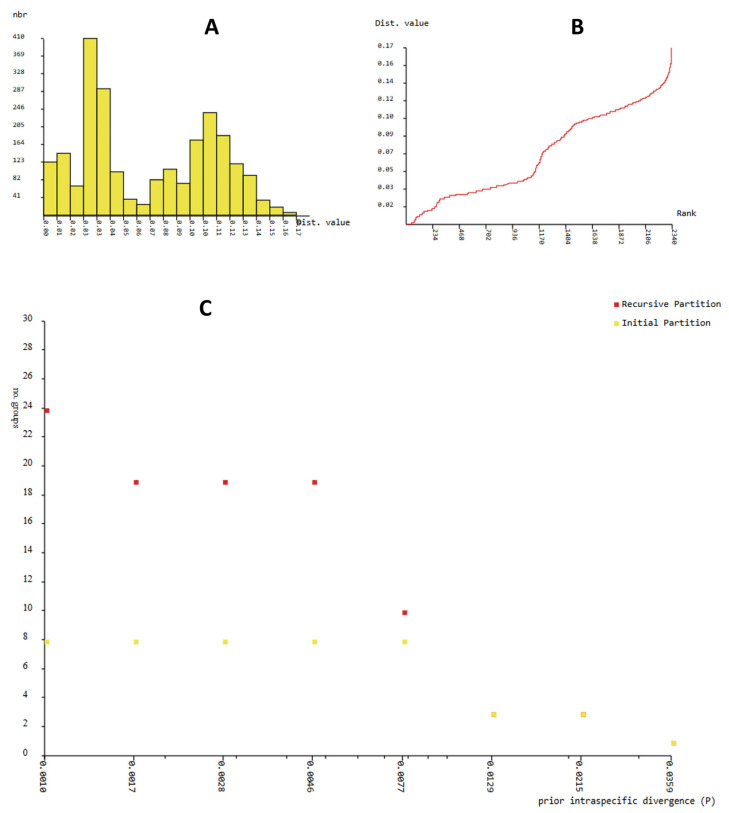
Output of the Automatic Barcode Gap Discovery (ABGD) analysis for COI-5P sequences of the *Serpusia* complex. (**A**) Distribution of pairwise distance; (**B**) Ranked pairwise distances; (**C**) Initial and recursive partitions.

**Figure 4 insects-16-01020-f004:**
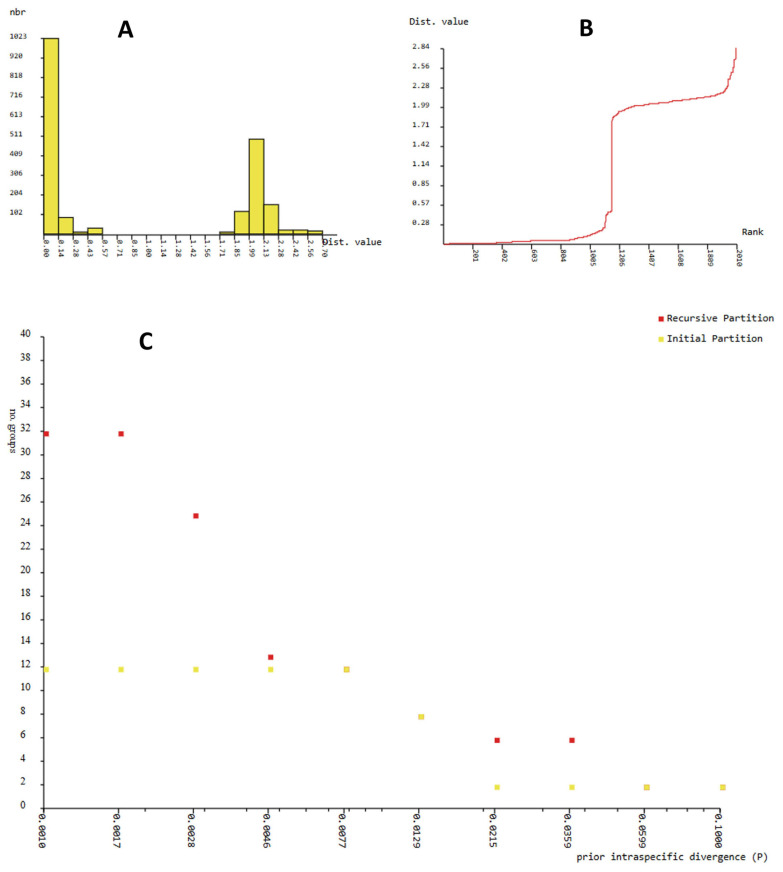
Output of the Automatic Barcode Gap Discovery (ABGD) analysis for 16S rDNA sequences of the *Serpusia* complex. (**A**) Distribution of pairwise distance; (**B**) Ranked pairwise distances; (**C**) Initial and recursive partitions.

**Figure 5 insects-16-01020-f005:**
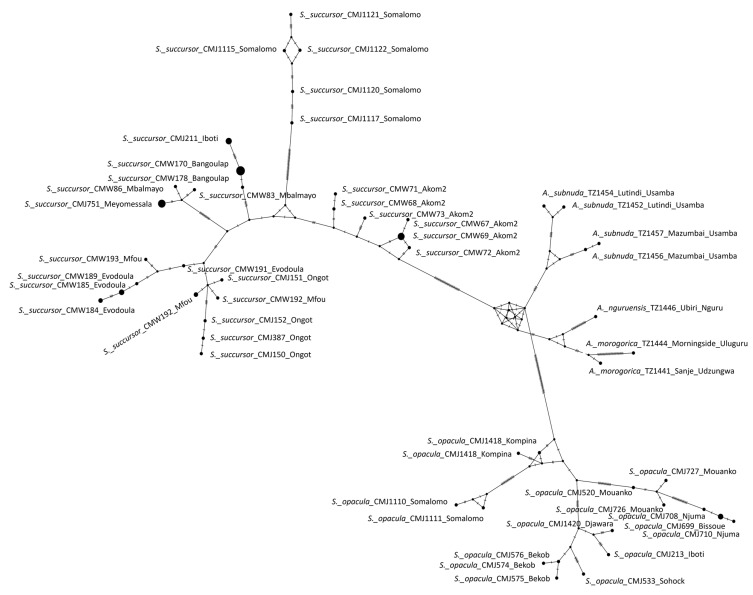
A median-joining haplotype network based on the COI-5P sequences.

**Figure 6 insects-16-01020-f006:**
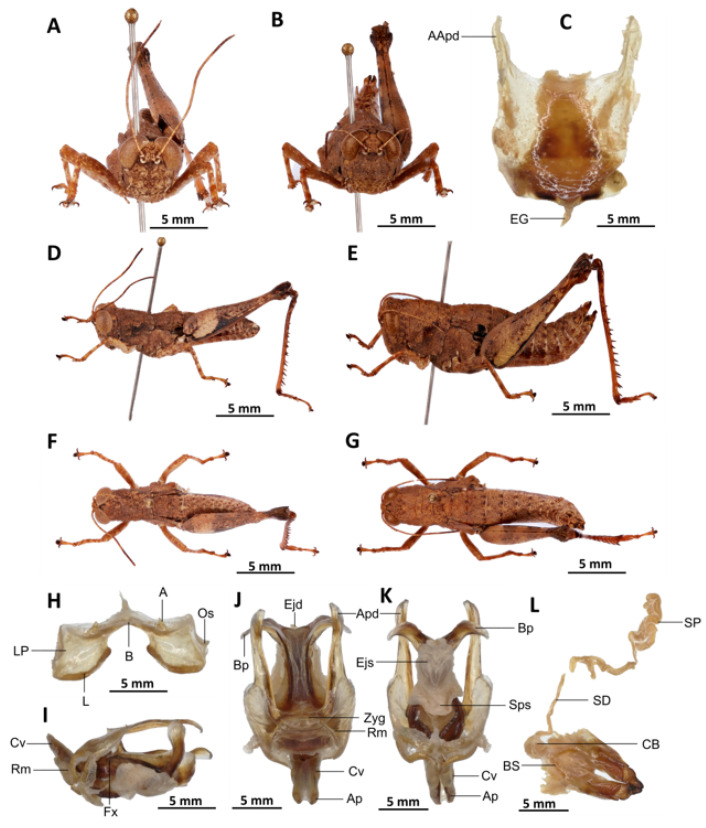
*Serpusia opacula* Karsch, 1891. (**A**) male frontal view; (**B**) female frontal view; (**C**) female subgenital plate; (**D**) male lateral view; (**E**) female lateral view; (**F**) male dorsal view; (**G**) female dorsal view; (**H**) epiphallus dorsal view; (**I**) phallic complex lateral view; (**J**) phallic complex dorsal view; (**K**) phallic complex ventral view; (**L**) female spermatheca. A: ancorae; AApd: anterior apodeme; Ap: apical valves of penis; Apd: apodeme of cingulum; B: bridge of epiphallus; Bp: basal valves of penis; BS: basivalvare sclerites; CB: corpulatory bursa; Cv: valve of cingulum; EG: egg-guide; Ejd: ejaculatory duct; Ejs: ejaculatory sac; Fx: flexure of the endophallus; L: lophus of epiphallus; LP: lateral plate of epiphallus; Os: oval sclerite; Rm: ramus of cingulum; SD: spermathecal duct; SP: spermatheca; Sps: spermatophore sac; Zyg: zygoma of cingulum.

**Figure 7 insects-16-01020-f007:**
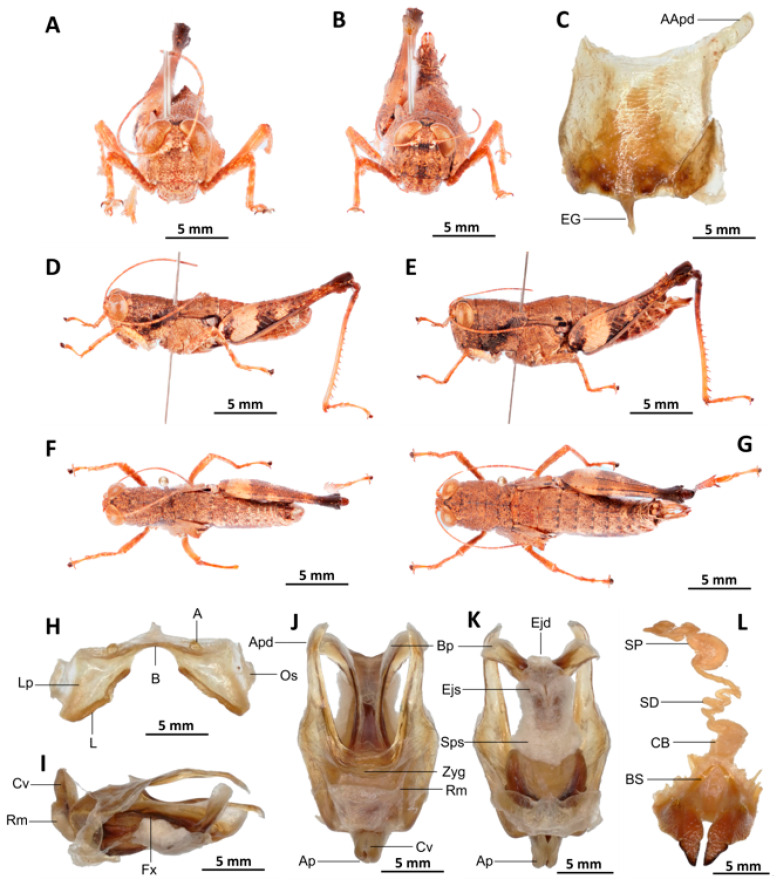
*Serpusia kennei* sp. nov. (**A**) male frontal view; (**B**) female frontal view; (**C**) female subgenital plate; (**D**) male lateral view; (**E**) female lateral view; (**F**) male dorsal view; (**G**) female dorsal view; (**H**) epiphallus dorsal view; (**I**) phallic complex lateral view; (**J**) phallic complex dorsal view; (**K**) phallic complex ventral view; (**L**) female spermatheca. A: ancorae; AApd: anterior apodeme; Ap: apical valves of penis; Apd: apodeme of cingulum; B: bridge of epiphallus; Bp: basal valves of penis; BS: basivalvare sclerites; CB: corpulatory bursa; Cv: valve of cingulum; EG: egg-guide; Ejd: ejaculatory duct; Ejs: ejaculatory sac; Fx: flexure of the endophallus; L: lophus of epiphallus; LP: lateral plate of epiphallus; Os: oval sclerite; Rm: ramus of cingulum; SD: spermathecal duct; SP: spermatheca; Sps: spermatophore sac; Zyg: zygoma of cingulum.

**Figure 8 insects-16-01020-f008:**
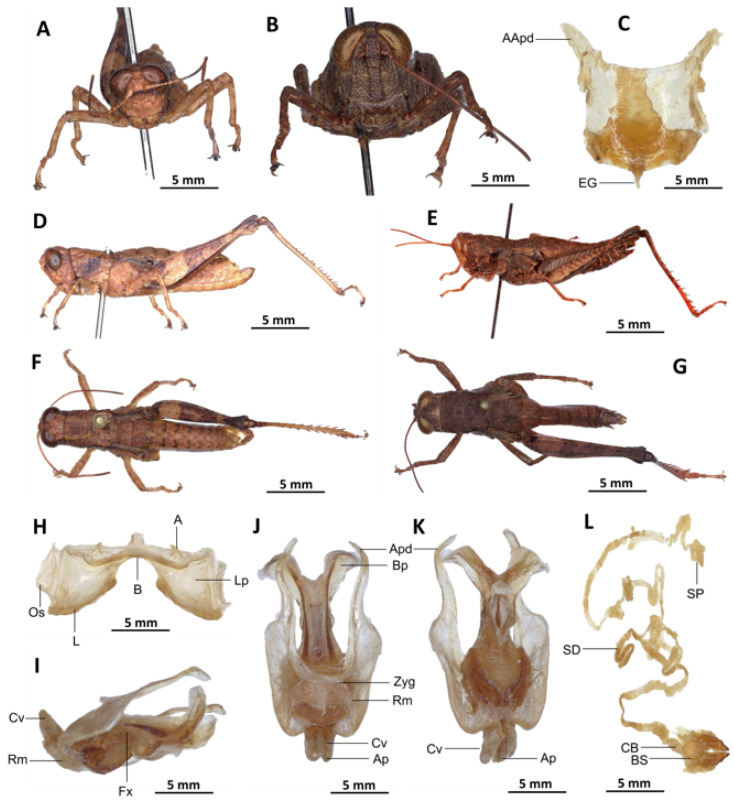
*Serpusia missoupi* sp. nov. (**A**) male frontal view; (**B**) female frontal view; (**C**) female subgenital plate; (**D**) male lateral view; (**E**) female lateral view; (**F**) male dorsal view; (**G**) female dorsal view; (**H**) epiphallus dorsal view; (**I**) phallic complex lateral view; (**J**) phallic complex dorsal view; (**K**) phallic complex ventral view; (**L**) female spermatheca. A: ancorae; AApd: anterior apodeme; Ap: apical valves of penis; Apd: apodeme of cingulum; B: bridge of epiphallus; Bp: basal valves of penis; BS: basivalvare sclerites; CB: corpulatory bursa; Cv valve of cingulum; EG: egg-guide; Ejd: ejaculatory duct; Ejs: ejaculatory sac; Fx: flexure of the endophallus; L: lophus of epiphallus; LP: lateral plate of epiphallus; Os: oval sclerite; Rm: ramus of cingulum; SD: spermathecal duct; SP: spermatheca; Sps: spermatophore sac; Zyg: zygoma of cingulum.

**Figure 10 insects-16-01020-f010:**
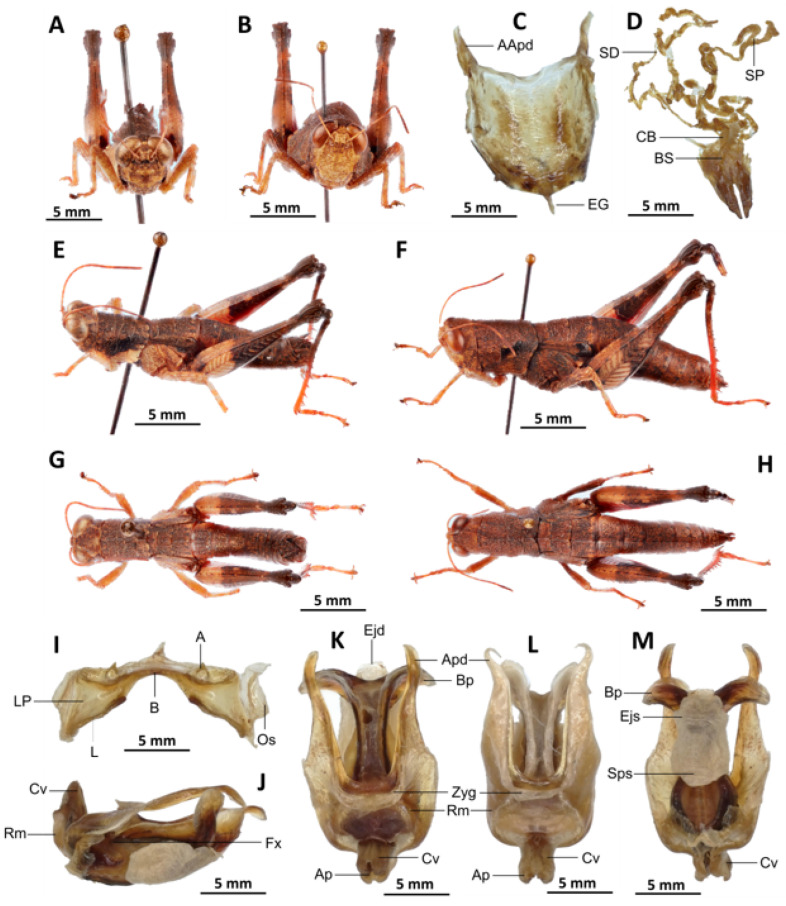
*Serpusia verhaaghi* sp. nov. (**A**) male frontal view; (**B**) female frontal view; (**C**) female subgenital plate; (**D**) female spermatheca; (**E**) male lateral view; (**F**) female lateral view; (**G**) male dorsal view; (**H**) female dorsal view; (**I**) epiphallus dorsal view; (**J**) phallic complex lateral view; (**K**) phallic complex dorsal view; (**L**) Phallic complex dorsal view; (**M**) phallic complex ventral view. A: ancorae; AApd: anterior apodeme; Ap: apical valves of penis; Apd: apodeme of cingulum; B: bridge of epiphallus; Bp: basal valves of penis; BS: basivalvare sclerites; CB: corpulatory bursa; Cv: valve of cingulum; EG: egg-guide; Ejd: ejaculatory duct; Ejs: ejaculatory sac; Fx: flexure of the endophallus; L: lophus of epiphallus; LP: lateral plate of epiphallus; Os: oval sclerite; Rm: ramus of cingulum; SD: spermathecal duct; SP: spermatheca; Sps: spermatophore sac; Zyg: zygoma of cingulum.

**Figure 11 insects-16-01020-f011:**
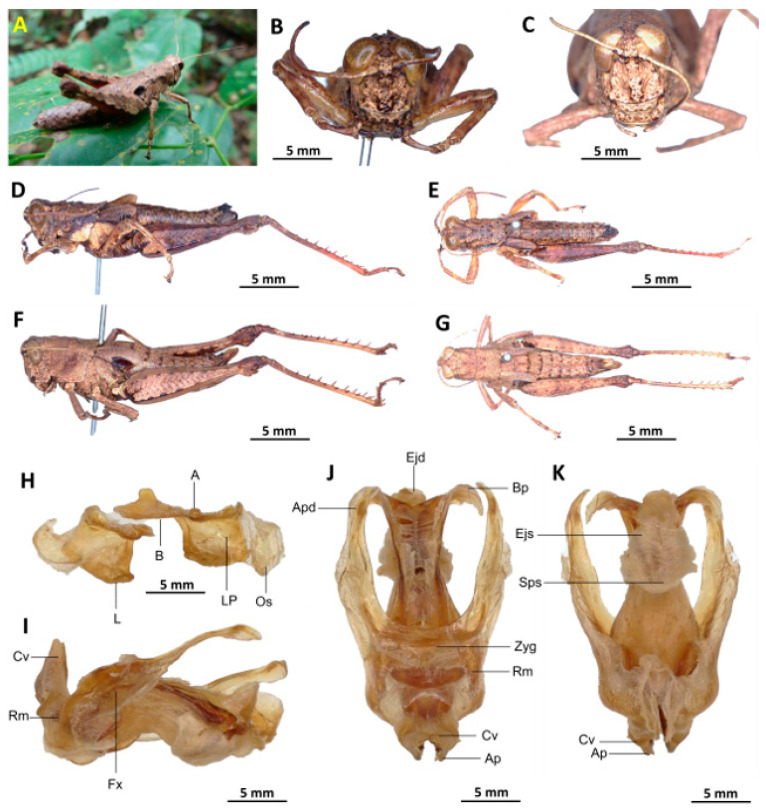
*Serpusia catamita* (**A**) habitus image of a female under natural conditions (photo by Mr. Hannes Öhm); (**B**) male frontal view; (**C**) female frontal view; (**D**) male lateral view; (**E**) male dorsal view; (**F**) female lateral view; (**G**) female dorsal view; (**H**) epiphallus dorsal view; (**I**) phallic complex lateral view; (**J**) phallic complex dorsal view; (**K**) phallic complex ventral view. A: ancorae; AApd: anterior apodeme; Ap: apical valves of penis; Apd: apodeme of cingulum; B: bridge of epiphallus; Bp: basal valves of penis; BS: basivalvare sclerites; CB: corpulatory bursa; Cv: valve of cingulum; EG: egg-guide; Ejd: ejaculatory duct; Ejs: ejaculatory sac; Fx: flexure of the endophallus; L: lophus of epiphallus; LP: lateral plate of epiphallus; Os: oval sclerite; Rm: ramus of cingulum; SD: spermathecal duct; SP: spermatheca; Sps: spermatophore sac; Zyg: zygoma of cingulum.

**Figure 12 insects-16-01020-f012:**
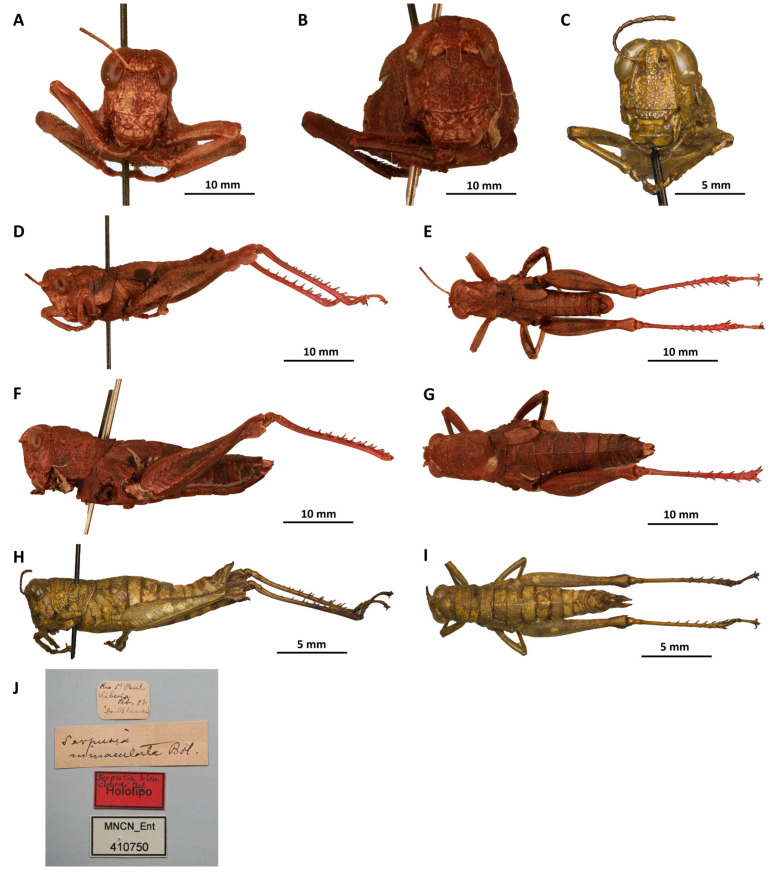
Images of holotypes and allotypes of *Serpusia blanchardi* (MNCN_ENT410759) and *Serpusia inflata* (NHMUK_02159839). (**A**) head of *S. inflata* in frontal view (holotype ♂); (**B**) head of *S. inflata* in frontal view (allotype ♀); (**C**) head of *S. blanchardi* in frontal view (holotype ♀); (**D**) *S. inflata* in lateral view (holotype ♂); (**E**) *S. inflata* in dorsal view (holotype ♂); (**F**) *S. inflata in lateral* view (allotype ♀); (**G**) *S. inflata* in dorsal view (allotype ♀); (**H**) *S. blanchardi* in lateral view (holotype ♀); (**I**) *S. blanchardi* in dorsal view (holotype ♀); (**J**) Label of *S. blanchardi* (holotype ♀).

**Figure 13 insects-16-01020-f013:**
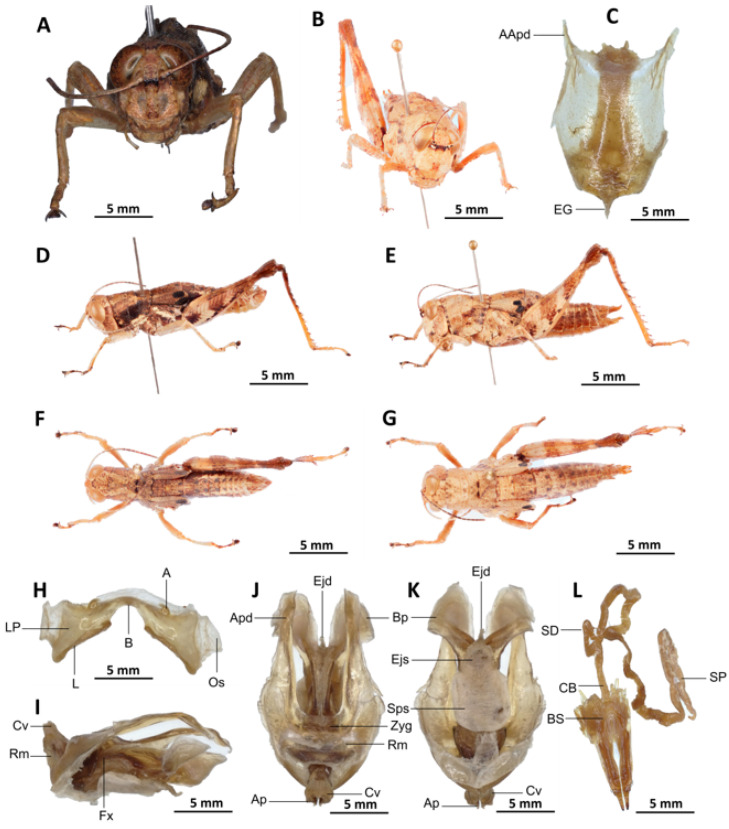
*Paraserpusia succursor*. (**A**) male frontal view; (**B**) female frontal view; (**C**) female subgenital plate; (**D**) male lateral view; (**E**) female lateral view; (**F**) male dorsal view; (**G**) female dorsal view; (**H**) epiphallus dorsal view; (**I**) phallic complex lateral view; (**J**) phallic complex dorsal view; (**K**) phallic complex ventral view; (**L**) female spermatheca. A: ancorae; AApd: anterior apodeme; Ap: apical valves of penis; Apd: apodeme of cingulum; B: bridge of epiphallus; Bp: basal valves of penis; BS: basivalvare sclerites; CB: corpulatory bursa; Cv: valve of cingulum; EG: egg-guide; Ejd: ejaculatory duct; Ejs: ejaculatory sac; Fx: flexure of the endophallus; L: lophus of epiphallus; LP: lateral plate of epiphallus; Os: oval sclerite; Rm: ramus of cingulum; SD: spermathecal duct; SP: spermatheca; Sps: spermatophore sac; Zyg: zygoma of cingulum.

**Figure 14 insects-16-01020-f014:**
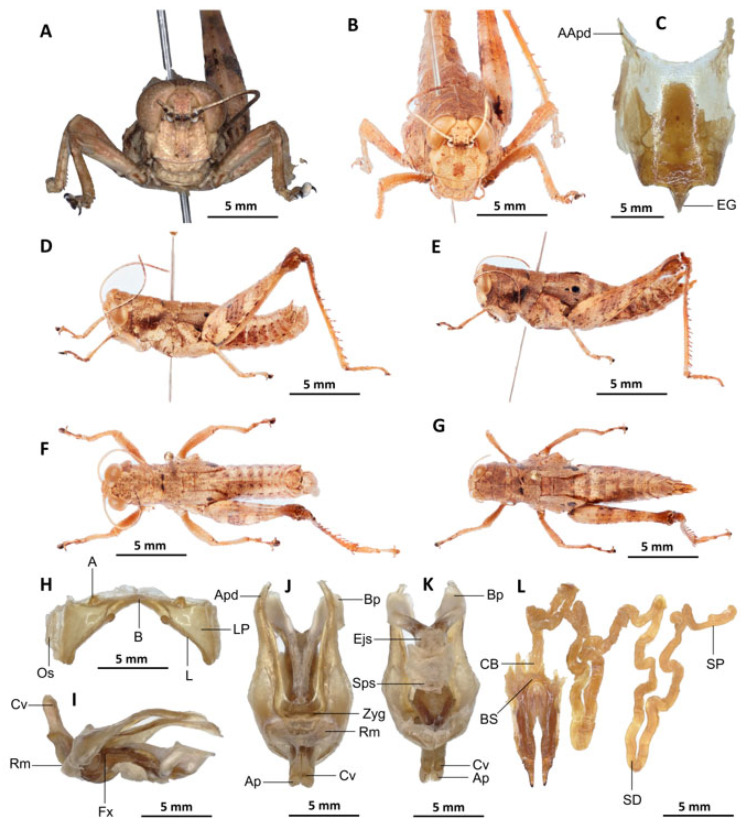
*Paraserpusia hoeferi* sp. nov. (**A**) male frontal view; (**B**) female frontal view; (**C**) female subgenital plate; (**D**) male lateral view; (**E**) female lateral view; (**F**) male dorsal view; (**G**) female dorsal view; (**H**) epiphallus dorsal view; (**I**) phallic complex lateral view; (**J**) phallic complex dorsal view; (**K**) phallic complex ventral view; (**L**) female spermatheca. A: ancorae; AApd: anterior apodeme; Ap: apical valves of penis; Apd: apodeme of cingulum; B: bridge of epiphallus; Bp: basal valves of penis; BS: basivalvare sclerites; CB: corpulatory bursa; Cv: valve of cingulum; EG: egg-guide; Ejd: ejaculatory duct; Ejs: ejaculatory sac; Fx: flexure of the endophallus; L: lophus of epiphallus; LP: lateral plate of epiphallus; Os: oval sclerite; Rm: ramus of cingulum; SD: spermathecal duct; SP: spermatheca; Sps: spermatophore sac; Zyg: zygoma of cingulum.

**Figure 15 insects-16-01020-f015:**
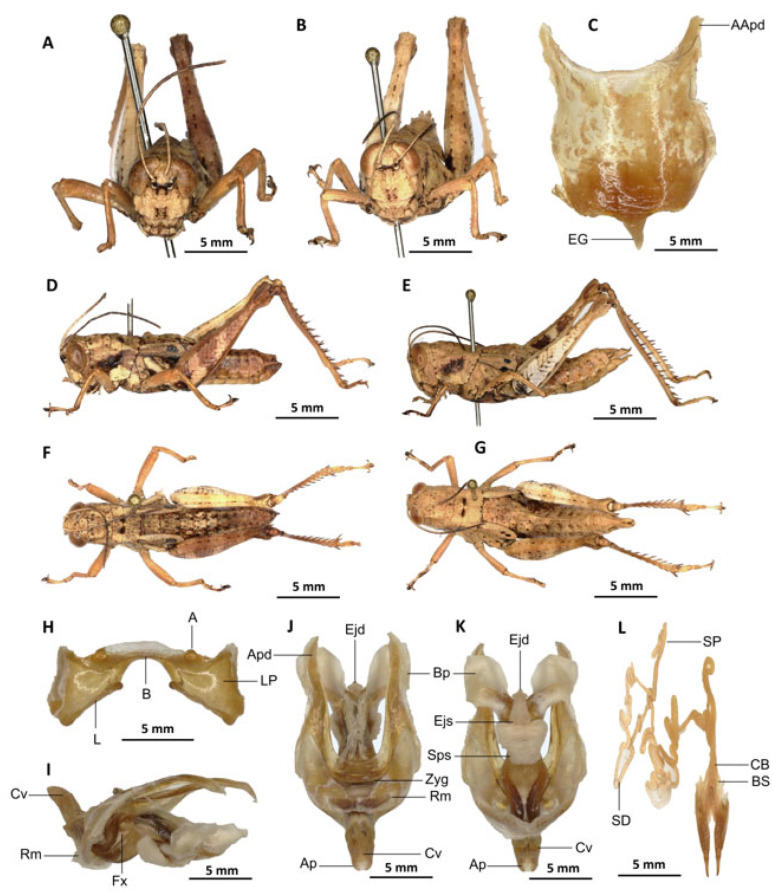
*Paraserpusia husemanni* sp. nov. (**A**) male frontal view; (**B**) female frontal view; (**C**) female subgenital plate; (**D**) male lateral view; (**E**) female lateral view; (**F**) male dorsal view; (**G**) female dorsal view; (**H**) epiphallus dorsal view; (**I**) phallic complex lateral view; (**J**) phallic complex dorsal view; (**K**) phallic complex ventral view; (**L**) female spermatheca. A: ancorae; AApd: anterior apodeme; Ap: apical valves of penis; Apd: apodeme of cingulum; B: bridge of epiphallus; Bp: basal valves of penis; BS: basivalvare sclerites; CB: corpulatory bursa; Cv: valve of cingulum; EG: egg-guide; Ejd: ejaculatory duct; Ejs: ejaculatory sac; Fx: flexure of the endophallus; L: lophus of epiphallus; LP: lateral plate of epiphallus; Os: oval sclerite; Rm: ramus of cingulum; SD: spermathecal duct; SP: spermatheca; Sps: spermatophore sac; Zyg: zygoma of cingulum.

**Figure 16 insects-16-01020-f016:**
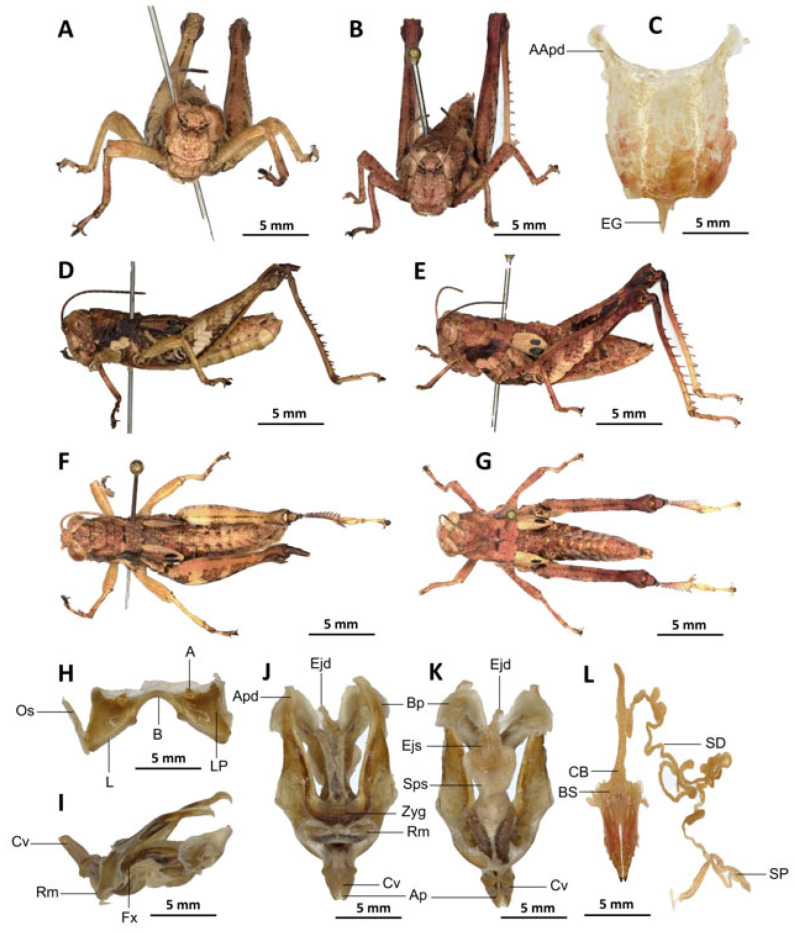
*Paraserpusia kekeunoui* sp. nov. (**A**) male frontal view; (**B**) female frontal view; (**C**) female subgenital plate; (**D**) male lateral view; (**E**) female lateral view; (**F**) male dorsal view; (**G**) female dorsal view; (**H**) epiphallus dorsal view; (**I**) phallic complex lateral view; (**J**) phallic complex dorsal view; (**K**) phallic complex ventral view; (**L**) female spermatheca. A: ancorae; AApd: anterior apodeme; Ap: apical valves of penis; Apd: apodeme of cingulum; B: bridge of epiphallus; Bp: basal valves of penis; BS: basivalvare sclerites; CB: corpulatory bursa; Cv: valve of cingulum; EG: egg-guide; Ejd: ejaculatory duct; Ejs: ejaculatory sac; Fx: flexure of the endophallus; L: lophus of epiphallus; LP: lateral plate of epiphallus; Os: oval sclerite; Rm: ramus of cingulum; SD: spermathecal duct; SP: spermatheca; Sps: spermatophore sac; Zyg: zygoma of cingulum.

**Table 1 insects-16-01020-t001:** List of specimens used, and GenBank accession numbers for the sequenced markers.

Species	Voucher Codes	Coll. Date	Country of Origin	GenBank Accession Number		Authors
				COI-5P	16S rDNA	
*P. carnapi*	CM1361	12 June 2022	Cameroon	PP707812	PP708833	[14]
*P. carnapi*	CM1363	12 June 2022	Cameroon	PP707813	PP708802	[14]
*P. carnapi*	CM1364	12 June 2022	Cameroon	PP700651	PP708803	[14]
*P. kennei*	CM1135	10 April 2022	Cameroon	PP707818	NA	[14]
*P. kennei*	CM1136	10 April 2022	Cameroon	PP700657	NA	[14]
*P. kennei*	CM1143	10 April 2022	Cameroon	NA	PP708813	[14]
*A. morogorica*	TZ1441	1 July 2020	Tanzania	PV904790	PV835415	This study
*A. morogorica*	TZ1444	1 April 2019	Tanzania	PV819833	PV835416	This study
*A. morogorica*	TZ1445	1 March 2019	Tanzania	PV819834	PV835417	This study
*A. nguruensis*	TZ1446	1 June 2017	Tanzania	PV808098	PV835418	This study
*A. subnuda*	TZ1452	1 August 2019	Tanzania	PV819835	PV835419	This study
*A. subnuda*	TZ1454	15 February 2017	Tanzania	PV819836	PV835420	This study
*A. subnuda*	TZ1456	15 February 2023	Tanzania	PV819837	PV835421	This study
*A. subnuda*	TZ1457	15 February 2023	Tanzania	PV819838	PV835422	This study
*S. nitidula*	CMW98	20 July 2016	Cameroon	PV819839	PV835588	This study
*S. verhaaghi* sp. nov.	CMJ574	2 March 2021	Cameroon	PV808099	PV835589	This study
*S. verhaaghi* sp. nov.	CMJ575	2 March 2021	Cameroon	PV819844	PV835424	This study
*S. verhaaghi* sp. nov.	CMJ576	2 March 2021	Cameroon	PV808100	NA	This study
*S. verhaaghi* sp. nov.	CMJ533	12 June 2020	Cameroon	PV808101	PV835423	This study
*S. verhaaghi* sp. nov.	CMJ213	7 January 2022	Cameroon	PV808102	NA	This study
*S. verhaaghi* sp. nov.	CMJ1420	27 March 2017	Cameroon	PV904789	PV835425	This study
*S. opacula*	CMJ726	3 May 2020	Cameroon	PV808103	PV835430	This study
*S. opacula*	CMJ727	3 May 2020	Cameroon	PV808104	PV835431	This study
*S. opacula*	CMJ520	11 October 2020	Cameroon	PV808105	NA	This study
*S. kennei* sp. nov.	CMJ708	18 April 2022	Cameroon	PV808106	PV835427	This study
*S. kennei* sp. nov.	CMJ709	18 April 2022	Cameroon	PV808107	PV835590	This study
*S. kennei* sp. nov.	CMJ711	18 April 2022	Cameroon	PV808108	PV835429	This study
*S. kennei* sp. npv.	CMJ699	17 April 2022	Cameroon	PV819840	PV835426	This study
*S. kennei* sp. nov.	CMJ710	18 April 2022	Cameroon	PV819841	PV835428	This study
*S. missoupi* sp. nov.	CMJ1418	16 September 2018	Cameroon	PV808109	NA	This study
*S. missoupi* sp. nov.	CMJ1421	28 February 2017	Cameroon	PV819842	PV835434	This study
*S. seinoi* sp. nov.	CMJ1110	10 April 2022	Cameroon	PV819843	PV835432	This study
*S. seinoi* sp. nov.	CMJ1111	10 April 2022	Cameroon	PV904788	PV835433	This study
*P. succursor*	CMJ150	20 March 2022	Cameroon	PV808118	PV835435	This study
*P. succursor*	CMJ151	20 March 2022	Cameroon	PV819857	PV835436	This study
*P. succursor*	CMJ152	20 March 2022	Cameroon	PV819858	PV835437	This study
*P. succursor*	CMJ387	5 December 2021	Cameroon	PV819859	PV835438	This study
*P. hoeferi* sp. nov.	CMJ211	7 January 2022	Cameroon	PV808110	PV835439	This study
*P. hoeferi* sp. nov.	CMJ212	7 January 2022	Cameroon	PV819851	PV835440	This study
*P. hoeferi* sp. nov.	CMJ214	7 January 2022	Cameroon	PV808111	PV835441	This study
*P. hoeferi* sp. nov.	CMJ215	7 January 2022	Cameroon	PV808112	PV835591	This study
*P. husemanni* sp. nov.	CMW67	3 July 2024	Cameroon	PV819845	PV835444	This study
*P. husemanni* sp. nov.	CMW68	3 July 2024	Cameroon	PV819846	PV835445	This study
*P. husemanni* sp. nov.	CMW69	3 July 2024	Cameroon	PV808119	PV835446	This study
*P. husemanni* sp. nov.	CMW70	3 July 2024	Cameroon	PV819847	PV835447	This study
*P. husemanni* sp. nov.	CMW71	3 July 2024	Cameroon	PV819848	PV835448	This study
*P. husemanni* sp. nov.	CMW72	3 July 2024	Cameroon	PV808120	PV835449	This study
*P. husemanni* sp. nov.	CMW73	3 July 2024	Cameroon	PV819849	PV835450	This study
*P. husemanni* sp. nov.	CMW74	3 July 2024	Cameroon	PV808121	PV835451	This study
*P. husemanni* sp. nov.	CMW75	3 July 2024	Cameroon	PV808122	PV835452	This study
*P. husemanni* sp. nov.	CMW76	3 July 2024	Cameroon	PV819850	PV835453	This study
*P. kekeunoui* sp. nov.	CMJ751	18 August 2021	Cameroon	PV819852	PV835454	This study
*P. kekeunoui* sp. nov.	CMW78	3 August 2024	Cameroon	PV808123	PV835455	This study
*P. kekeunoui* sp. nov.	CMW81	3 August 2024	Cameroon	PV819853	PV835456	This study
*P. kekeunoui* sp. nov.	CMW82	3 August 2024	Cameroon	PV808124	PV835457	This study
*P. kekeunoui* sp. nov.	CMW83	3 August 2024	Cameroon	PV819854	PV835458	This study
*P. kekeunoui* sp. nov.	CMW84	3 August 2024	Cameroon	PV819855	PV835459	This study
*P. kekeunoui* sp. nov.	CMW85	3 August 2024	Cameroon	PV808125	PV835460	This study
*P. kekeunoui* sp. nov.	CMW86	3 August 2024	Cameroon	PV819856	PV835461	This study
*P. tamessei* sp. nov.	CMW171	29 December 2024	Cameroon	PV808127	PV835463	This study
*P. tamessei* sp. nov.	CMW172	29 December 2024	Cameroon	PV819860	PV835464	This study
*P. tamessei* sp. nov.	CMW173	29 December 2024	Cameroon	PV808128	PV835465	This study
*P. tamessei* sp. nov.	CMW174	29 December 2024	Cameroon	PV808129	PV835595	This study
*P. tamessei* sp. nov.	CMW175	29 December 2024	Cameroon	PV808130	PV835596	This study
*P. tamessei* sp. nov.	CMW176	29 December 2024	Cameroon	PV808131	PV835597	This study
*P. tamessei* sp. nov.	CMW177	29 December 2024	Cameroon	PV819861	PV835598	This study
*P. tamessei* sp. nov.	CMW178	29 December 2024	Cameroon	PV808132	PV835599	This study
*P. tamessei* sp. nov.	CMW184	10 January 2025	Cameroon	PV808133	PV835600	This study
*P. tamessei* sp. nov.	CMW185	10 January 2025	Cameroon	PV808134	PV835466	This study
*P. tamessei* sp. nov.	CMW186	10 January 2025	Cameroon	PV808135	PV835467	This study
*P. tamessei* sp. nov.	CMW187	10 January 2025	Cameroon	PV808136	PV835601	This study
*P. tamessei* sp. nov.	CMW189	10 January 2025	Cameroon	PV808137	PV835468	This study
*P. tamessei* sp. nov.	CMW190	10 January 2025	Cameroon	PV808138	PV835469	This study
*P. tamessei* sp. nov.	CMW191	10 January 2025	Cameroon	PV808139	NA	This study
*P. tamessei* sp. nov.	CMW192	25 January 2025	Cameroon	PV808140	PV835470	This study
*P. tamessei* sp. nov.	CMW193	25 January 2025	Cameroon	PV808141	PV835471	This study
*P. tamessei* sp. nov.	CMW221	25 January 2025	Cameroon	PV808142	PV835472	This study
*P. tamessei* sp. nov.	CMW222	25 January 2025	Cameroon	PV808143	PV835473	This study
*P. tindoi* sp. nov.	CMJ1115	10 April 2022	Cameroon	PV808113	PV835594	This study
*P. tindoi* sp. nov.	CMJ1117	10 April 2022	Cameroon	PV808114	PV835442	This study
*P. tindoi* sp. nov.	CMJ1120	10 April 2022	Cameroon	PV808115	PV835443	This study
*P. tindoi* sp. nov.	CMJ1121	10 April 2022	Cameroon	PV808116	PV835592	This study
*P. tindoi* sp. nov.	CMJ1122	10 April 2022	Cameroon	PV808117	PV835593	This study

Coll. date: collection date.

**Table 3 insects-16-01020-t003:** The distribution of sequence divergence at each taxonomic level.

Label	n	Taxa	Comparisons	Min Dist. (%)	Mean Dist. (%)	Max Dist. (%)	SE Dist. (%)
Within Species	79	5	1440	0.00	3.57	9.04	0.00
Within Genus	77	2	937	3.51	10.57	14.26	0.00
Within Family	81	1	863	7.03	11.28	16.67	0.00

Min Dist.: Minimum Genetic Distance; Max Dist.: Maximum Genetic Distance; Mean Dist.: Mean Genetic Distance; SE Dist.: Standard Error.

**Table 4 insects-16-01020-t004:** Comparison of the mean and maximum intra-specific genetic distances with the nearest neighbour (NN). Where the species is a singleton, N/A is displayed for intra-specific values.

Species	Mean Intraspecific Distance	Max Intraspecific Distance	Nearest Neighbour Species	Distance to NN
*S. nitidula*	N/A	N/A	*S. succursor*	13.65
*P. carnapi*	0.54	0.81	*S. succursor*	14.07
*A. morogorica*	2.25	5.50	*Ar.subnuda*	3.62
*A. subnuda*	1.68	2.44	*A. morogorica*	3.62
*A. nguruensis*	N/A	N/A	*A. morogorica*	3.80
*S. succursor*	3.74	9.71	*A. morogorica*	7.40
*S. opacula*	3.73	6.71	*S. succursor*	9.13

**Table 5 insects-16-01020-t005:** BIN records of the morphologically identified species. N/A is displayed for intra-specific values where a species is a singleton.

Morphologically Identified Species	Mean Intraspecific Distance (%)	Max Intraspecific Distance (%)	BIN (Count)	Distance to NN
*S. succursor*	0.96	1.77	BOLD: AGK3115 (5)	6.09
*S. succursor*	1.02	1.77	BOLD: AGK3116 (15)	2.02
*S. succursor*	0.51	1.13	BOLD: AGK3117 (13)	2.02
*S. succursor*	0.28	0.65	BOLD: AGK3118 (7)	3.21
*S. succursor*	0.09	0.32	BOLD: AGO5270 (7)	1.01
*S. succursor*	0.16	0.16	BOLD: AGO5271 (2)	1.01
*S. succursor*	N/A	N/A	BOLD: AGO5272 (1)	1.35
*S. opacula*	0.95	1.36	BOLD: AGK3119 (5)	2.20
*S. opacula*	1.10	1.10	BOLD: AGK3120 (2)	2.20
*S. opacula*	0.43	0.48	BOLD: AGK3121 (3)	2.88
*S. opacula*	0.13	0.33	BOLD: AGK3122 (5)	2.88
*S. opacula*	0.51	0.51	BOLD: AGK3123 (2)	3.20
*S. nitidula*	N/A	N/A	BOLD: AGK3157 (1)	10.40
*P. kennei*	0.33	0.50	BOLD: AGK4105 (3)	8.16
*P. carnapi*	0.11	0.17	BOLD: AGK4107 (3)	8.16
*A. nguruensis*	N/A	N/A	BOLD: AGN6867 (1)	3.69
*A. morogorica*	0.85	0.85	BOLD: AGN6868 (2)	3.69
*A.subnuda*	0.17	0.17	BOLD: AGO1052 (2)	2.18
*A. subnuda*	0.67	0.67	BOLD: AGO1053 (2)	2.18

**Table 6 insects-16-01020-t006:** Comparisons of the mean and maximum intra-specific values with the nearest neighbour distance (NN) for each MOTU species. Where the species is a singleton, N/A is displayed for intra-specific values.

MOTU	Morphological Species	Sample ID	Species Delimited by BI/ML	Mean Intra-Sp	Max Intra-Sp	Distance to NN
1	*S. nitidula*	CMW98	*S. nitidula*	N/A	0	11.62
2	*P. carnapi*	CMJ1361, CMJ1363, CMJ1364	*P. carnapi*	0.53	0.80	8.16
3	*P. kennei*	CMJ1135, CMJ1136, CMJ1143	*P. kennei*	0.33	0.50	8.16
4	*A. morogorica*	TZ1441, TZ144, TZ1445	*A. morogorica*	2.20	5.30	3.51
5	*A. nguruensis*	TZ1446	*A.nguruensis*	N/A	0	3.67
6	*A.subnuda*	TZ1452, TZ1454	*A. subnuda*	0.80	0.80	2.07
7	*A. subnuda*	TZ1456, TZ1457	*A. subnuda*	0.16	0.16	2.07
8	*S. opacula*	CMJ520, CMJ726, CMJ727	*S. opacula*	1.12	1.53	2.92
9	*S. opacula*	CMJ699, CMJ708, CMJ709, CMJ710, CMJ711	*S. kennei* sp. nov.	0.44	1.11	2.92
10	*S. opacula*	CMJ1110, CMJ1111	*S. seinoi* sp. nov.	0.65	0.65	3.26
11	*S. opacula*	CMJ1418, CMJ1421	*S. missoupi* sp. nov.	1.04	1.04	2.09
12	*S. opacula*	CMJ533, CMJ213, CMJ574, CMJ575, CMJ576, CMJ1420	*S. verhaaghi* sp. nov.	1.71	3.41	2.09
13	*S. succursor*	CMJ1115, CMJ1117, CMJ1120, CMJ1121, CMJ1122	*P. tindoi* sp. nov.	1.04	1.99	6.12
14	*S. succursor*	CMJ751, CMW78, CMW81, CMW82, CMW83, CMW84, CMW85, CMW86	*P. kekeunoui* sp. nov.	0.29	0.65	3.21
15	*S. succursor*	CMJ150, CMJ151, CMJ152, CMJ387, CMW184, CMW185, CMW186,CMW187, CMW189, CMW190, CMW191, CMW19	*-P. succursor* *-P. tamessei* sp.nov.	1.04	1.68	1.92
16	*S. succursor*	CMZW67, CMW69, CMW70, CMW72, CMW74, CMW75, CMW75	*P. husemanni sp. nov*.	0.09	0.32	1.01
17	*S. succursor*	CMJ211, CMJ212, CMJ214, CMJ215, CMW170, CMW171, CMW172, CMW173, CMW174, CMW175, CMW176, CMW177, CMW178	*-P. höferi* sp. nov. *-P. tamessei* sp. nov.	0.65	1.53	1.92
18	*S. succursor*	CMW73	*P. husemanni* sp. nov.	N/A	0	1.35
19	*S. succursor*	CMW68, CMW71	*P. husemanni* sp. nov.	0.16	0.16	1.01

## Data Availability

All data will be available after a reasonable request to the corresponding authors.

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
