# Peer review of "Integrative Taxonomy Revealed Cryptic Diversity in the West African Grasshopper Genus Serpusia Karsch, 1891 (Orthoptera: Catantopinae)"

_insects, 2025, doi:10.3390/insects16101020_

Round 1
Reviewer 1 Report
Comments and Suggestions for Authors
The manuscript presents an integrative taxonomic revision of Serpusia, combining molecular data, species delimitation, phylogenetic reconstruction, and morphology. It addresses an important gap in African Orthoptera knowledge and introduces new taxa, including a new genus (Paraserpusia). As such, it presents valuable new data and significant taxonomic contributions, however, I do have some major concerns that should be considered for publication:
Diagnosis of Serpusia: Due to the changes from the original concept of the genus, a new and updated diagnosis of Serpusia must be provided.
Justification for taxonomic changes: The authors suggest that S. blanchardi, S. catamita, and S. inflata should be excluded from Serpusia and placed in a separate genus. Although they present this as a tentative hypothesis pending further study, no clear morphological evidence is provided to support such a taxonomic action. Moreover, in the redescription of S. catamita Karsch, 1893, the authors compare it with the “closely similar” S. inflata and S. opacula, which makes it unclear why S. catamita would need to be transferred to a new genus. In addition, the description of the phallic complex explicitly states that “its internal genitalia are similar to those of Serpusia and Aresceutica,” which further weakens the rationale for excluding it from Serpusia. I recommend that the authors revise this conceptual statement to clarify or justify the proposed taxonomic change.
Type species: It would be important to explicitly mention that the type species of the genus is Serpusia opacula Karsch, 1891.
Morphological descriptions: Some morphological descriptions are vague or ambiguous. These need to be revised for precision and consistency, see below.
Phallic complex: Since the phallic complex appears not to have been clarified in ammonia, it is difficult to distinguish the different sclerites. I strongly recommend adding clear indications to the different structures in the figures to facilitate interpretation and comparison across species.
Comparative plates: Comparative plates should ideally present the same set of structures across species (e.g., habitus, phallic complex) to allow more effective visualization and comparison.
Discussion: The discussion is very detailed, but it currently addresses too many different issues without a clear structure, which makes it difficult to follow. I suggest reorganizing it into thematic sections (e.g., efficacy of DNA barcoding, taxonomic implications, phylogeographic patterns, broader significance). In particular, the distinction between Serpusia and Aresceutica could be clarified, as the variability reported within Serpusia seems to blur the diagnostic features just described. Similarly, some descriptive information (lists of BINs, localities, climate descriptions) might be better condensed, leaving more space for interpretation and synthesis.
Following some other comments that should be addressed:
Line 24: Use the standard terminology “phallic complex” instead of “male phalli.”
Line 61: The phrase “[5] discussed the close …” is difficult to follow. Provide the full citation (e.g., “Rowell et al. (5) discussed …”) rather than only the bracketed reference.
Line 64: “[6] Synonymised Ptemoblax insidiosus Bolívar”, same comment as above: provide full citation for clarity when the name of the author is supposed to be read. When the sentence begins by referring to the author as in this case, the author’s name should be written out rather than using the citation number [6], so that it reads clearly for the reader.
Lines 69–70: Rowell et al. (2018) did not formally transfer S. catamita, S. blanchardi, or S. inflata. Thus, the statement that “the genus Serpusia now contains only two species: S. opacula and S. succusor” is wrong. Currently, it contains five valid species. Introduction and discussion should be clarified to reflect this matter.
Line 90: Revise “A total of hundreds of specimens” may be better to say “Hundreds of specimens” or “Several hundred specimens.”
Line 108: Please describe all measurements clearly and consistently, as you did for hind femur width. For example, specify body length, femur width, and any other structures measured, including how each measurement was taken.
Table 1: the measurement are described in text, so there is no need to repeat them in Table 1.
Line 163: “69 ingroup taxa” these are specimens, not taxa. Clarify how many taxa are represented by the specimens included in the sequences.
Table 2: Please include locality information for each specimen, and ideally also provide georeferences, in addition to the collection dates.
Line 258: The sentence “This section may be divided by subheadings…” appears to be leftover template text (likely from the journal guidelines) rather than part of the manuscript. Please remove it.
Line 277: The phrase “potential species” is unclear. Clarify whether it refers to undescribed species or putative lineages.
Line 280: Use neutral wording for phylogenetic support: “shows” or “supports the recognition of” rather than “demonstrates.”
Line 334: “Two recognised species and six hypothetical species” , use “putative” or “undescribed species” instead of “hypothetical.” In taxonomy, every species is essentially a hypothesis of an evolutionary lineage. I recommend replacing “hypothetical species” with a more precise term such as “putative species” or “undescribed species” to accurately reflect that these six lineages are inferred from your analyses but not yet formally described.
Line 368: Use consistent terminology: “specimens” instead of “individuals.”
Line 416: The wording “to hypothetical species… on the curated tree” is unclear. Revise for clarity; what do you mean by curated tree?
Line 483: Diagnosis: S. opacula bears resemblance to Paraserpusia succursor, but integrative delimitation procedure is not clear. Add details on morphological vs. molecular separation.
Line 495 / 528: “Ectophallic sheath elongated (vs. short in P. succursor)” – elongated is not precisely the opposite of short. Clarify or quantify.
Line 581: “Cingular valves, aedeagus, and gonopore sac” – not visible in photos; . Maybe and indication to the structures in the plates would help the reader to identify the structures.
Line 676: Statements like “identical to S. opacula” contradict differences listed. Revise wording for consistency.
Line 689: “The head is straight” – specify view (dorsal, lateral, frontal).
Line 922: Clarify whether original descriptions of S. blanchardi and S. inflata are transcribed or summarized.
Line 1079. “apodemes of cingular” should read apodemes of cingulum
Line1600: In the key, distinctions like “fairly reaching” vs. “barely reaching” are subtle and subjective. Revise for clarity and reproducibility.
Line 1662. “Therefore, we removed Serpusia succursor from the genus Serpusia and included it in a new genus that we erected, Paraserpusia gen. nov.” I ssume that the authors are referring to the Serpusia succursor species complex
Line 1730. The use of “signalisation” is not clear in English. I suggest revising the sentence for clarity. For example: “Molecular analyses revealed that the identification by [44] was in fact a misidentification, corresponding to the hidden species Paraserpusia kekeunoui sp. nov. and Paraserpusia tamesei sp. nov., respectively.”
Line 1739. “However, [44] reported Serpusia succursor (now Paraserpusia succursor) in Buea, Kumba, and Tombel in the Southwest region of Cameroon. The species reported by these authors was Serpusia opacula rather than 'Serpusia succursor'. It is not clear why the authors are confident that the record by [44] represents a misidentification. Please clarify what evidence supports this conclusion (e.g., re-examination of specimens, molecular data, or comparison with diagnostic characters). Without specifying the basis, the statement remains speculative.
Line 1756. This paragraph is somewhat confusing. At first, it seems intended to establish clear morphological differences between Serpusia and Aresceutica, but then it shifts to emphasize variability within Serpusia (in the cingular valves and aedeagus), which blurs the distinction. I suggest clarifying whether the main point is (i) to highlight the diagnostic characters that separate the two genera, or (ii) to discuss morphological variability within Serpusia.
Author Response
Dear Reviewer, thank you very much for your valuable comments on our manuscript. Below are the responses to the comments.
Response to Reviewer 1.
Remarks: All the comments of Reviewer 1 are highlighted in yellow in the revised manuscript.
Comment 1: Diagnosis of Serpusia: Due to the changes from the original concept of the genus, a new and updated diagnosis of Serpusia must be provided.
Response 1: Thank you for pointing this out. The authors have provided a diagnosis of Serpusia first in the section when differentiating the true Serpusia from S. blanchardi, S. catamita, and S. inflata; second in the taxonomic implication of the discussion section, when differentiating the true Serpusia from Paraserpusia. Providing a diagnosis of Serpusia in a separate paragraph may seem redundant. This diagnosis can be found on page 38, Lines 1028-1048, then on page 63, paragraph 3, Lines 1805-1814 of the revised manuscript.
Comment 2: Justification for taxonomic changes: The authors suggest that S. blanchardi, S. catamita, and S. inflata should be excluded from Serpusia and placed in a separate genus. Although they present this as a tentative hypothesis pending further study, no clear morphological evidence is provided to support such a taxonomic action. Moreover, in the redescription of S. catamita Karsch, 1893, the authors compare it with the “closely similar” S. inflata and S. opacula, which makes it unclear why S. catamita would need to be transferred to a new genus. In addition, the description of the phallic complex explicitly states that “its internal genitalia are similar to those of Serpusia and Aresceutica,” which further weakens the rationale for excluding it from Serpusia. I recommend that the authors revise this conceptual statement to clarify or justify the proposed taxonomic change.
Response 2: The comments are very interesting. However, on page 38, Lines 1028-1048 of the revised manuscript, in the remarks under ´´the other species´´ section, the authors have provided consistent morphological differences between the true Serpusia s. str. and S. blanchardi, S. catamita, and S. inflata, thus justifying why these species should be removed from Serpusia. For instance, (1) the true Serpusia has vestigial or rudimentary forewings, whereas the so-called S. catamita and S. inflata have large and lobiform forewings; (2) the forewings in the true Serpusia have two shiny black spots, whereas there is always only one in S. catamita and S. inflata; (3) the posterior margin of the pronotum is distinctly incised and strongly emarginated in the middle, forming an acute angle in Serpusia. In contrast, it is rounded and distinctly wavy, with a light brown mottling in the middle in S. catamita, and more or less smooth and flat in S. inflata; (4) the ancorae of the epiphallus point directly inwards in Serpusia, whereas they are slanted forwards in S. catamita; (5) the lophi of the epiphallus are large, lobiform, and rounded in Serpusia, whereas they are curved upwards, with angular apices in S. catamita; (6) the phallic complex is robust and strongly sclerotised in S. catamita than in Serpusia; (7) the ejaculatory sac is sac is large, roughly covering the two-thirds of the endophallus and well covering the endophallic sclerites ventrally in Serpusia, whereas it is strongly reduced in S. catamita; and (8) the endophallic valves are flattened basally in Serpusia, whereas they are narrow basally in S. catamita. In addition, the tegmina in S. blanchardi are much reduced and not spotted as in true Serpusia; the hind femora are longer basally and less enlarged than in Serpusia; the antennae are shorter than the head and pronotum together in S. blanchardi, whereas they are longer than the head and pronotum together in Serpusia.
Comment 3: Type species: It would be important to explicitly mention that the type species of the genus is Serpusia opacula Karsch, 1891.
Response 3: We agree with this comment. Therefore, the suggestion has been incorporated accordingly, on page 19, Line 443 of the revised manuscript.
Comment 4: Morphological descriptions: Some morphological descriptions are vague or ambiguous. These need to be revised for precision and consistency, see below.
Phallic complex: Since the phallic complex appears not to have been clarified in ammonia, it is difficult to distinguish the different sclerites. I strongly recommend adding clear indications to the different structures in the figures to facilitate interpretation and comparison across species.
Response 4: Thank you for pointing this out. The different structures of the internal genitalia of species have been indicated in all relevant plates, and the explanations of the structures are given and highlighted in the figure titles. Corrections can be found on figures 6-11, 13-18, on pages 20, 24, 27, 29, 32, 36, 40, 43, 46, 49, 52, and 55 of the revised manuscript.
Comment 5: Comparative plates: Comparative plates should ideally present the same set of structures across species (e.g., habitus, phallic complex) to allow more effective visualization and comparison.
Response 5: Apart from S. seinoi, for which we did not succeed in collecting a male specimen, the authors have presented the same set of structures across species. See figures 6-11, 13-18, on pages 20, 24, 27, 29, 32, 36, 40, 43, 46, 49, 52, and 55 of the revised manuscript for confirmation.
Comment 6: Discussion: The discussion is very detailed, but it currently addresses too many different issues without a clear structure, which makes it difficult to follow. I suggest reorganizing it into thematic sections (e.g., efficacy of DNA barcoding, taxonomic implications, phylogeographic patterns, broader significance). In particular, the distinction between Serpusia and Aresceutica could be clarified, as the variability reported within Serpusia seems to blur the diagnostic features just described. Similarly, some descriptive information (lists of BINs, localities, climate descriptions) might be better condensed, leaving more space for interpretation and synthesis.
Response 6: The discussion has been reorganized into thematic sections as suggested by the reviewer. As such, the efficacy of DNA barcoding is found on pages 61-62, Lines 1711-1774 of the revised manuscript; taxonomic implications follow on pages 62-63, Lines 1776-1824; and phylogeographic patterns can be found on pages 63-64, Lines 1826-1867 of the revised manuscript. However, the distinction between Serpusia and Aresceutica can not be highlighted in this work as it has already been done in the revision of the African grasshopper genus Areceutica by Rowell, Jago, and Hemp (2018). The authors only included Aresceutica species in the molecular analysis to verify their close relationship with Serpusia.
Comment 7: Following some other comments that should be addressed:
Line 24: Use the standard terminology “phallic complex” instead of “male phalli.”
Response 7: The terminology ´´phalli´´ refers to the plural of phallus. However, the authors have used the standard terminology ´´phallic complex´´ throughout the manuscript. Corrections are made on page 61, Line 1730; page 63, Line 1824
Comment 8: Line 61: The phrase “[5] discussed the close …” is difficult to follow. Provide the full citation (e.g., “Rowell et al. (5) discussed …”) rather than only the bracketed reference.
Response 8: The authors cannot provide the full citation in the manuscript. We used the citation Style Guide for MDPI ACS Journals as recommended in the MDPI instructions for authors. However, the statement has been changed to match the other citations. The corrections are highlighted on page 2, paragraph 2, Lines 62-65 of the revised manuscript.
Comment 9: Line 64: “[6] Synonymised Ptemoblax insidiosus Bolívar”, same comment as above: provide full citation for clarity when the name of the author is supposed to be read. When the sentence begins by referring to the author, as in this case, the author’s name should be written out rather than using the citation number [6], so that it reads clearly for the reader.
Response 9: Corrections are highlighted on page 2, paragraph 2, Lines 65-67 of the revised manuscript.
Comment 10: Lines 69–70: Rowell et al. (2018) did not formally transfer S. catamita, S. blanchardi, or S. inflata. Thus, the statement that “the genus Serpusia now contains only two species: S. opacula and S. succusor” is wrong. Currently, it contains five valid species. Introduction and discussion should be clarified to reflect this matter.
Response 10: The comments are very interesting. However, on page 38, Lines 1028-1048 of the revised manuscript, in the remarks under ´´the other species´´ section, the authors present reliably distinguishable characteristics between the Serpusia s.str. and the so-called S. blanchardi, S. catamita, and S. inflata. For instance, (1) the true Serpusia have vestigial or rudimentary forewings, whereas the so-called S. catamita and S. inflata have large and lobiform forewings; (2) the forewings in the true Serpusia have two shiny black spots, whereas there is always only one in S. catamita and S. inflata; (3) the posterior margin of the pronotum is distinctly incised and strongly emarginated in the middle, forming an acute angle in Serpusia. In contrast, it is rounded and distinctly wavy, with a light brown mottling in the middle in S. catamita, and more or less smooth and flat in S. inflata; (4) the ancorae of the epiphallus point directly inwards in Serpusia, whereas they are slanted forwards in S. catamita; (5) the lophi of the epiphallus are large, lobiform, and rounded in Serpusia, whereas they are curved upwards, with angular apices in S. catamita; (6) the phallic complex is robust and strongly sclerotised in S. catamita than in Serpusia; (7) the ejaculatory sac is sac is large, roughly covering the two-thirds of the endophallus and well covering the endophallic sclerites ventrally in Serpusia, whereas it is strongly reduced in S. catamita; and (8) the endophallic valves are flattened basally in Serpusia, whereas they are narrow basally in S. catamita. In addition, the tegmina in S. blanchardi are much reduced and not spotted as in true Serpusia; the hind femora are longer basally and less enlarged than in Serpusia; the antennae are shorter than the head and pronotum together in S. blanchardi, whereas they are longer than the head and pronotum together in Serpusia. The aforementioned morphological differences between the Serpusia s. str. and S. blanchardi, S. catamita, and S. inflata are consistent and therefore justify why these species should be removed from Serpusia and included in a separate genus, pending confirmation by genetic analysis.
Comment 11: Line 90: Revise “A total of hundreds of specimens” may be better to say “Hundreds of specimens” or “Several hundred specimens.”
Response 11: This sentence has been revised and the exact number of specimens examined specified on page 3, Line 90 of the revised manuscript.
Comment 12: Line 108: Please describe all measurements clearly and consistently, as you did for hind femur width. For example, specify body length, femur width, and any other structures measured, including how each measurement was taken.
Response 12: All measurements have been described consistently on page 3, Line 107-121 of the revised manuscript.
Comment 13: Table 1: The measurements are described in text, so there is no need to repeat them in Table 1.
Response 13: All the descriptions of the measurements have been deleted from this table, and this table, now Table 6, has been placed in the taxonomic section of the revised manuscript from page 58 to page 60.
Comment 14: Line 163: “69 ingroup taxa” these are specimens, not taxa. Clarify how many taxa are represented by the specimens included in the sequences.
Response 14: The sentence has been corrected accordingly. (see page 4, Line 133 of the revised manuscript).
Comment 15: Table 2: Please include locality information for each specimen, and ideally also provide georeferences, in addition to the collection dates.
Response 15: Including the requested information would make the table too large to fit on the A4 format. Therefore, the authors reserve the right to make that information available after a reasonable request to the corresponding authors.
Comment 16: Line 258: The sentence “This section may be divided by subheadings…” appears to be leftover template text (likely from the journal guidelines) rather than part of the manuscript. Please remove it.
Response 16: The sentence has been removed.
Comment 17: Line 277: The phrase “potential species” is unclear. Clarify whether it refers to undescribed species or putative lineages.
Response 17: The sentence has been revised on page 8, Line 288 and on page 16, Line 379.
Comment 18: Line 280: Use neutral wording for phylogenetic support: “shows” or “supports the recognition of” rather than “demonstrates.”
Responses 18: the suggestion has been incorporated accordingly on page 6, Line 235 and on page 8, Line 261.
Comment 19: Line 334: “Two recognised species and six hypothetical species”, use “putative” or “undescribed species” instead of “hypothetical.” In taxonomy, every species is essentially a hypothesis of an evolutionary lineage. I recommend replacing “hypothetical species” with a more precise term such as “putative species” or “undescribed species” to accurately reflect that these six lineages are inferred from your analyses but not yet formally described.
Response 19: The word ´´hypothetical´´ has been replaced in the text with ´´putative species´´ as recommended by the reviewer, on page 9, Line 288.
Comment 20: Line 368: Use consistent terminology: “specimens” instead of “individuals.”
Response 20: The consistent terminology has been used on page 12, Lines 335-337.
Comment 21: Line 416: The wording “to hypothetical species… on the curated tree” is unclear. Revise for clarity; what do you mean by curated tree?
Response 21: By a curated tree, the authors refer to the tree with full names of all newly described species. However, the terminology has been revised.
Comment 22: Line 483: Diagnosis: S. opacula bears resemblance to Paraserpusia succursor, but integrative delimitation procedure is not clear. Add details on morphological vs. molecular separation.
Response 22: The statement has been revised on page 19, Line 453. However, we justified as followed: Serpusia opacula and P. succursor form quite distinct COI-16S clades, and do not overlap geographically as S. opacula occurs exclusively in the Coastal and South-west regions of Cameroon (Mouanko, Barombi station, and Buea) and P. succursor in the Centre region of Cameroon (Ongot). Both also exhibit discrete forewing and genitalia differences—thus supporting their recognition as separate species.
Comment 23: Line 495 / 528: “Ectophallic sheath elongated (vs. short in P. succursor)” – elongated is not precisely the opposite of short. Clarify or quantify.
Response 23: The terminology has been clarified throughout the manuscript. See page 19, Line 476; page 21, Line 519; page 22, Line 558; page 25, Line 669; page 30, Line 798; page 33, Line 897.
Comment 24: Line 581: “Cingular valves, aedeagus, and gonopore sac” – not visible in photos. Maybe and indication to the structures in the plates would help the reader to identify the structures.
Response 24: The structures have been clearly indicated in the plates whenever possible, and the explanations of the structures are given and highlighted in the figure titles. Corrections can be found on figures 6-11, 13-18, on pages 20, 24, 27, 29, 32, 36, 40, 43, 46, 49, 52, and 55 of the revised manuscript.
Comment 25: Line 676: Statements like “identical to S. opacula” contradict differences listed. Revise wording for consistency.
Response 25: The sentence has been revised on page 25, Line 657.
Comment 26: Line 689: “The head is straight” – specify view (dorsal, lateral, frontal).
Response 26: The statement has been revised accordingly on page 25, Line 670.
Comment 27: Line 922: Clarify whether original descriptions of S. blanchardi and S. inflata are transcribed or summarized.
Response 27: The original descriptions S. blanchardi and S. inflata are repeated and transcribed in this work. The corrections are made on pages 33-34, Lines 913-914.
Comment 28: Line 1079. “apodemes of cingular” should read apodemes of cingulum
Response 28: The terminology has been revised on page 19, Lines 459-460; page 21, Line 510; page 22, Line 555; page 23, Line 591; page 25, Lines 650, 663; page 26, Lines 690, 702; page 30, Line 795; page 31, Lines 824, 837; page 33, Lines 897, 901; page 35, Line 958; page 39, Lines 1079, 1080; page 41, Lines 1123, 1124, 1138; page 42, Lines 1193, 1200; page 44, Line 1258; page 45, Lines 1283, 1293; page 47, Line 1351; page 48, Lines 1377, 1388; page 51, Lines 1467, 1476; page 53, Lines 1518, 1530, 1544; page 54, Line 1576.
Comment 29: Line 1600: In the key, distinctions like “fairly reaching” vs. “barely reaching” are subtle and subjective. Revise for clarity and reproducibility.
Response 29: The sentences have been revised on page 57, Lines 1634, 1636.
Comment 30: Line 1662. “Therefore, we removed Serpusia succursor from the genus Serpusia and included it in a new genus that we erected, Paraserpusia gen. nov.” I assume that the authors are referring to the Serpusia succursor species complex.
Response 30: The authors are referring to the S. succursor species complex. This has been revised in the text on page 63, Line 1788.
Comment 31: Line 1730. The use of “signalisation” is not clear in English. I suggest revising the sentence for clarity. For example: “Molecular analyses revealed that the identification by [44] was in fact a misidentification, corresponding to the hidden species Paraserpusia kekeunoui sp. nov. and Paraserpusia tamesei sp. nov., respectively.”
Response 31: The sentence has been revised accordingly on page 64, lines 1847.
Comment 32: Line 1739. “However, [44] reported Serpusia succursor (now Paraserpusia succursor) in Buea, Kumba, and Tombel in the Southwest region of Cameroon. The species reported by these authors was Serpusia opacula rather than 'Serpusia succursor'. It is not clear why the authors are confident that the record by [44] represents a misidentification. Please clarify what evidence supports this conclusion (e.g., re-examination of specimens, molecular data, or comparison with diagnostic characters). Without specifying the basis, the statement remains speculative.
Response 32: The record by [44] represents a misidentification, as specimens collected by [44] from Buea, Kumba, and Tombel were also examined in this work. Clarification is mentioned on page 64, lines 1857-1858.
Comment 33: Line 1756. This paragraph is somewhat confusing. At first, it seems intended to establish clear morphological differences between Serpusia and Aresceutica, but then it shifts to emphasize variability within Serpusia (in the cingular valves and aedeagus), which blurs the distinction. I suggest clarifying whether the main point is (i) to highlight the diagnostic characters that separate the two genera, or (ii) to discuss morphological variability within Serpusia.
Response 33: The paragraph has been revised on page 63, Lines 1792-1818.
Language has also been edited and is highlighted in light blue in the revised manuscript.
Yours sincerely
Reviewer 2 Report
Comments and Suggestions for Authors
The manuscript is generally well written and presents a commendable effort in terms of systematic work. The quality of the descriptions and redescriptions—each preceded by a diagnosis—deserves recognition, as this level of rigor has unfortunately become uncommon in recent taxonomic literature. I also appreciate the explicit indication of absent characters (e.g., absence of a black spot), a detail too often overlooked in contemporary, hastily prepared descriptions. Similarly, the identification keys are clear and well-constructed—an excellent contribution.
Specific comments
Overlap between Methods and Results
Some elements currently placed in the Methods section belong more appropriately to the Results. For instance:
-Entire Table 1
-Lines 173–179
These should be relocated to maintain structural clarity.
Figures and visual consistency
-Please use the same color code for Figure 19 and the phylogenetic tree (Figure 1) to facilitate reader comprehension.
-Several figures (e.g., Figures 11 and 12) exhibit apparent color balance issues that should be corrected.
-Scales in all figures require careful verification. Most currently indicate 5 mm, which is almost certainly incorrect.
Results section adjustments
-Lines 257–260 should be removed.
-Line 1486: remove the capital letter in succursor.
Major comment: workflow and taxonomic rationale
As is often the case in integrative taxonomy, the workflow is not clearly articulated. For transparency and reproducibility, the methodology should explicitly state all steps in the sequence they were performed. This organization should then be mirrored in the Results section, prior to the taxonomic treatment. A suggested logical framework is as follows:
Specimen Selection
How were specimens selected? Was the basis morphological, geographical, or both? The text suggests an initial grouping into six morphospecies, but this needs to be explicitly clarified.
Molecular Analysis and Candidate Species Delimitation
The use of COI (660 bp) as the sole marker provides a useful evolutionary perspective, but also carries limitations, particularly the risk of artifacts when sampling is geographically sparse. These caveats should be acknowledged.
Morphological Examination of Candidate Species
Detail the correspondence between specimens examined morphologically and those subjected to genetic analysis. This is crucial to understanding whether both datasets refer to the same individuals or different ones.
Taxonomic decisions and justifications
Gaps in geographic sampling can artifactually produce distinct MOTUs within widely distributed species, potentially leading to misinterpretation of morphological variation (e.g., ecotypes).
I strongly recommend including a pre-taxonomy paragraph summarizing the rationale behind each species decision, integrating molecular, morphological, and geographic evidence. For example:
Paraserpusia succursor and P. tamessei form distinct COI clades, and geographic overlap rules out a sampling artifact. Both also exhibit discrete forewing and genitalia differences—thus supporting their recognition as separate species.
A similar analysis should be provided for other potentially contentious cases, such as Serpusia missoupi vs. S. verhaaghi, and all specimens formerly assigned to S. opacula.
In situations of uncertainty, a conservative approach (avoiding the proliferation of weakly supported names) may be preferable.
Overall, this is a valuable and well-executed contribution to integrative taxonomy. Addressing the points above—particularly clarifying the methodological workflow, ensuring consistency in figure presentation, and explicitly justifying taxonomic decisions—will significantly enhance the manuscript’s clarity, rigor, and long-term utility.
Author Response
Dear Reviewer, thank you for your valuable comments on our manuscript.
Response to Reviewer 2.
Remarks: All the responses to Reviewer 2's comments are highlighted in purple in the revised manuscript.
Comment 1: Overlap between Methods and Results
Some elements currently placed in the Methods section belong more appropriately to the Results. For instance:
-Entire Table 1
-Lines 173–179
These should be relocated to maintain structural clarity.
Response 1: Table 1 has been relocated to the results section. As such, the order of the tables has been changed throughout the manuscript. Table 1 in the previous version of the manuscript, now Table 6, has been placed in the taxonomic section of the revised manuscript from page 58 to page 60.
Comment 2: Figures and visual consistency. Please use the same color code for Figure 19 and the phylogenetic tree (Figure 1) to facilitate reader comprehension.
Response 2: The same color code has been used for the phylogenetic tree (Figure 1) and the distribution map (Figure 19). The revsed Figure 19 can be found on page 56.
Comment 3: Several figures (e.g., Figures 11 and 12) exhibit apparent color balance issues that should be corrected.
Response 3: All images except for images presented in Figure 12 were taken/made by us. While those in Figure 12 were made in part by the Natural History Museum, London, UK, and in part by the Museum of Natural Science, Madrid, Spain. The authors are unable to modify those images as they receive some restrictions regarding any type of image editing.
Comment 4: Scales in all figures require careful verification. Most currently indicate 5 mm, which is almost certainly incorrect.
Response 4: The scales of 5 mm and 10 mm, corresponding respectively to S. blanchardi and S. inflata, were made by the Museum of Natural Sciences, Madrid, and the Natural History Museum of London, UK. The authors have no right to modify or edit any of these images. The authors confirm that the remaining figures indicating the scale of 5mm have been taken by the authors, and they are correct.
Results section adjustments
Comment 5: Lines 257–260 should be removed.
Response 5: The statement has been removed.
Comment 6: Line 1486: remove the capital letter in succursor.
Response 6: The species name has been revised on page 53, Line 1513.
Comment 7: Major comment: workflow and taxonomic rationale
As is often the case in integrative taxonomy, the workflow is not clearly articulated. For transparency and reproducibility, the methodology should explicitly state all steps in the sequence they were performed. This organization should then be mirrored in the Results section, prior to the taxonomic treatment. A suggested logical framework is as follows:
Specimen Selection
How were specimens selected? Was the basis morphological, geographical, or both? The text suggests an initial grouping into six morphospecies, but this needs to be explicitly clarified.
Response 7: This has been clarified accordingly in the text on page 4, Lines 134-135.
Comment 8: Molecular Analysis and Candidate Species Delimitation
The use of COI (660 bp) as the sole marker provides a useful evolutionary perspective, but also carries limitations, particularly the risk of artifacts when sampling is geographically sparse. These caveats should be acknowledged.
Response 8: The authors did not use the COI alone, but they used both the COI-5P and the 16S rDNA markers for the Phylogenetic analysis and for the species delimitation by ABGD.
Comment 9: Morphological Examination of Candidate Species
Detail the correspondence between specimens examined morphologically and those subjected to genetic analysis. This is crucial to understanding whether both datasets refer to the same individuals or different ones.
Response 9: All the specimens examined morphologically (101 in total) were subjected to genetic analysis. However, the COI-5P of 78 specimens and the 16S rDNA of 73 specimens were successfully sequenced and used for the genetic analysis. Corrections are made on page 6, Line 219-220.
Comment 10: Taxonomic decisions and justifications
Gaps in geographic sampling can artifactually produce distinct MOTUs within widely distributed species, potentially leading to misinterpretation of morphological variation (e.g., ecotypes).
I strongly recommend including a pre-taxonomy paragraph summarizing the rationale behind each species decision, integrating molecular, morphological, and geographic evidence. For example: Paraserpusia succursor and P. tamessei form distinct COI clades, and geographic overlap rules out a sampling artifact. Both also exhibit discrete forewing and genitalia differences—thus supporting their recognition as separate species.
A similar analysis should be provided for other potentially contentious cases, such as Serpusia missoupi vs. S. verhaaghi, and all specimens formerly assigned to S. opacula.
In situations of uncertainty, a conservative approach (avoiding the proliferation of weakly supported names) may be preferable.
Response 10: Thank you for pointing this out. We agree and the explanations are as followed: Although P. succursor and P. tamessei populations overlap geographically, they form distinct COI-16S clades, and both also exhibit discrete forewing and genitalia differences—thus supporting their recognition as separate species. Paraserpusia tindoi and P. hoeferi form distinct COI-16S clades, and do not overlap geographically as P. tindoi occurs exclusively in the Eastern region of Cameroon (Somalomo) and P. hoeferi in the Littoral region of Cameroon (Iboti). Both also exhibit discrete forewing and genitalia differences—thus supporting their recognition as separate species. Paraserpusia kekeunoui occurs exclusively in Mbalmayo (centre region of Cameroon), whereas P. husemanni is restricted to the southern region of Cameroon (Akom 2). Both species form distinct COI-16S clades, and exhibit discrete forewing and genitalia differences—thus supporting their recognition as separate species. Additionally, although S. missoupi and S. verhaaghi overlap geographically (both species being found in the Littoral region of Cameroon), they exhibit discrete forewing and genitalia differences, and form well-supported clades in the phylogenetic tree inferred from a combined COI and 16S dataset—thus supporting their recognition as separate species. Serpusia kennei is geographically distributed in the Littoral region of Cameroon, whereas S. seinoi occurs exclusively in the Eastern region of Cameroon. Both species exhibit discrete forewing and form well-supported clades in the phylogenetic tree inferred from a combined COI and 16S dataset—thus supporting their recognition as separate species. These corrections are made on pages 60-61, Lines 1694-1712.
Reviewer 3 Report
Comments and Suggestions for Authors
This article constitutes an important contribution to the taxonomy and systematics of African Grasshoppers (Catantopinae), using DNA barcoding data and molecular analysis, in addition to traditional morphological techniques.
Overall, the article is well written and presented, minor corrections are highlighted in the manuscript and are included in this revision.
It would be beneficial to the discussion to address questions about how cryptic species or cryptic speciation arise and why and how their conservation is important from a biodiversity perspective.
In the References, the year of publication appears in bold in most references. However, in some cases it is not.
August/13/2025
Comments to: Integrative taxonomy revealed cryptic diversity in the West African grasshopper genus Serpusia Karsch, 1891 (Orthoptera: Catantopinae)
Introduction
- 67. formally transferred Ptemoblax lemarineli… transferred to
Materials and methods
L. 90. A total of hundreds of specimens. Please provide the total number of specimens examined.
L. 94. Serpusia inflata. S. inflata
L. 106-107. All measurements are expressed in millimeters (mm). Referred to the table, is it Table 1?
L. 176-178. Aresceutica morogorica Dirsh, 1984, Aresceutica nguruensis Rowell, Jago & Hemp, 2018, Aresceutica subnuda Karsch, 1896, Pteropera carnapi Ramme, 1929, Pteropera kennei Yetchom & Husemann, 2024…
Suggest: Aresceutica morogorica Dirsh, 1984, A. nguruensis Rowell, Jago & Hemp, 176 2018, A. subnuda Karsch, 1896, Pteropera carnapi Ramme, 1929, P. kennei 177 Yetchom & Husemann, 2024…
Results
- L. Aresceutica Nguruensis. Change to Aresceutica nguruensis
- 385. Aresceutica subnuda. Change to A. subnuda
- Figure 6A–L. It is suggested that each internal genitalia structure be indicated and named to help less experienced readers compare between the genera and species described.
- Sepusia opacula. Change to Serpusia opacula
- It is suggested that each internal genitalia structure be indicated and named to help less experienced readers compare between the genera and species described.
- cingular. Change to cingulum? Please check throughout the manuscript that the name of the phallic complex structure is correct…
- Epiphallus (Figure 6H). It is suggested that each structure be indicated and named in the figure.
- Cingulum instead of cingular?
- (J change to (J)
- Cingular. cingulums
- Figure 11. Serpusia catamita sp. nov. Please verify, this species is reported as a valid species, previously described, so far.
- The remaining 1 m of length is similar to that of the male. This sentence is no correct or no clear
- Serpusia catamita, Serpusia blanchardi, and Serpusia inflata. Change to Serpusia catamita, S. blanchardi, and S. inflate…
Line. 1041. S. catamita
1051-1052. Serpusia inflata. Change to S. inflate
1082. and; an
L.1209. bar ground, bare ground?
1231. straight pex; straight apex
- Serpusioides surccusor. Paraserpusia succursor?
- Aresceutica’’
- Serpusia blanchardi, Serpusia catamita, and Serpusia inflate. Change to: Serpusia blanchardi, S. catamita, and S. inflata…
- Outer morphology; external morphology?
References
- Orthoptera Species File 2024. Please update this reference

Author Response
Dear Reviewer, thank you for your valuable comment on our manuscript. Please respond to your comments below.
Response to Reviewer 3
Remarks: All the responses to Reviewer 3's comments are highlighted in light blue in the revised manuscript.
Comment 1: It would be beneficial to the discussion to address questions about how cryptic species or cryptic speciation arise and why and how their conservation is important from a biodiversity perspective.
Response 1: We agree with Reviewer 3 regarding this valuable comment. The reviewers´ suggestion has been added to the discussion. Corrections can be found on page 62, Lines 1751-1771. Additionally, the references listed in this section have been incorporated into the reference list on page 66, Lines 1997-2017.
Comment 2: In the References, the year of publication appears in bold in most references. However, in some cases it is not.
Response 2: The correction has been made on page 66, Lines 1944, 1947, 1948, 1951.
Comments to: Integrative taxonomy revealed cryptic diversity in the West African grasshopper genus Serpusia Karsch, 1891 (Orthoptera: Catantopinae)
Introduction
Comment 3: Line 67. formally transferred Ptemoblax lemarineli… transferred to
Response 3: The statement has been revised accordingly on page 2, Line 66.
Materials and methods
Comment 4: Line 90. A total of hundreds of specimens. Please provide the total number of specimens examined.
Response 4: The total number of specimens examined has been provided on page 3, Line 90 of the revised manuscript.
Comment 5: Line 94. Serpusia inflata. S. inflata
Response 5: The species name has been revised on page 3, Line 94.
Comment 6: Line 106-107. All measurements are expressed in millimeters (mm). Referred to the table, is it Table 1?
Response 6: This is revised on page 3, Lines 108 and 122. This table, now Table 6, has been placed in the taxonomic section of the revised manuscript from page 58 to page 60.
Comment 7: Line 176-178. Aresceutica morogorica Dirsh, 1984, Aresceutica nguruensis Rowell, Jago & Hemp, 2018, Aresceutica subnuda Karsch, 1896, Pteropera carnapi Ramme, 1929, Pteropera kennei Yetchom & Husemann, 2024…
Suggest: Aresceutica morogorica Dirsh, 1984, A. nguruensis Rowell, Jago & Hemp, 176 2018, A. subnuda Karsch, 1896, Pteropera carnapi Ramme, 1929, P. kennei 177 Yetchom & Husemann, 2024…
Response 7: The corrections have been incorporated accordingly on page 4, Lines 139-140. However, we maintain Segellia nitidula in the sentence ´´Segellia nitidula was used as the root of the phylogeny" since it is placed at the beginning of the sentence.
Results
Comment 8: Line 370. L. Aresceutica Nguruensis. Change to Aresceutica nguruensis
Response 8: The changes have been applied on page 12, Line 336.
Comment 9: Line 371. Aresceutica subnuda. Change to A. subnuda
Response 9: The changes have been applied on page 12, Line 336.
Comment 10: Line 372. Figure 6A–L. It is suggested that each internal genitalia structure be indicated and named to help less experienced readers compare between the genera and species described.
Response 10: The structures have been indicated on the plates where relevant, and the explanations of the structures are given and highlighted in the figure titles. Corrections can be found on figures 6-11, 13-18, on pages 20, 24, 27, 29, 32, 36, 40, 43, 46, 49, 52, and 55 of the revised manuscript.
Comment 11: Line 373. Sepusia opacula. Change to Serpusia opacula
Response 11: The changes have been applied in the text on page 12, Line 335.
Comment 12: Line 374. It is suggested that each internal genitalia structure be indicated and named to help less experienced readers compare between the genera and species described.
Response 12: The structures have been indicated in the plates, and the explanations of the structures are given and highlighted in the figure titles. Corrections can be found on figures 6-11, 13-18, on pages 20, 24, 27, 29, 32, 36, 40, 43, 46, 49, 52, and 55 of the revised manuscript.
Comment 13: Line 375. cingular. Change to cingulum? Please check throughout the manuscript that the name of the phallic complex structure is correct…
Response 13: The correction has been made throughout the manuscript, specifically on page 19, Lines 459-460; page 21, Lines 509-510; page 23, Line 591; page 26, Lines 688, 690, 702; page 31, Lines 824, 837; page 33, Lines 894, 897, 900-901; page 35, Lines 959-962; page 39, Lines 1079-1080, 1092; page 41, Lines 1123-1124, 1138; page, 42, Lines 1191, 1193, 1200; page 44, Lines 1258; page 45, Lines 1281, 1283, 1293-1294; page 47, Lines 1336, 1352; page 48, Lines 1375,1377, 1388; page 53, Lines 1518-1519, 1530, 1544; page 54, Lines 1573, 1576, 1583 page 57, Lines 1654, 1657, 1659.
Comment 14: Line 376. Epiphallus (Figure 6H). It is suggested that each structure be indicated and named in the figure.
Response 14: The structures have been indicated on the plates, and the explanations of the structures are given and highlighted in the figure titles. Corrections can be found on figures 6-11, 13-18, on pages 20, 24, 27, 29, 32, 36, 40, 43, 46, 49, 52, and 55 of the revised manuscript.
Comment 15: Line 377. Cingulum instead of cingular?
Response 15: The correction has been made, specifically on page 19, Lines 459-460; page 21, Lines 509-510; page 23, Line 591; page 26, Lines 688, 690, 702; page 31, Lines 824, 837; page 33, Lines 894, 897, 900-901; page 35, Lines 959-962; page 39, Lines 1079-1080, 1092; page 41, Lines 1123-1124, 1138; page, 42, Lines 1191, 1193, 1200; page 44, Lines 1258; page 45, Lines 1281, 1283, 1293-1294; page 47, Lines 1336, 1352; page 48, Lines 1375,1377, 1388; page 53, Lines 1518-1519, 1530, 1544; page 54, Lines 1573, 1576, 1583 page 57, Lines 1654, 1657, 1659..
Comment 16: Line 378. (J change to (J)
Response 16: The correction has been made on page 25, Line 627.
Comment 17: Line 379. Cingular. Cingulums
Response 17: The correction has been made, specifically on page 19, Lines 459-460; page 21, Lines 509-510; page 23, Line 591; page 26, Lines 688, 690, 702; page 31, Lines 824, 837; page 33, Lines 894, 897, 900-901; page 35, Lines 959-962; page 39, Lines 1079-1080, 1092; page 41, Lines 1123-1124, 1138; page, 42, Lines 1191, 1193, 1200; page 44, Lines 1258; page 45, Lines 1281, 1283, 1293-1294; page 47, Lines 1336, 1352; page 48, Lines 1375,1377, 1388; page 53, Lines 1518-1519, 1530, 1544; page 54, Lines 1573, 1576, 1583 page 57, Lines 1654, 1657, 1659..
Comment 18: Line 380. Figure 11. Serpusia catamita sp. nov. Please verify this species is reported as a valid species, previously described, so far.
Response 18: The species name has been revised on page 35, Line 968.
Comment 19: Line 381. The remaining 1 m of length is similar to that of the male. This sentence is not correct or not clear.
Response 19: This sentence was deleted.
Comment 20: Line 382. Serpusia catamita, Serpusia blanchardi, and Serpusia inflata. Change to Serpusia catamita, S. blanchardi, and S. inflate…
Response 20: The correction has been made on page 38, Line 1029. However, all species names placed at the beginning of the sentence are written in full.
Comment 21: Line 1041. S. catamita
Response 21: The correction has been made on page 38, Line 1041.
Comment 22: Line 1051-1052. Serpusia inflata. Change to S. inflate
Response 22: The correction has been made on page 38, Line 1051.
Comment 23: Line 1082. and; an
Response 23: The correction has been made on page 39, Line 1082.
Comment 24: Line 1209. bar ground, bare ground?
Response 24: The correction has been made on page 44, Line 1225.
Comment 25: Line 1231. straight pex; straight apex
Response 25: The correction has been made on page 44, Line 1240.
Comment 26: Line 1666. Serpusioides surccusor. Paraserpusia succursor?
Response 26: The species name has been revised on page 63, Line 1781.
Comment 27: Line 1667. Aresceutica’’
Response 27: The species name has been revised on page 62, Line 1784.
Comment 28: Line 1668.
- Serpusia blanchardi, Serpusia catamita, and Serpusia inflate. Change to: Serpusia
Response 28: The changes have been applied on page 64, Line 1828.
Comment 29: Line 1669.
- Outer morphology; external morphology?
Response 29: The suggestion has been incorporated throughout the text on page 63, Lines 1797 and 1809.
References
Comment 30: Line 1825. Orthoptera Species File 2024. Please update this reference
Response 30: This reference has been updated on page 65, Lines 1915-1916.
The corrections related to the language editing have also been highlighted in light blue.
Yours sincerely.
Round 2
Reviewer 1 Report
Comments and Suggestions for Authors
I am pleased with the revised version of the manuscript and do not have further comments to the authors